# The Real Tropical Geometry of Neural Networks for Binary Classification

**Marie-Charlotte Brandenburg** *brandenburg@kth.se*
*KTH Royal Institute of Technology*

**Georg Loho** *georg.loho@math.fu-berlin.de*
*University of Twente & Freie Universität Berlin*

**Guido Montúfar** *montufar@math.ucla.edu*
*University of California, Los Angeles & Max Planck Institute MiS*

**Reviewed on OpenReview:** *https://openreview.net/forum?id=I7JWf8XA2w*

## Abstract

We consider a binary classifier defined as the sign of a tropical rational function, that is, as the difference of two convex piecewise linear functions. In particular, the set of functions represented by a ReLU neural network can be regarded as a subset in the parameter space of tropical rational functions, specifically, it is contained as a semialgebraic set. We initiate the study of two different subdivisions of the parameter space of tropical rational functions with fixed number of terms in the numerator and denominator: a subdivision into semialgebraic sets, on which the combinatorial type of the decision boundary is fixed, and a subdivision into a polyhedral fan, capturing the combinatorics of the partitions of the dataset. The sublevel sets of the 0/1-loss function arise as subfans of this classification fan, and we show that the level-sets are not necessarily connected. We describe the classification fan i) geometrically, as normal fan of the activation polytope, and ii) combinatorially through a list of properties of associated bipartite graphs, in analogy to covector axioms of oriented matroids and tropical oriented matroids. Our findings extend and refine the connection between neural networks and tropical geometry by observing structures established in real tropical geometry, such as positive tropicalizations of hypersurfaces and tropical semialgebraic sets.

## 1 Introduction

We consider a *binary classification task* with hypotheses given by signs of real-valued functions parameterized by artificial neural networks with piecewise linear activation functions. Given a classification task, we are interested in the sets of parameters for which the network perfectly classifies the training data, or for which it makes a certain number of errors, that is, the level sets of the *0/1-loss function*. We seek to understand the combinatorial and discrete-geometric structures underlying such a classification task using polyhedral methods. One of our aims is to enhance the synergies between neural networks and real tropical geometry to facilitate progress in both communities by translating some concepts and results.

The combinatorics of the functions represented by neural networks with piecewise linear activations has received significant attention over the years, e.g., in the works of Pascanu et al. (2014); Montúfar et al. (2014); Telgarsky (2016); Raghu et al. (2017); Serra et al. (2018); Balestriero et al. (2019), or works listed in the overview article of Huchette et al. (2023). A new trend in theoretical considerations of (deep) neural networks with piecewise linear activation functions is to study them through the lens of *tropical geometry*, a mathematical framework which is tailored to understand the geometry of piecewise linear functions. The language of tropical geometry in the context of binary classification already appeared in the work of Charisopoulos & Maragos (2017), and the relation to feedforward neural networks with ReLU activation

was expanded independently by Charisopoulos & Maragos (2018) and Zhang et al. (2018) and to maxout networks by Charisopoulos & Maragos (2018) and Montúfar et al. (2022). The connection arises through the fact that any function represented by such a neural network is continuous piecewise linear (and vice versa, Arora et al. 2018) and can thus be written as the difference of two convex piecewise linear functions (Hartman, 1959; Wang, 2004). Such a difference is called a *tropical rational function*. The representation as such a difference is not unique, and has been studied for instance by Kripfganz & Schulze (1987) and Melzer (1986); Schlüter & Darup (2020); Tran & Wang (2023).

In recent years, there has been increased interest in *real tropical geometry*, which also encompasses sign information and aims to study tropical varieties inside all orthants (see, e.g., Jell et al. 2020 and Rau et al. 2022). This is the perspective that we take in the present article. Concretely, we consider differences $g - h$ of two convex (and continuous) piecewise linear functions $g, h$, where we fix the maximum possible number of linear terms of each, $g$ and $h$.

The simplest instance of this setup is a *linear classifier*, which is the special case where $g$ and $h$ are linear functions. Here, results have been rediscovered in many different mathematical communities. The well-known work of Cover (1964) studied the structure of the space of parameters of linear classifiers, which is subdivided by an arrangement of hyperplanes, and described the solution set of the classification task as a polyhedral cone. These structures have been expanded widely by the combinatorics community in the years since. One notable example is the counting formula for chambers in a general hyperplane arrangement by Zaslavsky (1975), and the development of oriented matroids (Björner et al., 1999), which are combinatorial objects that capture the combinatorial structure of the data and their corresponding hyperplane arrangements. Linear classifiers can be modeled by linear networks, which (over-)parametrize linear functions as compositions of linear functions, one for each layer of the network. The space of parameters of linear functions that can be represented by a linear network consists of matrices whose rank is at most equal to the minimum width of any of the network layers, which is an algebraic variety inside the space of matrices. In analogy, in this article we describe the space of parameters of piecewise linear functions that can be represented by a ReLU neural network as a semialgebraic set inside the space of tropical rational functions.

In general, the combinatorial structure of the parameter space refers to its subdivision into regions representing different dichotomies of the input data. Such subdivisions have been considered in several classic works studying the growth function and Vapnik-Chervonenkis dimension of binary classifiers (Vapnik & Chervonenkis, 1971); see Anthony & Bartlett (1999, Part I) and references. These subdivisions naturally interact with the level sets of the training loss. The *loss landscape* of neural networks has been studied in a series of works, obtaining results on the connectivity of sublevel sets or the existence of paths descending to a global minimum. For instance, under some conditions it is known that if the data is linearly separable then all local minima are global minima (Gori & Tesi, 1992). In general, it has been observed that local minima are not necessarily global minima (Brady et al., 1989; Sontag & Sussmann, 1989) and, depending on the loss function, the number of local minima may grow exponentially in the input dimension even for single neurons (Auer et al., 1995). For networks with piecewise linear activations, spurious local minima are common (Safran & Shamir, 2018; Liu, 2022). This can be rectified under a catalogue of assumptions. In particular, it can be shown that, given an appropriate level of overparametrization, most differentiable local minima are global minima (Soudry & Hoffer, 2018; Karhadkar et al., 2024). Various works have studied how any two parameters can be connected through a continuous path in which the energy gap remains bounded (Freeman & Bruna, 2017; Kuditipudi et al., 2019; Nguyen et al., 2021). We also note a stream of works (Wang et al., 2022; Mishkin et al., 2022; Mishkin & Pilanci, 2023; Ergen & Pilanci, 2021; Matena & Raffel, 2022) which considers the solution sets in terms of duality and the solution set for convex reformulations of the non-convex optimization problem. Further, the symmetries of neural network parametrization maps have been studied for instance by Rolnick & Kording (2020) and Grigsby et al. (2023) and also in relation to the optimization landscape by Simsek et al. (2021).

For classification with piecewise linear functions, not only the parameter space exhibits a (not necessarily polyhedral) subdivision, but also the input space exhibits a polyhedral subdivision, which is induced by the *decision boundary*. To this day, understanding the decision boundary of deep neural networks remains a difficult task both in theory and in practice (Humayun et al., 2023). For a ReLU neural network, it is known that the decision regions and the decision boundary are unbounded whenever the width of every layer is

at most equal to the input dimension $d$ (Beise et al., 2021; Johnson, 2019; Nguyen et al., 2018), and if the network consists of a single hidden layer of width $d + 1$ then the decision regions can have no more than one bounded connected component (Grigsby & Lindsey, 2022). The connection between decision boundaries and tropical geometry is straightforward through subcomplexes of tropical hypersurfaces (Zhang et al., 2018; Charisopoulos & Maragos, 2018; Alfarra et al., 2023; Piwek et al., 2023). Notably, this subcomplex is a familiar object in *real tropical geometry* as the (signed-)positive tropicalization of a hypersurface over the field of real or complex Puiseux series (Viro, 2006; Speyer & Williams, 2005; Brandenburg et al., 2023). This connection to real tropical geometry goes beyond decision boundaries: In parameter space we observe that the set of solutions of a classification task can be formulated as a tropical semialgebraic set, a class of sets which has only recently started to enjoy systematic considerations (Allamigeon et al., 2020; Jell et al., 2020).

**Contributions.** In this article, we consider the combinatorics of continuous piecewise linear classifiers. We consider a finite data set $\mathcal{D} \subseteq \mathbb{R}^d$ and classifiers defined as signs of functions $g - h \colon \mathbb{R}^d \to \mathbb{R}$, where $g$ and $h$ are convex and continuous piecewise linear functions with at most $n$ and $m$ linear pieces, respectively. We initiate the study of two different subdivisions of the parameter space $\Theta(d, n, m)$ of such classifiers: The first is a subdivision of the parameter space into semialgebraic sets where the combinatorial type of the decision boundary is fixed. The second is a polyhedral subdivision, called the *classification fan*. This is obtained from the *activation fan*, a polyhedral fan which captures the *activation patterns* of the tropical rational function. The different possible dichotomies of the data and the sublevel sets of the 0/1-loss function arise as subfans of the classification fan. We describe the fan i) geometrically, as the normal fan of the *activation polytope*, and ii) combinatorially through a list of properties of bipartite graphs associated with the activation patterns in analogy to covector axioms of oriented matroids. From this we can show that in the separable case the sublevel sets of the 0/1-loss function are connected for linear classifiers, but are disconnected for general piecewise linear classifiers. Furthermore, we show that the parameter space of a fixed architecture of ReLU neural networks is a semialgebraic set of bounded degree inside the parameter space of tropical rational functions, and thus intersects both subdivisions of $\Theta(d, n, m)$ nontrivially.

This article aims to address multiple audiences: A data-driven audience with a background in machine learning interested in understanding the underlying combinatorics, and a (discrete) geometric audience with an interest towards applications concerning neural networks. We try to accommodate for the different backgrounds throughout the exposition.

**Overview.** In Section 2 we begin by recalling known results and concepts describing linear classification. We introduce three different points of view to study natural subdivisions in parameter space: As chambers of a hyperplane arrangement $\mathcal{H}_{\mathcal{D}}$ (i.e., connected components of the complement of $\mathcal{H}_{\mathcal{D}}$), as cones in the normal fan $\Sigma_{\mathcal{D}}$ of a polytope $P_{\mathcal{D}}$, and as maximal covectors of a realizable oriented matroid. The subsequent sections are devoted to generalizing this theory to classification by continuous piecewise linear functions. In Section 3 we describe the connection of continuous piecewise linear functions to ReLU neural networks and tropical geometry. While in the linear case the decision boundary is a hyperplane, in Section 4 the decision boundary is piecewise linear. In Section 5 we introduce the activation polytope and the activation fan, a polytope and its normal fan which are in analogy to the fan $\Sigma_{\mathcal{D}}$ and $P_{\mathcal{D}}$ from the linear case. Each cone is labeled by an activation pattern, and we relate the set of activation patterns to sets of covectors of oriented and tropical oriented matroids. In Section 6 we consider subdivisions of the parameter space of continuous piecewise linear functions: We first introduce the classification fan (Section 6.1). Afterwards (Section 6.2) we describe an arrangement of indecision surfaces, which is the natural analog of the hyperplane arrangement $\mathcal{H}_{\mathcal{D}}$, and the cells of this arrangement are compatible with the activation and classification fan. We show that the sublevel sets of the 0/1-loss can be viewed as subfans of the classification fan, resulting in a study of the perfect classification fan (Section 6.3) and the (sub-)level sets of the 0/1-loss function (Section 6.4).

## 2  Linear Classifiers

We motivate our studies from classical results about linear classification problems. Many results presented in this section have been rediscovered in different mathematical communities, using the language of classifiers,

hyperplane arrangements or realizable oriented matroids. These different viewpoints serve as an inspiration for what follows, as in the upcoming sections we will generalize this to a theory for classifiers defined by neural networks with ReLU activation and tropical rational functions.

Let $\mathcal{D} = \{\mathbf{p}_1, \ldots, \mathbf{p}_M\} \subset \mathbb{R}^d$ be a finite set of data points. A *linear binary classifier* or *simple perceptron* is a function $f_\theta \colon \mathbb{R}^d \to \{-1, 0, 1\}$, $f_\theta(x) = \operatorname{sgn}(\langle \mathbf{s}, x \rangle + a)$, defined by taking the sign of a linear function with parameters $\theta = (a, \mathbf{s})$, where $a \in \mathbb{R}, \mathbf{s} \in \mathbb{R}^d$. The parameter space is $\Theta(d) = \{\theta = (a, \mathbf{s}) \mid a \in \mathbb{R}, \mathbf{s} \in \mathbb{R}^d\} \cong \mathbb{R}^{d+1}$. The function $f_\theta$ defines an affine hyperplane which separates the data points into three classes, $\{\mathbf{p} \in \mathcal{D} \mid f_\theta(\mathbf{p}) > 0\}$, $\{\mathbf{p} \in \mathcal{D} \mid f_\theta(\mathbf{p}) < 0\}$ and $\{\mathbf{p} \in \mathcal{D} \mid f_\theta(\mathbf{p}) = 0\}$. The last of these sets is empty for generic choices of parameters. In this case, $f_\theta$ induces a *dichotomy* $C \in \{-, +\}^M$ via $C_i = f_\theta(\mathbf{p}_i)$.

We now discuss three different points of view on these dichotomies. We first state the equivalent viewpoints, and define the terms in this statement afterwards.

**Theorem 2.1.** *Let $\mathcal{D} \subset \mathbb{R}^d$ be a finite data set. Then*

    (i) *the hyperplane arrangement $\mathcal{H}_\mathcal{D} = \bigcup_{\mathbf{p} \in \mathcal{D}} (1, \mathbf{p})^\perp$ subdivides the parameter space $\Theta(d)$ into regions according to the represented dichotomies,*

    (ii) *$\mathcal{H}_D$ induces the normal fan of the zonotope $P_\mathcal{D} = \sum_{\mathbf{p} \in \mathcal{D}} \operatorname{conv}(\mathbf{0}, \left(\begin{smallmatrix} 1 \\ \mathbf{p} \end{smallmatrix}\right))$,*

    (iii) *the dichotomies are the maximal covectors of a realizable oriented matroid.*

For $\mathbf{p} \in \mathbb{R}^d$ we consider $(1, \mathbf{p})^\perp = \{(a, \mathbf{s}) \in \Theta(d) \mid \langle \left(\begin{smallmatrix} a \\ \mathbf{s} \end{smallmatrix}\right), \left(\begin{smallmatrix} 1 \\ \mathbf{p} \end{smallmatrix}\right) \rangle = 0\}$ as a hyperplane through the origin. A *linear hyperplane arrangement* $\mathcal{H} \subset \mathbb{R}^{d+1}$ is a collection of hyperplanes through the origin, and it subdivides the ambient space into *chambers*, which are the connected components of $\mathbb{R}^{d+1} \setminus \mathcal{H}$ and are open polyhedral cones.

We recall a few more definitions from polyhedral geometry. A *polyhedron* is the intersection of finitely many closed halfspaces and a *polytope* is a bounded polyhedron. Equivalently, a polytope $P \subseteq \mathbb{R}^d$ is the *convex hull* of finitely many points $v_1, \ldots, v_n \in \mathbb{R}^d$ (Ziegler, 1995, Theorem 1.1), i.e. $P = \operatorname{conv}(v_1, \ldots, v_n)$, where

$$\operatorname{conv}(v_1, \ldots, v_n) = \{\lambda_1 v_1 + \cdots + \lambda_n v_n \mid \lambda_i \in [0, 1], \ \lambda_1 + \cdots + \lambda_n = 1\}.$$

A hyperplane *supports* $P$ if it bounds a closed halfspace containing $P$, and any intersection of $P$ with such a supporting hyperplane yields a *face* $F$ of $P$. A face is a *proper face* if $F \subsetneq P$ and inclusion-maximal proper faces are referred to as *facets*. Note that also the empty set is a face of $P$ and by convention $\dim(\emptyset) = -1$.

The *Minkowski sum* of polytopes $P_1, \ldots, P_k \subseteq \mathbb{R}^d$ is

$$P_1 + \cdots + P_k = \{x_1 + \cdots + x_k \mid x_i \in P_i \text{ for } i \in [k]\},$$

and a *zonotope* is a Minkowski sum of line segments.

A *polyhedral cone* $C \subseteq \mathbb{R}^d$ is a polyhedron such that $\lambda u + \mu v \in C$ for every $u, v \in C$ and $\lambda, \mu \in \mathbb{R}_{\geq 0}$. Equivalently, it is the *conical hull* of finitely many vectors $u_1, \ldots, u_n \in \mathbb{R}^d$ (Ziegler, 1995, Theorem 1.3), i.e.

$$C = \operatorname{cone}(u_1, \ldots, u_n) = \{\mu_1 u_1 + \cdots + \mu_n r_n \mid \mu_1, \ldots, \mu_n \geq 0\}.$$

The *lineality space* of $C$ is the linear space $\mathcal{L}(C) = C \cap (-C)$ and a cone is *pointed* if its lineality space is trivial. The *rays* of $C$ are its 1-dimensional faces. A *polyhedral fan* $\Sigma \subseteq \mathbb{R}^d$ is a finite family of nonempty polyhedral cones such that every nonempty face of a cone in $\Sigma$ is also a cone in $\Sigma$, and the intersection of any two cones in $\Sigma$ is a face of both. An inclusion-maximal cone of $\Sigma$ is called a *maximal* cone. The fan $\Sigma$ is *complete* if $\bigcup_{C \in \Sigma} C = \mathbb{R}^d$ and it is *pure* if all inclusion-maximal cones have the same dimension.

Any hyperplane arrangement $\mathcal{H}$ uniquely induces a polyhedral fan $\Sigma$, whose maximal cones are the Euclidean closures of the chambers of $\mathcal{H}$. A *wall* of $\Sigma$ is an intersection of maximal cones that has codimension 1. In particular, if $\Sigma \subseteq \mathbb{R}^d$ is a complete (and thus pure) polyhedral fan, then the walls are the cones of dimension $d - 1$, and must always exists.

Given any polytope $P \subset \mathbb{R}^{d+1}$, the (outer) normal cone of a face $F$ of $P$ is

$$N_F(P) = \{y \in (\mathbb{R}^{d+1})^* \mid \langle z, y \rangle = \max_{x \in P} \langle x, y \rangle \text{ for all } z \in F\},$$

and the *normal fan* of $P$ is the collection of normal cones over all faces of $P$. A polytope is a zonotope if and only if its normal fan is induced by a hyperplane arrangement $\mathcal{H}$.

The hyperplane arrangement $\mathcal{H}_\mathcal{D} = \bigcup_{i=1}^{M}(1, \mathbf{p}_i)^\perp \subset \Theta(d)$ uniquely induces a polyhedral fan $\Sigma_\mathcal{D} \subset \Theta(d)$, which is the normal fan of the zonotope $P_\mathcal{D} = \sum_{\mathbf{p} \in \mathcal{D}} \operatorname{conv}((\mathbf{0}, \left( \begin{smallmatrix} 1 \\ \mathbf{p} \end{smallmatrix} \right)))$. Here, $\mathbf{0} = (0, \dots, 0)$ denotes the zero vector in $\mathbb{R}^{d+1}$. We can label each cone $\sigma$ of the fan $\Sigma_\mathcal{D}$ by a vector of signs,

$$C_\sigma = \left( \operatorname{sgn}(\langle \theta, \left( \begin{smallmatrix} 1 \\ \mathbf{p}_1 \end{smallmatrix} \right) \rangle), \dots, \operatorname{sgn}(\langle \theta, \left( \begin{smallmatrix} 1 \\ \mathbf{p}_M \end{smallmatrix} \right) \rangle) \right) = (f_\theta(\mathbf{p}_1), \dots, f_\theta(\mathbf{p}_M)) \in \{-, 0, +\}^M,$$

where we can choose any parameter vector $\theta$ contained in the relative interior of $\sigma$. We call $C_\sigma$ the *(signed) covector* of $\sigma$. The interiors of maximal cones of $\Sigma_\mathcal{D}$ consist of parameters which define a strict separation of the data, and are labeled with covectors $C \in \{-, +\}^M$ without zero entries (i.e. dichotomies). On the other hand, lower-dimensional cones are non-strict, i.e. there exist some data points $\mathbf{p}_i$ which lie on the separating hyperplane, and $C_i = 0$ for these data points. Duality between polytopes and normal fans implies that the vertices of $P_\mathcal{D}$ are in bijection with the dichotomies that the simple perceptron can compute.

**Example 2.2.** Consider the 1-dimensional dataset $\mathcal{D} = \{-2, -1, 0, 1, 2\} \subset \mathbb{R}^1$. The parameter space $\Theta(1) \cong \mathbb{R}^2$ contains the hyperplane arrangement $\mathcal{H}_\mathcal{D}$ consisting of five hyperplanes $H_{-2} = (1, -2)^\perp, H_{-1} = (1, -1)^\perp, H_0 = (1, 0)^\perp, H_1 = (1, 1)^\perp, H_2 = (1, 2)^\perp$, as depicted in Figure 1. The induced polyhedral fan $\Sigma_\mathcal{D}$ consists of 10 maximal (2-dimensional) cones. Any two neighboring maximal cones in this fan are separated by one of the 10 1-dimensional walls. The fan $\Sigma_\mathcal{D}$ is the normal fan of the polytope $P_\mathcal{D}$ with vertices

$$\left( \begin{smallmatrix} 0 \\ 0 \end{smallmatrix} \right), \left( \begin{smallmatrix} 1 \\ -2 \end{smallmatrix} \right), \left( \begin{smallmatrix} 2 \\ -3 \end{smallmatrix} \right), \left( \begin{smallmatrix} 3 \\ -3 \end{smallmatrix} \right), \left( \begin{smallmatrix} 4 \\ -2 \end{smallmatrix} \right), \left( \begin{smallmatrix} 5 \\ 0 \end{smallmatrix} \right), \left( \begin{smallmatrix} 4 \\ 2 \end{smallmatrix} \right), \left( \begin{smallmatrix} 3 \\ 3 \end{smallmatrix} \right), \left( \begin{smallmatrix} 2 \\ 3 \end{smallmatrix} \right), \left( \begin{smallmatrix} 1 \\ 2 \end{smallmatrix} \right).$$

This polytope can be written as the Minkowski sum of line segments

$$P_\mathcal{D} = \operatorname{conv}(\mathbf{0}, \left( \begin{smallmatrix} 1 \\ -2 \end{smallmatrix} \right)) + \operatorname{conv}(\mathbf{0}, \left( \begin{smallmatrix} 1 \\ -1 \end{smallmatrix} \right)) + \operatorname{conv}(\mathbf{0}, \left( \begin{smallmatrix} 1 \\ 0 \end{smallmatrix} \right)) + \operatorname{conv}(\mathbf{0}, \left( \begin{smallmatrix} 1 \\ 1 \end{smallmatrix} \right)) + \operatorname{conv}(\mathbf{0}, \left( \begin{smallmatrix} 1 \\ 2 \end{smallmatrix} \right))$$

and is hence a zonotope. The vertices of the zonotope are dual to the 2-dimensional cones of $\Sigma_\mathcal{D}$. As can be seen in Figure 1, each maximal cone of $\Sigma_\mathcal{D}$ is the normal cone of a vertex of $P_\mathcal{D}$. The figure also shows the covectors of these maximal cones $\sigma$, i.e., the dichotomies $(f_\theta(-2), f_\theta(-1), f_\theta(0), f_\theta(1), f_\theta(2))$ for $\theta \in \sigma$ in the interior of the cones. The value $f_\theta(p)$ expresses on which side of the hyperplane $H_p$ the corresponding cone lies. For example, let $C$ be the 1-dimensional cone that is the wall between the cone with covector $(-, -, +, +, +)$ and $(-, +, +, +, +)$. Then the covector of $C$ is $(-, 0, +, +, +)$, expressing that $C$ is contained in the hyperplane $H_{-1}$.

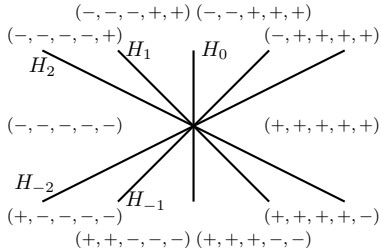

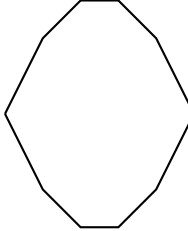

(a) The hyperplane arrangement $\mathcal{H}_\mathcal{D}$.        (b) The zonotope $P_\mathcal{D}$.

Figure 1: Illustration of Example 2.2. The left panel shows the parameter space and its subdivision by the hyperplane arrangement $\mathcal{H}_\mathcal{D}$ for a dataset $\mathcal{D}$ consisting of five points in $\mathbb{R}^1$. The right panel shows the polytope $P_\mathcal{D}$.

As all covectors in this article are signed covectors, we omit the word "signed" throughout this article. The *maximal covectors* (or *topes*) are the covectors with inclusion-maximal support, i.e. the dichotomies.

We will use the notation $[M] = \{1, \ldots, M\}$. For two covectors $C, D$, we define the *separation set* as $S(C, D) = \{i \in [M] \mid C_i = -D_i \neq 0\}$. The set of all covectors forms an oriented matroid. More formally, an *oriented matroid* is a pair $([M], \mathcal{C})$, where $\mathcal{C}$ is a collection of elements of $\{-, 0, +\}^{[M]}$ (called [signed] covectors) satisfying the following axioms (Goodman et al., 2018, Section 6.2.1):

(C I) (Zero) $(0, \ldots, 0) \in \mathcal{C}$;

(C II) (Symmetry) $C \in \mathcal{C} \implies -C \in \mathcal{C}$;

(C III) (Composition) if $C, D \in \mathcal{C}$ then $(C \circ D) \in \mathcal{C}$, where $(C \circ D)_i = \begin{cases} C_i & \text{if } C_i \neq 0, \\ D_i & \text{otherwise}; \end{cases}$

(C IV) (Elimination) if $C, D \in \mathcal{C}$ and $i \in S(C, D)$ then there exists some $Z \in \mathcal{C}$ such that $Z_i = 0$ and $Z_j = (C \circ D)_j \ \forall j \in [M] \setminus S(C, D)$.

Any collection of covectors that arises through a hyperplane arrangement is called a *realizable oriented matroid*. To each oriented matroid, one can associate a dual oriented matroid; for realizable oriented matroids this captures the geometry of the vector configuration given by the normal vectors of the hyperplanes, i.e. the dataset $\mathcal{D}$.

In a classification task, we are typically interested in a strict separation of the data, represented by a dichotomy. We now fix a target dichotomy $C^* \in \{-, +\}^M$, which divides the data into two sets $\mathcal{D}_+^{C^*} = \{\mathbf{p}_i \in \mathcal{D} \mid C_i^* = +\}, \mathcal{D}_-^{C^*} = \mathcal{D} \setminus \mathcal{D}_+^{C^*}$. The target dichotomy $C^*$ is a covector in the oriented matroid if and only if there exists a hyperplane separating the data into $\mathcal{D}_+^{C^*}, \mathcal{D}_-^{C^*}$, i.e. the data is *linearly separable* according to $C^*$. Equivalently, by Farkas' Lemma, the covector exists if and only if $\operatorname{conv}(\mathcal{D}_+^{C^*}) \cap \operatorname{conv}(\mathcal{D}_-^{C^*}) = \emptyset$. For any parameter vector $\theta$, the *0/1-loss-function* $\operatorname{err}_{C^*}$ counts the number of mistakes, i.e.

$$\operatorname{err}_{C^*}(\theta) = |\{i \in [M] \mid \operatorname{sgn}(f_\theta(\mathbf{p}_i)) = -C_i^*\}|.$$

Since the 0/1-loss function is constant along chambers of $\mathcal{H}_\mathcal{D}$ and (relative interiors of) cones of $\Sigma_\mathcal{D}$, we allow ourselves to write $\operatorname{err}_{C^*}(\sigma)$ for chambers or cones $\sigma \in \Sigma_\mathcal{D}$. In the notation of separation sets given above, if $D$ is the covector associated to $\sigma$ then $\operatorname{err}_{C^*}(\sigma) = |S(C^*, D)|$. Note that we choose to interpret data points which lie on the classifying hyperplane to be correctly classified. This technical distinction is irrelevant on interiors of maximal cones and allows us to consider polyhedral fans in Section 6.

The $k^{th}$ *level set* is the polyhedral subfan $\Sigma_\mathcal{D}^k = \{\sigma \in \Sigma \mid \operatorname{err}_{C^*}(\sigma) = k\}$, and the *sublevel set* is $\Sigma_\mathcal{D}^{\leq k} = \bigcup_{l=0}^k \Sigma_\mathcal{D}^k$. If the data is linearly separable according to $C^*$, then the set $\Sigma_\mathcal{D}^0$ of parameters with 0 error is a maximal cone of $\Sigma_\mathcal{D}$. On the other hand, if the data is not linearly separable, then the minimum error on the interior of any maximal cone will be larger than 0 and multiple maximal cones may be minima of the 0/1-loss.

**Example 2.3.** We continue with Example 2.2. Consider a target dichotomy $C_1^* = (+, -, +, -, +)$. This asks for a separation of the input data points $\{-2, -1, 0, 1, 2\}$ by a hyperplane in $\mathbb{R}^1$ such that $\{-2, 0, 2\}$ lie on the positive side of the hyperplane, and $\{-1, 1\}$ on the negative side. Similarly, $C_2^* = (+, -, -, +, +)$ asks for a separation into $\{-2, 1, 2\}$ and $\{-1, 0\}$. Clearly the data is not linearly separable in accordance with either of these target dichotomies. Figure 2 shows the value of $\operatorname{err}_{C^*}$ on each cone of $\mathcal{H}_\mathcal{D}$ for both target dichotomies. Notably, for $C_1^*$ the set of minimizers of $\operatorname{err}_C^*$ consists of 5 maximal cones, which pairwise intersect only in the origin, and are thus not connected through walls. On the other hand, for $C_2^*$ the set of minimizers consists of a single cone and is a convex set.

In the remainder of this section, we study the connectivity of sublevel sets for linear classifiers. If one seeks to find a set of parameters $\theta \in \Theta(d)$ which separates the data, then in practice this can be achieved by minimizing a suitable loss function in the parameter space $\Theta(d)$ using an iterative optimization procedure. In this case the search variable is likely to move from chamber to chamber with transitions going through walls of codimension 1. We would like to understand under which circumstances there exists a path along which the loss function is monotonically decreasing. The following statement is an adaptation of Björner et al. (1999, Proposition 4.2.3) and we give a proof for completeness. Recall that for covectors $C, D$, the separation set is $S(C, D) = \{i \in [M] \mid C_i = -D_i \neq 0\}$.

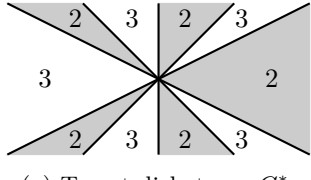
(a) Target dichotomy $C_1^*$.

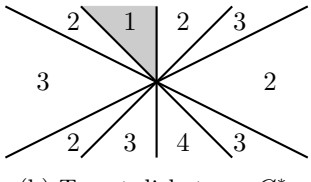
(b) Target dichotomy $C_2^*$.

Figure 2: Illustration of Example 2.3. For a given set of input data points the figure shows the polyhedral fan $\Sigma_{\mathcal{D}}$ in parameter space along with the value of the 0/1-loss for the two target dichotomies $C_1^*$ (left) and $C_2^*$ (right). The sets of minimizers are shown in gray.

**Proposition 2.4.** *Let $C, D$ be two maximal covectors corresponding to maximal cones of $\Sigma_{\mathcal{D}}$. Then there exists a sequence of maximal covectors $D = D^0, D^1, \ldots, D^k = C$ such that the cones corresponding to $D^i, D^{i+1}$ are connected through a wall of codimension 1 for $i = 0, \ldots, k-1$ and $S(C, D^i) > S(C, D^{i+1})$.*

*Proof.* First note that the maximal cones are labelled by covectors without any zero entry. For two such covectors, we have $D \circ C = D$, and for any $i \in S(C, D)$, axiom (C IV) translates to the existence of a covector $Z$ such that $Z_i = 0$ and $Z_j = C_j$ for all $j \in [M] \setminus S(C, D)$. We prove the statement by induction on the size of $S(C, D)$. Since $C, D$ can be assumed not to have zero entries, this is the number of entries in which $C$ and $D$ differ. Suppose $S(C, D) = \{i\}$. By (C IV) there exists a covector $Z$ such that $Z_i = 0$ and $Z_j = C_j = D_j$ for all $j \in [M] \setminus \{i\}$. Thus, the chambers corresponding to $C, D$ are separated by the hyperplane $(1, \mathbf{p}_i)^\perp$ and $Z$ corresponds to the wall $C \cap D$.

Suppose now the statement holds for any $D', D''$ such that $|S(D', D'')| < k$ and let $S(C, D) = k$. Fix $i \in S(C, D)$. By (C IV) there exists some covector $Z$ such that $Z_i = 0$ and $Z_j = C_j = D_j$ for all $j \in [M] \setminus S(C, D)$. Let $A = \{i \in [M] \mid Z_i = 0\}$ and $l = |A| \geq 1$. Among all such covectors, choose $Z$ such that $l$ is minimal. Thus, there exists no strict subset $A' \subsetneq A$ and covector $Z'$ such that $Z'_i = 0$, $Z'_j = C_j = D_j$ for all $[M] \setminus S(C, D)$ and $Z'_j \neq 0$ for all $j \in [M] \setminus A'$. Recall that $Z'$ corresponds to a cone of a $\Sigma_{\mathcal{D}}$, which is contained in the hyperplane $(1, \mathbf{p}_j)^\perp$ if and only if $Z'_j = 0$. Since $Z$ exists, but no such $Z'$ exists, this implies that the hyperplanes $(1, \mathbf{p}_j), j \in A$ all coincide. In other words, if $Z$ is chosen minimally, then $\mathbf{p}_j = \mathbf{p}_{j'}$ for all $j, j' \in A$, and hence $Z$ represents a wall of $\Sigma_{\mathcal{D}}$. By (C III) we have that also $(Z \circ C)$ and $(Z \circ D)$ are covectors. Note that $(Z \circ C)_j = Z_j = (Z \circ D)_j \neq 0$ for all $j \in [M] \setminus A$, and $\{(Z \circ C)_i, Z_i, (Z \circ D)_i\} = \{+, 0, -\}$ for all $i \in A$. Thus, $(Z \circ C), (Z \circ D)$ correspond to adjacent maximal chambers separated by $(1, \mathbf{p}_i), i \in A$, and the separating wall corresponds to $Z$. Since $S(C, Z \circ C) < k$ and $S(Z \circ D, D) < k$, there exist strictly decreasing sequences from $C$ to $(Z \circ C)$ and from $Z \circ D$ to $D$ by induction, which together form a sequence from $C$ to $D$. $\square$

Given a sequence $\sigma_1, \ldots, \sigma_k \in \Sigma_{\mathcal{D}}$ of maximal cones, we say that the sequence *forms a path* in $\Sigma_{\mathcal{D}}$ connecting $\sigma_1$ and $\sigma_k$ if $\sigma_i, \sigma_{i+1}$ intersect in a wall of codimension 1 for all $i \in [k-1]$. Proposition 2.4 allows us to characterize the set of local and global minima, and bound the error along paths between two minima.

**Theorem 2.5.** *If the data is linearly separable, then the sublevel sets $\Sigma_{\mathcal{D}}^{\leq k}$ of the 0/1-loss function are connected through walls of codimension 1 for any $k \geq 0$. Conversely, if the data is not linearly separable and the set of minimizers of the 0/1-loss function consists of more than one maximal cone, then it is not connected through codimension 1. Moreover, if $\sigma, \sigma'$ are distinct maximal cones which are local minima of the 0/1-loss, if $m = \mathrm{err}_{C^*}(\sigma)$ denotes the minimum error and if $\sigma, \sigma_1, \ldots, \sigma_l, \sigma'$ forms a path, then $\mathrm{err}_{C^*}$ is bounded from above along this path by $m + \lfloor \frac{l+1}{2} \rfloor$.*

The first two statements of this theorem are well-established in the literature; the (sub-) level sets of the 0/1-loss function are dual to $k$-facets heavily studied in computational geometry (Wagner, 2008, Section 1.2), and to linear programs with violated constraints (Matoušek, 1995). In Section 6 we will make related statements for the case of continuous piecewise linear classifiers. Therefore, we give a proof for completeness also for this theorem. Before we prove the statement, we point out that the assertion of the second sentence is not void, and truly necessary, as can be seen from the two target dichotomies given in Example 2.3.

*Proof.* Suppose the data $\mathcal{D}$ is linearly separable. Then the set which minimizes $\text{err}_{C^*}$ is $\Sigma^0_{\mathcal{D}} = \{C^*\}$ and consists of a single chamber. In particular, $\Sigma^0_{\mathcal{D}}$ is connected. For any $k > 0$ and $D \in \Sigma^k_{\mathcal{D}}$, we have that $\text{err}_{C^*}(D) = S(C^*, D) = k$, and Proposition 2.4 implies that there is a decreasing path from $D$ to $C^*$ through walls of codimension 1. Thus, the sublevel sets are connected through walls. For the second statement, suppose that $\mathcal{D}$ is not separable and the set of minimizers contains at least two chambers. Now, assume for contradiction that it is connected through a wall of codimension 1. That means there are covectors $C, D$ with minimal $|S(C^*, C)| = |S(C^*, D)|$ and an index $i$ such that the corresponding chambers are separated by the $i$th hyperplane $(1, \mathbf{p}_i)^\perp$. In other words, $C_i = -D_i$ and $C_j = D_j$ for all $j \in [M], j \neq i$. The latter implies $|S(C^*, C)| \neq |S(C^*, D)|$, a contradiction. For the last statement, let $\sigma = \sigma_0, \sigma_1, \ldots, \sigma_l, \sigma' = \sigma_{l+1}$ be a path in $\Sigma_{\mathcal{D}}$, and let $D_i$ be the covector corresponding to $\sigma_i$. Then $\text{err}^*_C(\sigma_i) = S(C, D_i)$. However, since $\sigma_i, \sigma_{i+1}$ intersect in codimension 1, we have that $S(D_i, D_{i+1}) = 1$ and so $\text{err}^*_C(\sigma_{i+1}) \leq \text{err}\, C^*(\sigma_i) + 1$, i.e. the error increases in each cone at most by one. Since $\text{err}^*_C(\sigma_0) = \text{err}^*_C(\sigma_{l+1})$ this implies that the increase of the error along the path is bounded in terms of the length of the path by $\lfloor \frac{l+1}{2} \rfloor$, yielding a total bound of $m + \lfloor \frac{l+1}{2} \rfloor$ on the path. $\qquad\square$

The bound of $m + \lfloor \frac{l+1}{2} \rfloor$ can indeed be attained even as minimum among all paths between $\sigma$ and $\sigma'$. One example is the 1-dimensional data with coordinates $1, 2, \ldots, 4k$ for some $k \in \mathbb{N}$ and target dichotomy $(\underbrace{+, \ldots, +}_{k \text{ times}}, \underbrace{-, \ldots, -}_{2k \text{ times}}, \underbrace{+, \ldots, +}_{k \text{ times}})$. The covectors minimizing the 0/1-loss are

$$(\underbrace{+, \ldots, +}_{k \text{ times}}, \underbrace{-, \ldots, -}_{3k \text{ times}},) \quad \text{and} \quad (\underbrace{-, \ldots, -}_{3k \text{ times}}, \underbrace{+, \ldots, +}_{k \text{ times}}),$$

each making $m = k$ mistakes, and are connected by two distinct paths, each of length $l + 1 = 2k + 1$, so $m + \lfloor \frac{l+1}{2} \rfloor = 2k$. One of the paths contains the covector $(+, \ldots, +)$, and the other contains $(-, \ldots, -)$, each of them making $2k$ mistakes and thus attaining the bound.

## 3 Piecewise Linear Classifiers, ReLU Networks, and Tropical Rational Functions

In this section we extend the theoretical framework from linear classifiers to the more general case of classification with continuous piecewise linear functions. An important instance of this are neural networks with piecewise linear activation functions.

Recall that an $L$-layer feedforward neural network represents a function $f : \mathbb{R}^{d_{\text{in}}} \to \mathbb{R}^{d_{\text{out}}}$ which is obtained as a composition $f = \sigma^{(L)} \circ \varphi^{(L)} \circ \cdots \circ \sigma^{(1)} \circ \varphi^{(1)}$. For each $\ell \in [L]$ the *preactivation function* $\varphi^{(\ell)} : \mathbb{R}^{d_{\ell-1}} \to \mathbb{R}^{d_\ell}$ is an affine function $\varphi^{(\ell)}(x) = W^{(\ell)}x + c^{(\ell)}$ with *weights* $W^{(\ell)} \in \mathbb{R}^{d_\ell \times d_{\ell-1}}$ and *biases* $c^{(\ell)} \in \mathbb{R}^{(d_\ell)}$ of the $\ell^{th}$ *layer*, and $d_{\text{in}} = d_0, d_{\text{out}} = d_L$. A ReLU neural network is a neural network with *ReLU activation function (Rectified Linear Unit)* $\sigma^{(\ell)}(x_1, \ldots, x_{d_\ell}) = (\max(x_1, 0), \ldots, \max(x_{d_\ell}, 0))$. We denote $f^{(\ell)} : \mathbb{R}^{d_{\text{in}}} \to \mathbb{R}^{d_\ell}$, $f^{(\ell)} = \sigma^{(\ell)} \circ \varphi^{(\ell)} \circ \cdots \circ \sigma^{(1)} \circ \varphi^{(1)}$ and $f^{(0)}(x) = x$. The number $L$ of such compositions is the *depth* of the network, and the dimension $d_\ell, \ell \in [L]$ is the *width* of the $\ell^{\text{th}}$ layer. We do not impose any further restrictions on the weights and biases and so the choices of the depth and width of each layer determine the architecture of the network. The network consists of all of its activation and preactivation functions.

Any function represented by a ReLU neural network is a continuous piecewise-linear function. A powerful framework to study piecewise-linear functions is provided by the language of tropical geometry. This is the geometry over the tropical semiring (or max-plus algebra) where we define tropical addition, multiplication, exponentiation and division as

$$a \oplus b = \max(a, b), \ a \odot b = a + b, \ a^{\odot b} = a \cdot b, \ a \oslash b = a - b \ \text{ for } a, b \in \mathbb{R}.$$

For vectors $\mathbf{x}, \mathbf{s} \in \mathbb{R}^d$ we will write $\mathbf{x}^{\odot \mathbf{s}} = x_1^{\odot s_1} \odot \cdots \odot x_d^{\odot s_d} = \langle \mathbf{x}, \mathbf{s} \rangle$. A *tropical Laurent polynomial* is a function $g : \mathbb{R}^d \to \mathbb{R}$ of the form

$$g(\mathbf{x}) = \bigoplus_{i \in [n]} a_i \odot \mathbf{x}^{\odot \mathbf{s}_i} = \max_{i \in [n]} (a_i + \langle \mathbf{s}_i, \mathbf{x} \rangle),$$

where $\mathbf{x} = (x_1, \ldots, x_d)$ is the input variable and $a_i \in \mathbb{R}$ and $\mathbf{s}_i \in \mathbb{Z}^d, i \in [n]$, are parameters. A *tropical signomial* follows the same definition, except that we allow $\mathbf{s}_i \in \mathbb{R}^d, i \in [n]$. For the purposes of this article, this distinction makes no difference, so we use these terms interchangeably. In the same spirit, we define a *tropical rational function* as a function

$$f(\mathbf{x}) = (g \oslash h)(\mathbf{x}) = \bigoplus_{i \in [n]} a_i \odot \mathbf{x}^{\odot \mathbf{s}_i} \oslash \bigoplus_{j \in [m]} b_j \odot \mathbf{x}^{\odot \mathbf{t}_j} = \max_{i \in [n]} \left( a_i + \langle \mathbf{s}_i, \mathbf{x} \rangle \right) - \max_{j \in [m]} \left( b_j + \langle \mathbf{t}_j, \mathbf{x} \rangle \right),$$

which is determined by its parameter vector $\theta = (a_1, \mathbf{s}_1 \ldots, a_n, \mathbf{s}_n, b_1, \mathbf{t}_1, \ldots, b_m, \mathbf{t}_m)$, where $a_i \in \mathbb{R}, \mathbf{s}_i \in \mathbb{R}^d$ for $i \in [n]$ and $b_j \in \mathbb{R}, \mathbf{t}_j \in \mathbb{R}^d$ for $j \in [m]$. We denote by $\Theta(d, n, m)$ the $(n+m)(d+1)$-dimensional space of parameters of tropical rational functions with $n$ terms in the numerator and $m$ terms in the denominator, and for a fixed $\theta \in \Theta(d, n, m)$ we denote by $g_\theta \oslash h_\theta$ the associated tropical rational function. In Section 6 we show that $\Theta(d, n, m)$ allows a natural polyhedral fan structure in analogy to the fan and hyperplane arrangement from Section 2.

A tropical signomial is a convex and continuous piecewise linear function, and vice versa. Every continuous (possibly non-convex) piecewise linear function can be written as the difference of two convex piecewise linear functions, and this difference is a tropical rational function. This establishes the connection to ReLU neural networks.

**Theorem 3.1** (Arora et al. 2018, Theorem 2.1 and Zhang et al. 2018, Theorem 5.4). *A function $f\colon \mathbb{R}^d \to \mathbb{R}$ is a tropical rational function if and only if $f$ can be represented by a feedforward ReLU network. Any tropical rational function $g \oslash h$ with $n$ terms in the numerator and $m$ terms in the denominator can be represented by a ReLU network with depth at most $\min(\lceil \log_2(d+1) \rceil + 1, \max(\lceil \log_2(n) \rceil, \lceil \log_2(m) \rceil) + 2)$.*

A similar result has also appeared in work of Siahkamari et al. (2020). The original formulation of Zhang et al. (2018, Theorem 5.4) has an additional condition on the weights to be integer, however this is merely an artifact of the distinction between tropical Laurent polynomials and signomials. Given the class of tropical rational functions with a bounded number of terms in the numerator and denominator, this gives a sufficient condition on the depth of the architecture.

We may also consider the reverse direction: Let $\mathbf{ReLU}(d_0, d_1, \ldots, d_{L-1}, d_L)$ be the set of piecewise linear functions that can be represented by a fully-connected ReLU network with $d_0 = d$ inputs and $L$ layers of sizes $d_1, \ldots, d_L \in \mathbb{N}$, with $d_L = 1$. Given a fixed such architecture, we seek to find $n, m \in \mathbb{N}$ such that for any function $f \in \mathbf{ReLU}(d, d_1, \ldots, d_{L-1}, 1)$ there exists a parameter $\theta \in \Theta(d, n, m)$ such that $f = f_\theta$. For lower bounds on $n, m$, let $k_{\mathrm{convex}}$ be the maximum number of linear pieces over all convex functions in $\mathbf{ReLU}(d, d_1, \ldots, d_{L-1}, 1)$, and $k_{\mathrm{concave}}$ the maximum number of linear pieces of any concave function. Then necessarily we have $n \geq k_{\mathrm{convex}}$ and $m \geq k_{\mathrm{concave}}$. To obtain upper bounds, we consider a simple decomposition of the functions represented by a ReLU network as differences of convex piecewise linear functions, whose size depends on $k_{\mathrm{convex}}$ and $k_{\mathrm{concave}}$ in each step. For most of the functions in $\mathbf{ReLU}(d, d_1, \ldots, d_{L-1}, 1)$ the decomposition that we apply is by no means minimal for the individual function. However, this decomposition allows us to consider the space of ReLU networks as semialgebraic sets inside $\Theta(d, n, m)$. We will discuss minimal decompositions at the end of this section.

**Theorem 3.2.** *Let $d = d_0, d_1, \ldots, d_{L-1}, d_L = 1 \in \mathbb{N}$. There exist $n, m \in \mathbb{N}$ such that each function in $\mathbf{ReLU}(d, d_1, \ldots, d_{L-1}, 1)$ can be represented by a point in $\Theta(d, n, m)$, and there is a semialgebraic subset of $\Theta(d, n, m)$ (described by polynomial inequalities) representing exactly the points in $\mathbf{ReLU}(d, d_1, \ldots, d_{L-1}, 1)$. This semialgebraic set can be described by polynomial inequalities of degree $\leq L + 1$. The $n, m$ can be chosen as $n = 2m$ and $\log_2(m) \leq \sum_{k=1}^{L-1} 2^{L-1-k} \prod_{l=k}^{L-1} d_l$.*

We make the statement of this theorem more precise. Recall that any vector of parameters $\theta \in \Theta(d, n, m)$ defines a tropical rational function $g_\theta \oslash h_\theta$ with at most $n$ monomials in the numerator and at most $m$ monomials in the denominator. Let $\mathbf{CPWL}(\mathbb{R}^d, \mathbb{R})$ be the space of all continuous piecewise linear functions from $\mathbb{R}^d$ to $\mathbb{R}$, and let $\psi\colon \Theta(d, n, m) \to \mathbf{CPWL}(\mathbb{R}^d, \mathbb{R})$ with $\psi(\theta) = g_\theta \oslash h_\theta$. Then Theorem 3.2 says that for any $d_1, \ldots, d_{L-1} \in \mathbb{N}$ there exist values $n, m \in \mathbb{N}$ such that $\mathbf{ReLU}(d, d_1, \ldots, d_{L-1}, 1) \subseteq \psi(\Theta(d, n, m))$, and there exists a semialgebraic set $\mathcal{S} \subseteq \Theta(d, n, m)$ such that $\psi(\mathcal{S}) = \mathbf{ReLU}(d, d_1, \ldots, d_{L-1}, 1)$.

*Proof.* We show this by induction on $L$. For $L = 1$, let $W \in \mathbb{R}^{1 \times d}, c \in \mathbb{R}$ denote the weights and biases. Then the network represents the function $f^{(1)} = \max(Wx + c, 0)$. This is already a convex piecewise linear function (so $\max(Wx + c, 0) - 0$ is a representation as a difference with $n = 2, m = 1$). We still perform a transformation to illustrate the procedure that we will use for $L > 1$ in the induction step. We write $W = W_+ - W_-$, where $W_+, W_- \in \mathbb{R}^{1 \times d}_{\geq 0}$. Then one representation of $f^{(1)} = \max(Wx + c, 0)$ as difference of convex functions is

$$g_\theta - h_\theta = \max(W_+ x + c, W_- x) - W_- x$$

for the parameter vector $\theta = (c, W_+, 0, W_-, 0, W_-) \in \psi^{-1}(f^{(1)}) \subseteq \Theta(d, 2, 1)$. Since $W_+, W_-$ are nonnegative vectors with disjoint support, we have that $\theta$ is contained in the semialgebraic set

$$\mathcal{S} = \{(a_1, \mathbf{s}_1, a_2, \mathbf{s}_2, b_1, \mathbf{t}_1) \in \mathbb{R}^{3(d+1)} \mid a_2 = b_1 = 0, \mathbf{s}_2 = \mathbf{t}_2 \leq 0, \mathbf{s}_1 \geq 0, \mathbf{s}_{1i}\mathbf{s}_{2i} = 0 \ \forall i \in [d]\},$$

which is defined by $d + 2$ linear equations, $2d$ linear inequalities and $d$ quadratic equations.

Conversely, for any $\theta \in \mathcal{S}$, the function $\psi(\theta)$ is represented by a ReLU network as follows. If $\theta = (a_1, \mathbf{s}_1, 0, \mathbf{s}_2, 0, \mathbf{t}_1)$, where $\mathbf{s}_1, \mathbf{s}_2$ are nonnegative and have disjoint support, then defining $W = \mathbf{s}_1 - \mathbf{s}_2 \in \mathbb{R}^{1 \times d}$ gives $\psi(\theta) = \max(Wx + a_1, 0)$, which is represented by the network with layers of sizes $d_0 = d, d_1 = 1$, weights $W$ and biases $a_1$.

We now proceed by induction $L \to L + 1$, and consider a ReLU network with $L + 1$ layers. The idea of the induction is as follows. Let $W$ and $c$ be the weights and biases of the last layer. Then $W = W^+ - W^-$ has a canonical decomposition into nonnegative matrices, and the entire function is of the form $f^{(L+1)} = \max(\sum_{k=1}^{d_L} W_k f_k + c, 0)$, where $f_1 \ldots, f_k$ are the tropical rational functions in the $L^{\text{th}}$ layer. Writing $f_k = g_k - h_k$ as a difference of two convex functions, we obtain the decomposition $f^{(L+1)} = \max(\sum_{k=1}^{d_L} (W_k^+ - W_k^-)(g_k - h_k) + c, 0) = \max(\sum_{k=1}^{d_L} W_k^+ g_k + W_k^- h_k + c, W_k^- g_k + W_k^+ h_k) - (W_k^- g_k + W_k^+ h_k)$. We now perform this computation in more detail.

The first $L$ layers represent a collection of $d_L$ tropical rational functions, i.e. $f^{(L)} : \mathbb{R}^{d_0} \to \mathbb{R}^{d_L}$,

$$f^{(L)}(\mathbf{x}) = \begin{pmatrix} g_1^{(L)}(\mathbf{x}) \oslash h_1^{(L)}(\mathbf{x}) \\ \vdots \\ g_{d_L}^{(L)}(\mathbf{x}) \oslash h_{d_L}^{(L)}(\mathbf{x}) \end{pmatrix},$$

$$g_k^{(L)}(\mathbf{x}) = \bigoplus_{i \in [n_{k,L}]} a_i^{k,L} \odot \mathbf{x}^{\mathbf{s}_i^{k,L}}, \quad h_k^{(L)}(\mathbf{x}) = \bigoplus_{j \in [m_{k,L}]} b_j^{k,L} \odot \mathbf{x}^{\mathbf{t}_j^{k,L}},$$

where $n_{k,L}, m_{k,L} \in \mathbb{N}, k \in [d_L]$ and $a_i^{k,L}, b_j^{k,L} \in \mathbb{R}, \mathbf{s}_i^{k,L}, \mathbf{t}_j^{k,L} \in \mathbb{R}^{d_0}$. Again, let $W = W^+ - W^-$ with $W_+, W_- \in \mathbb{R}^{1 \times d_L}_{\geq 0}, c \in \mathbb{R}$ be the weights and biases of the last layer.

Let

$$Y_{\text{convex}} = \sum_{k=1}^{d_L} (W_k^+ g_k^{(L)}(\mathbf{x}) + W_k^- h_k^{(L)}(\mathbf{x})),$$

$$Y_{\text{concave}} = \sum_{k=1}^{d_L} (W_k^- g_k^{(L)}(\mathbf{x}) + W_k^+ h_k^{(L)}(\mathbf{x})).$$

Then $f^{(L+1)}(\mathbf{x}) = \max(Y_{\text{convex}} - Y_{\text{concave}} + c, 0) = \max(Y_{\text{convex}} + c, Y_{\text{concave}}) - Y_{\text{concave}}$. We now compute $Y_{\text{convex}}$. $Y_{\text{concave}}$ can be computed analogously. For any nonnegative number $W_k$ holds $W_k \max(a, b) =$

$\max(W_k a, W_k b)$, and furthermore $\max(a, b) + \max(c, d) = \max(a + c, a + d, b + c, b + d)$. Thus,

$$
\begin{aligned}
Y_{\text{convex}} &= \sum_{k=1}^{d_L}(W_k^+ g_k^{(L)}(\mathbf{x}) + W_k^- h_k^{(L)}(\mathbf{x})) \\
&= \sum_{k=1}^{d_L} W_k^+ \max_{i \in [n_{k,L}]} \left( a_i^{k,L} + \left\langle \mathbf{s}_i^{k,L}, \mathbf{x} \right\rangle \right) + W_k^- \max_{j \in [m_{k,L}]} \left( b_j^{k,L} + \left\langle \mathbf{t}_j^{k,L}, \mathbf{x} \right\rangle \right) \\
&= \sum_{k=1}^{d_L} \max_{i \in [n_{k,L}]} \left( W_k^+ \left( a_i^{k,L} + \left\langle \mathbf{s}_i^{k,L}, \mathbf{x} \right\rangle \right) \right) + \max_{j \in [m_{k,L}]} \left( W_k^- \left( b_j^{k,L} + \left\langle \mathbf{t}_j^{k,L}, \mathbf{x} \right\rangle \right) \right) \\
&= \sum_{k=1}^{d_L} \max_{\substack{i \in [n_{k,L}] \\ j \in [m_{k,L}]}} \left( W_k^+ \left( a_i^{k,L} + \left\langle \mathbf{s}_i^{k,L}, \mathbf{x} \right\rangle \right) + W_k^- \left( b_j^{k,L} + \left\langle \mathbf{t}_j^{k,L}, \mathbf{x} \right\rangle \right) \right).
\end{aligned}
$$

In the following, we use the notation

$$
\max_{\substack{k \in [d_L] \\ i_k \in [n_{k,L}] \\ j_k \in [m_{k,L}]}}
$$

to denote the maximum over all $i_1 \in [n_{1,L}], \dots, i_{d_L} \in [n_{d_L,L}], j_1 \in [m_{1,L}], \dots, j_{d_L} \in [m_{d_L,L}]$. Applying again distributivity of tropical multiplication (i.e. that $\max(a, b) + \max(c, d) = \max(a + c, a + d, b + c, b + d)$), and bilinearity of $\langle \cdot, \cdot \rangle$ yields

$$
\begin{aligned}
&= \max_{\substack{k \in [d_L] \\ i_k \in [n_{k,L}] \\ j_k \in [m_{k,L}]}} \left( \sum_{k=1}^{d} W_k^+ \left( a_{i_k}^{k,L} + \left\langle \mathbf{s}_{i_k}^{k,L}, \mathbf{x} \right\rangle \right) + W_k^- \left( b_{j_k}^{k,L} + \left\langle \mathbf{t}_{j_k}^{k,L}, \mathbf{x} \right\rangle \right) \right) \\
&= \max_{\substack{k \in [d_L] \\ i_k \in [n_{k,L}] \\ j_k \in [m_{k,L}]}} \left( \left\langle \sum_{k=1}^{d_L} W_k^+ \mathbf{s}_{i_k}^{k,L} + W_k^- \mathbf{t}_{j_k}^{k,L}, \mathbf{x} \right\rangle + \sum_{k=1}^{d_L} (W_k^+ a_{i_k}^{k,L} + W_k^- b_{j_k}^{k,L}) \right),
\end{aligned} \tag{1}
$$

and this expression consists of $\prod_{k=1}^{d_L} n_{k,L} \prod_{k=1}^{d_L} m_{k,L}$ linear terms. A similar reasoning applies to $Y_{\text{concave}}$. Recall that $W$ is a $(1 \times d_L)$-matrix, and thus $(W_k^+ a_{i_k}^{k,L} + W_k^- b_{j_k}^{k,L}) \in \mathbb{R}$.

In the following, we use the shorthand notations

$$
a_{k,i_k,j_k}^{(L+1)} = a_{i_1,\dots,i_{d_L},j_1,\dots,j_{d_L}}^{(L+1)},
$$
$$
a_{k,n_{k,L}+i_k,m_{k,L}+j_k}^{(L+1)} = a_{n_{1,L}+i_1,\dots,n_{d_L,L}+i_{d_L},m_{1,L}+j_1,\dots,m_{d_L,L}+j_{d_L}}^{(L+1)},
$$

where $i_k \in [n_{k,d_L}], j_k \in [m_{k,d_L}]$ for all $k \in [d_L]$. With this, we have expressed $f^{(L+1)}$ as a quotient of two tropical polynomials:

$$
\begin{aligned}
f^{(L+1)}(\mathbf{x}) = &\max_{\substack{k \in [d_L] \\ i_k \in [n_{k,L}] \\ j_k \in [m_{k,L}]}} \left( a_{k,i_k,j_k}^{(L+1)} + \left\langle \mathbf{s}_{k,i_k,j_k}^{(L+1)}, \mathbf{x} \right\rangle, a_{k,n_{k,L}+i_k,m_{k,L}+j_k}^{(L+1)} + \left\langle \mathbf{s}_{k,n_{k,L}+i_k,m_{k,L}+j_k}^{(L+1)}, \mathbf{x} \right\rangle \right) - \\
&\max_{\substack{k \in [d_L] \\ i_k \in [n_{k,L}] \\ j_k \in [m_{k,L}]}} \left( b_{k,i_k,j_k}^{(L+1)} + \left\langle \mathbf{t}_{k,i_k,j_k}^{(L+1)}, \mathbf{x} \right\rangle \right)
\end{aligned}
$$

for some parameters $a_{k,i_k,j_k}^{(L+1)}, b_{k,i_k,j_k}^{(L+1)} \in \mathbb{R}$ and $\mathbf{s}_{k,i_k,j_k}^{(L+1)}, \mathbf{t}_{k,i_k,j_k}^{(L+1)} \in \mathbb{R}^d$ where $i_k \in [n_{k,L}], j_k \in [m_{k,L}]$ for any $k \in [d_L]$. In other words, we have proven that

$$
f^{(L+1)}(x) \in \psi(\Theta(d, 2 \prod_{k=1}^{d_L} n_{k,L} \prod_{k=1}^{d_L} m_{k,L}, \prod_{k=1}^{d_L} n_{k,L} \prod_{k=1}^{d_L} m_{k,L})).
$$

We first discuss the semialgebraic constraints on these parameters. By induction, the parameters of the tropical rational function representation $f_k^{(L)} = g_k^{(L)} - h_k^{(L)}, k \in [d_L]$ are contained in a semialgebraic set $\mathcal{S}^{(L)}$ which can be described by polynomial inequalities of degree at most $L + 1$. The above computation (1) imposes linear relations between the parameters of $f^{(L+1)}$ in terms of the parameters of $f_k^{(L)}, k \in [d_L]$. More specifically,

$$
\begin{aligned}
\mathbf{s}_{k,i_k,j_k}^{(L+1)} &= \sum_{k=1}^{d_L} W_k^+ \mathbf{s}_{i_k}^{k,L} + W_k^- \mathbf{t}_{j_k}^{k,L}, & a_{k,i_k,j_k}^{(L+1)} &= \sum_{k=1}^{d_L} (W_k^+ a_{i_k}^{k,L} + W_k^- b_{j_k}^{k,L}) + c, \\
\mathbf{s}_{k,n_{k,L}+i_k,m_{k,L}+j_k}^{(L+1)} &= \sum_{k=1}^{d_L} W_k^- \mathbf{s}_{i_k}^{k,L} + W_k^+ \mathbf{t}_{j_k}^{k,L}, & a_{k,n_{k,L}+i_k,m_{k,L}+j_k}^{(L+1)} &= \sum_{k=1}^{d_L} W_k^- a_{i_k}^{k,L} + W_k^+ b_{j_k}^{k,L}, \\
\mathbf{t}_{k,i_k,j_k}^{(L+1)} &= \mathbf{s}_{k,n_{k,L}+i_k,m_{k,L}+j_k}^{(L+1)}, & b_{k,i_k,j_k}^{(L+1)} &= a_{k,n_{k,L}+i_k,m_{k,L}+j_k}^{(L+1)}.
\end{aligned}
\tag{2}
$$

Thus the feasible parameters are given as the image of a polynomial map evaluated over a semialgebraic set. In turn, they form a semialgebraic set. Moreover, we have written the variables $\mathbf{s}_{k,i_k,j_k}^{(L+1)}$ etc. as a linear combination of the variables $\mathbf{s}_{i_k}^{k,L}$ etc., i.e. as the solution of a polynomial of degree 1 in these variables. By induction, each of the $\mathbf{s}_{i_k}^{k,L}$'s is a solution to a system of polynomials of degree $L + 1$. Substituting these polynomials into a polynomial of degree 1 yields a system of polynomials of degree $L + 2$.

We now discuss the number of monomials in this representation. By induction, we can choose each $n_{k,L} = 2m_{k,L}$ and $\log_2(m_{k,L}) \leq \sum_{k=1}^{L-1} 2^{L-1-k} \prod_{l=k}^{L-1} d_l$. Therefore

$$
\begin{aligned}
\log_2(m_{L+1}) &= \sum_{k=1}^{d_L} \log_2(n_{k,L}) + \sum_{k=1}^{d_L} \log_2(m_{k,L}) \\
&\leq d_L \left( \sum_{k=1}^{L-1} 2^{L-1-k} \prod_{l=k}^{L-1} d_l + 1 \right) + d_L \left( \sum_{k=1}^{L-1} 2^{L-1-k} \prod_{l=k}^{L-1} d_l \right) \\
&= \sum_{k=1}^{L-1} 2^{L-1-k} \prod_{l=k}^{L} d_l + d_L + \sum_{k=1}^{L-1} 2^{L-1-k} \prod_{l=k}^{L} d_l \\
&= 2 \left( \sum_{k=1}^{L-1} 2^{L-1-k} \prod_{l=k}^{L} d_l \right) + \sum_{k=L}^{L} 2^{L-k} \prod_{l=k}^{L} d_l \\
&= \sum_{k=1}^{L-1} 2^{L-k} \prod_{l=k}^{L} d_l + \sum_{k=L}^{L} 2^{L-k} \prod_{l=k}^{L} d_l \\
&= \sum_{k=1}^{L} 2^{L-k} \prod_{l=k}^{L} d_l
\end{aligned}
$$

and the stated bound follows.

We have just proven $\mathbf{ReLU}(d, d_1, \ldots, d_L, 1) \subseteq \psi(S)$, where $\mathcal{S}$ is the semialgebraic set which is implicitly defined through the recursive relations in (2). For the reverse inclusion, let $\theta \in \mathcal{S}$. By induction, there exist weights and biases defining a network in $\mathbf{ReLU}(d, d_1, \ldots, d_L)$ such that $a_{i_k}^{k,L}, \mathbf{s}_{i_k}^{k,L}, b_{j_k}^{k,L}, \mathbf{t}_{j_k}^{k,L}$ are the parameters of respective tropical rational function representations of some $f_k^{(L)}$'s. By definition of $\mathcal{S}$ there exists a solution for the system of linear equations (2) in indeterminates $W_k^+, W_k^-$. Any such solution gives rise to the weights of the last layer. $\qquad\square$

**Remark 3.3** ($\mathcal{S}$ is not unique). The proof of Theorem 3.2 reveals that in the above representation there are many redundancies which are expressed in the linear equations defining the semialgebraic set. Moreover, the choice of $\mathcal{S}$ is not unique: already for $L = 1$, choosing the representation $\theta' = (c, W, 0, \mathbf{0}, 0, \mathbf{0}) \in \Theta(d, 2, 1)$ yields the semialgebraic set $\mathcal{S}' = \{(a_1, \mathbf{s}_1, a_2, \mathbf{s}_2, b_1, \mathbf{t}_1) \mid a_2 = b_1 = 0, \mathbf{s}_2 = \mathbf{t}_1 = 0\}$. This reflects the fact that

for a given continuous piecewise linear function there exist several possible decompositions into a difference of convex piecewise linear functions. Either way, since these sets are low-dimensional relative to their ambient space, one may want to project this into a smaller ambient space. However, for any (nontrivial) projection, the degrees of the defining polynomials will in general be higher, and the number of defining inequalities will increase. There is no general statement which reasonably bounds the degree and number of the defining polynomials of the projection of a semialgebraic set.

For the number of monomials, we observe that a ReLU network with a total of $N$ ReLUs can only represent continuous piecewise linear functions which have at most $2^N$ linear pieces. Several more refined bounds on the number of linear pieces are available depending on the specific network architecture (Montúfar et al., 2014; Montúfar, 2017; Serra et al., 2018; Hinz & van de Geer, 2020). In particular, Zhang et al. (2018, Theorem 6.3) use similar arguments to the ones used in Theorem 3.2 to obtain a bound on the number of linear regions. We obtain a weaker bound since we focus on the (in)equalities defining the semialgebraic set. From such bounds on the number of linear regions one can directly obtain upper bounds on the minimal possible $n$ and $m$ by considering decompositions of continuous piecewise linear functions as differences of convex piecewise linear functions, such as those discussed by Kripfganz & Schulze (1987) and Schlüter & Darup (2020). Certain upper bounds for functions in two variables have been presented by Tran & Wang (2023) and for general piecewise linear functions by Hertrich et al. (2021, Proposition 4.3).

We close this section by noting that the statement of Theorem 3.2 is not exclusive to ReLU networks, but holds generally for networks with piecewise linear activation functions, for appropriate choices of $n$, $m$ and degree bounds on the polynomials. For example, the proof can be adapted to yield a representation of fixed network architectures with maxout units into a semialgebraic set in the parameter space of tropical rational functions for some bounded $n$ and $m$.

## 4 Decision Boundaries

In Section 2 we considered linear classifiers which separate the data into two parts by a hyperplane. In this section, we consider the analogous separating set for continuous piecewise-linear functions. This leads to a description of the decision boundary through the lens of (real and positive) tropical geometry.

Consider the tropical rational function $g \oslash h : \mathbb{R}^d \to \mathbb{R}$, where

$$g(\mathbf{x}) = \max_{i \in [n]}(a_i + \langle \mathbf{s}_i, \mathbf{x} \rangle) \quad \text{and} \quad h(\mathbf{x}) = \max_{j \in [m]}(b_j + \langle \mathbf{t}_j, \mathbf{x} \rangle)$$

are tropical signomials with parameters $a_i \in \mathbb{R}$, $\mathbf{s}_i \in \mathbb{R}^d$, $i \in [n]$ and $b_j \in \mathbb{R}$, $\mathbf{t}_j \in \mathbb{R}^d$, $j \in [m]$. The *decision boundary* of $g \oslash h$ is

$$\mathcal{B}(g \oslash h) = \left\{ \mathbf{x} \in \mathbb{R}^d \mid g(\mathbf{x}) \oslash h(\mathbf{x}) = 0 \right\} = \left\{ \mathbf{x} \in \mathbb{R}^d \mid \max_{i \in [n]}(a_i + \langle \mathbf{s}_i, \mathbf{x} \rangle) = \max_{j \in [m]}(b_j + \langle \mathbf{t}_j, \mathbf{x} \rangle) \right\}.$$

The decision boundary $\mathcal{B}(g \oslash h)$ splits the input space into two open parts $\mathcal{B}^+, \mathcal{B}^-$, where either the numerator $g$ or the denominator $h$ attains a higher value. A classifier $g \oslash h$ then separates the data $\mathcal{D}$ into two classes $\mathcal{D}_+ \subseteq \mathcal{B}^+$ and $\mathcal{D}_- \subseteq \mathcal{B}^-$.

It was noted (Zhang et al., 2018, Proposition 6.1) that $\mathcal{B}(g \oslash h)$ is a subcomplex of the tropical hypersurface $\mathcal{T}(g \oplus h)$. This observation was also used in the work of Alfarra et al. (2023). We will now make this statement more precise. Consider the tropical signomial $f(\mathbf{x}) = g(\mathbf{x}) \oplus h(\mathbf{x}) = \max_{i \in [N]}(a_i + \langle \mathbf{s}_i, \mathbf{x} \rangle)$, where $N = n + m$, $a_{n+j} = b_j$, $\mathbf{s}_{n+j} = \mathbf{t}_j$ for $j \in [m]$. Each such tropical polynomial divides the input space $\mathbb{R}^d$ into $N$ polyhedral regions, one for each $i^* \in [N]$, which are of the form

$$\mathcal{R}_{i^*} = \left\{ \mathbf{x} \in \mathbb{R}^d \mid a_{i^*} + \langle \mathbf{s}_{i^*}, \mathbf{x} \rangle = \max_{i \in [N]}(a_i + \langle \mathbf{s}_i, \mathbf{x} \rangle) \right\}.$$

Depending on the coefficients, some of these regions might be empty. The collection of regions forms a polyhedral subdivision of the input space $\mathbb{R}^d$. The collection of $(d-1)$-dimensional cells in this polyhedral subdivision is the *tropical hypersurface*

$$\mathcal{T}(f) = \left\{ \mathbf{x} \in \mathbb{R}^d \mid \text{the maximum in } f(\mathbf{x}) \text{ is attained in at least two terms} \right\}.$$

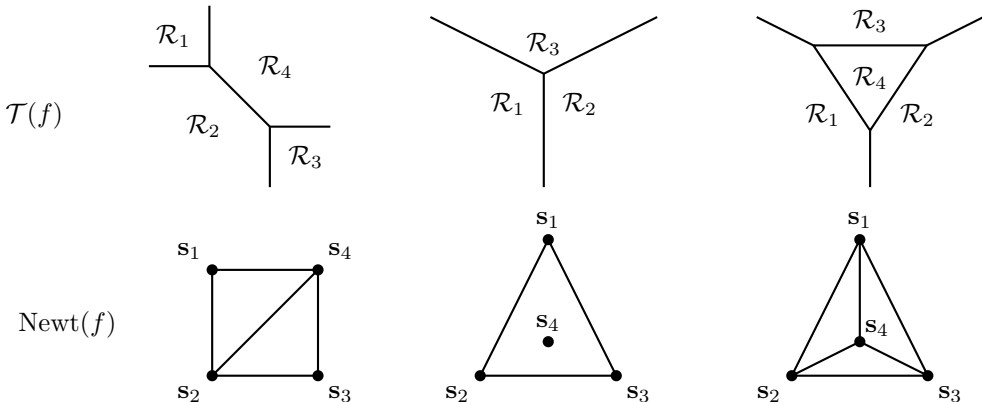

Figure 3: The three combinatorial types of generic hypersurfaces defined by tropical polynomials in two variables and 4 terms (top), and the dual regular subdivisions of their Newton polygons (bottom), as described in Example 4.1.

The tropical hypersurface is dual to a regular subdivision of the *Newton polytope* $\text{Newt}(f) = \text{conv}(\mathbf{s}_1, \ldots, \mathbf{s}_N) \subset \mathbb{R}^d$, which can be obtained in the following way: Consider the lifted polytope $\text{conv}((\begin{smallmatrix} a_1 \\ \mathbf{s}_1 \end{smallmatrix}), \ldots, (\begin{smallmatrix} a_N \\ \mathbf{s}_N \end{smallmatrix})) \subset \mathbb{R}^{d+1}$. Any facet of the lifted polytope has a unique outer normal vector (up to positive scaling). The upper hull is the collection of facets whose normal vector has a positive entry in the first coordinate. The projection of the upper hull onto the remaining $d$ coordinates forms a subdivision of $\text{Newt}(f)$, called a *regular subdivision*. The region $\mathcal{R}_{i*}$ is the set of linear functionals maximizing the vertex $(\begin{smallmatrix} a_{i*} \\ \mathbf{s}_{i*} \end{smallmatrix})$ of the lifted Newton polytope, and the intersection $\mathcal{R}_{i*} \cap \mathcal{R}_{j*}$ is contained in $\mathcal{T}(f)$ if and only if the pair $\mathbf{s}_{i*}, \mathbf{s}_{j*}$ form an edge in the regular subdivision. For more detailed expositions on this duality we refer the reader to Maclagan & Sturmfels (2015, Chapter 3.1) and Joswig (2021, Chapter 1.2).

**Example 4.1** (Tropical Hypersurfaces and Newton Polytopes)**.** Let $N = 4$ and $d = 2$, i.e. $f(\mathbf{x}) = a_1 \odot \mathbf{x}^{\odot \mathbf{s}_1} \oplus a_2 \odot \mathbf{x}^{\odot \mathbf{s}_2} \oplus a_3 \odot \mathbf{x}^{\odot \mathbf{s}_3} \oplus a_4 \odot \mathbf{x}^{\odot \mathbf{s}_4}$, where $a_i \in \mathbb{R}$ and $\mathbf{s}_i \in \mathbb{R}^2$. The Newton polytope $\text{Newt}(f)$ is the convex hull of the 4 points $\mathbf{s}_1, \mathbf{s}_2, \mathbf{s}_3, \mathbf{s}_4$ in $\mathbb{R}^2$, and is thus either a triangle or a square (except for degenerate cases where $\text{Newt}(f)$ is not full dimensional). Choosing generic values $a_1, a_2, a_3, a_4 \in \mathbb{R}$, we obtain three possible regular subdivisions and their respective dual complexes, as shown in Figure 3.

We now extend this duality for understanding tropical rational functions by assigning signs to each region. We have seen that $\mathcal{T}(g \oplus h)$ divides the input space into $n+m$ regions. We call the full-dimensional regions $\mathcal{R}_i$ with $i \in [n]$, i.e. those that correspond to terms of $g$, the *positive regions*. The regions $\mathcal{R}_j, j \in [m]$ corresponding to terms of $h$ are the *negative regions*. This terminology stems from the fact that $g(\mathbf{p}) \oslash h(\mathbf{p}) \geq 0$ if $\mathbf{p}$ lies in a positive region, and $g(\mathbf{p}) \oslash h(\mathbf{p}) \leq 0$ if $\mathbf{p}$ lies in a negative region. These regions are dual to vertices in the regular subdivision of the Newton polytope $\text{Newt}(g \oplus h)$, i.e. to those vertices which lie in the upper hull of the lifted Newton polytope. We equip each such vertex with the corresponding sign, obtaining *positive* and *negative vertices* of the regular subdivision. An edge of the subdivision is a *sign-mixed edge* if one of its vertices is positive and the other one is negative. Such a sign-mixed edge is dual to the intersection of a positive and a negative region, and thus $g \oslash h \equiv 0$ along this intersection. We summarize this construction as follows:

**Theorem 4.2.** *The decision boundary $\mathcal{B}(g \oslash h)$ is the $(d-1)$-dimensional subcomplex of $\mathcal{T}(g \oplus h)$ whose maximal regions are dual to sign-mixed edges of the subdivision of the Newton polytope $\text{Newt}(g \oplus h)$.*

This statement is already implicit in earlier literature on real and positive tropicalization. If $g, h$ are tropical polynomials, then the decision boundary $\mathcal{B}(g \oslash h)$ is the tropicalization of the intersection of the positive orthant with a (family of) hypersurface(s). Such a hypersurface is defined through any polynomial $G(\mathbf{x}) - H(\mathbf{x})$ such that $\text{trop}(G) = g, \text{trop}(H) = h$ and both $G$ and $H$ have only nonnegative coefficients. These connections between decision boundaries and (tropical) algebraic geometry are, as far as we know, widely unexplored. For more details on tropical positivity we refer the reader to Speyer & Williams (2005),Viro (2006) and

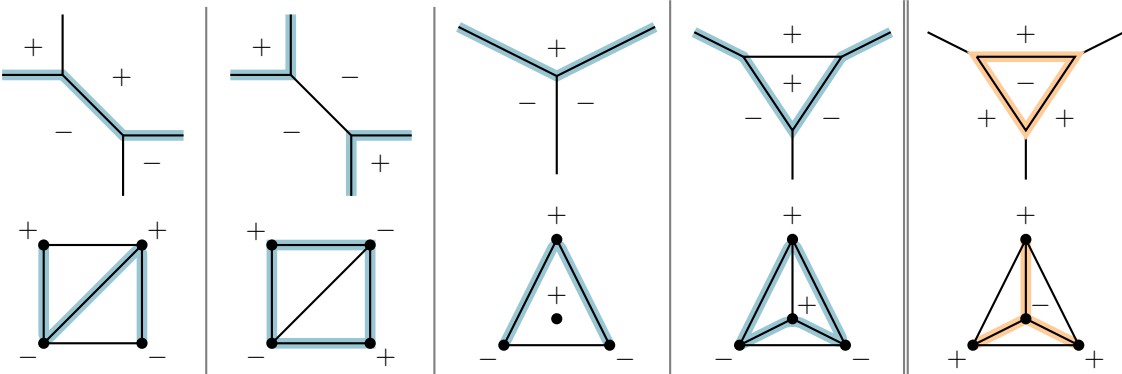

Figure 4: All combinatorial possibilities of positive and negative regions of $\mathcal{T}(g \oplus h)$ for $n = m = 2$, up to switching "+" and "−", as explained in Example 4.3. The decision boundary $\mathcal{B}(g \oslash h)$ (top) and its dual complex (bottom) are highlighted in blue. The right most column shows configurations of three positive regions and one negative region ($n = 3, m = 1$). The decision boundary is highlighted in orange.

Brandenburg et al. (2023). For an introduction to tropicalization of polynomials and hypersurfaces, we recommend the texts of Joswig (2021, Chapter 2) and Maclagan & Sturmfels (2015, Chapter 3.1).

**Example 4.3** (Decision Boundaries and Sign-Mixed Subcomplexes). Let $d = 2$ and let $g = a_1 \odot x^{\odot \mathbf{s}_1} \oplus a_2 \odot x^{\odot \mathbf{s}_2}$ and $h = b_1 \odot x^{\odot \mathbf{t}_1} \oplus b_2 \odot x^{\odot \mathbf{t}_2}$. The polynomial $g \oplus h$ consists of $N = 4$ terms, as the one in Example 4.1. Figure 4 shows all nondegenerate combinatorial possibilities of positive and negative regions, where a "+" indicates a region or vertex corresponding to a term of $g$, and a "−" indicates a region or vertex corresponding to a term of $h$. The decision boundary is the 1-dimensional subcomplex of $\mathcal{T}(g \oplus h)$ consisting of line segments and rays incident to a positive and a negative region. Figure 4 also shows a configuration where $g$ consists of $n = 3$ terms and $h$ is a monomial ($m = 1$). There, the decision boundary separates the space into a bounded (negative) cell and its (positive) complement.

The main focus of this article is the parameter space $\Theta(d, n, m)$ of tropical rational functions with $n$ terms in the numerator and $m$ terms in the denominator. It is thus natural to ask about the geometry of the set of parameters such that the decision boundary has a fixed combinatorial type. The decision boundary is a polyhedral complex. The *combinatorial type* of a polyhedral complex captures the combinatorics of the intersections and inclusions of its faces. Formally, it is defined as the isomorphism class of the partially ordered set of faces, ordered by inclusion.

By the discussion above, the decision boundary is dual to a subcomplex of a Newton polytope, so the set of parameters which gives a fixed combinatorial type of decision boundaries is subdivided into multiple (but finitely many) smaller sets, one for each combinatorial type of regular subdivisions of Newton polytopes defined by $n + m$ monomials. Each such smaller set is itself determined by the combinatorial type of the lifted Newton polytope. The set of parameters such that the Newton polytope has a fixed combinatorial type is the *realization space* of the lifted polytope, and the space of parameters $\Theta(d, n, m)$ can be partitioned into these realization spaces. It is known that such a realization space is a semialgebraic set, i.e. a finite union and intersection of solution sets to polynomial inequalities. We thus obtain the following result.

**Theorem 4.4.** *The parameter space $\Theta(d, n, m)$ of tropical rational functions is partitioned into semialgebraic sets, one for each combinatorial type of regular subdivisions of $n + m$ points in $d$-dimensional input space.*

However, the Universality Theorem of Realization Spaces certifies that realization spaces of polytopes can be arbitrarily complicated (Richter-Gebert & Ziegler, 1995). Moreover, observe that this partition into semialgebraic sets is completely independent of the data set $\mathcal{D}$. In Section 5 we will introduce the activation fan, which is a polyhedral fan subdividing the parameter space, and its geometry heavily depends on the geometry of the data. The aforementioned semialgebraic sets thus may or may not intersect cones in the activation fan. In other words, these are distinct partitions of the parameter space whose structures are incompatible with one another.

**Example 4.5** (Realization Space of Decision Boundaries). We consider a 1-dimensional input space $d = 1$, where $g(\mathbf{x}) = \max(a_1 + \langle \mathbf{s}_1, x \rangle, a_2 + \langle \mathbf{s}_2, x \rangle)$ and $h(\mathbf{x}) = b_1 + \langle \mathbf{t}_1, \mathbf{x} \rangle$. We describe the semialgebraic nature of the realization space in $(d+1)(n+m) = 6$-dimensional parameter space $\Theta(1, 2, 1)$. The Newton polytope $\text{Newt}(g \oplus h)$ is a line segment which is the convex hull of 3 points $\mathbf{s}_1, \mathbf{s}_2, \mathbf{t}_1 \in \mathbb{R}^1$ (dual to two positive regions and one negative region), and the decision boundary consists of at most 2 distinct points. We want to characterize the set of parameters such that $\mathcal{B}(g \oslash h)$ consists of two distinct points. In terms of the Newton polytope, this is the case when $\mathbf{t}_1 \in \text{int}(\text{conv}(\mathbf{s}_1, \mathbf{s}_2))$ and $\binom{b_1}{\mathbf{t}_1}$ lies above the lifted line segment $\text{conv}(\binom{a_1}{\mathbf{s}_1}, \binom{a_2}{\mathbf{s}_2})$, so that $\text{Newt}(g \oplus h)$ is subdivided into the two line segments $\text{conv}(\mathbf{s}_1, \mathbf{t}_1)$ and $\text{conv}(\mathbf{t}_1, \mathbf{s}_2)$. A subdivision into two line segments arises if and only if there exists a $0 < \lambda < 1$ such that $\lambda \mathbf{s}_1 + (1 - \lambda)\mathbf{s}_2 = \mathbf{t}_1$, and $\lambda a_1 + (1 - \lambda)a_2 < b_1$. The semialgebraic set in parameter space is the coordinate projection of the set of vectors $(a_1, \mathbf{s}_1, a_2, \mathbf{s}_2, a_3, \mathbf{s}_3, \lambda) \in \mathbb{R}^7$ satisfying the above inequalities, by projecting away the $\lambda$-coordinate. Concretely, the coordinate projection is the semialgebraic set

$$\begin{aligned}
\mathcal{S} = &\{(a_1, \mathbf{s}_1, a_2, \mathbf{s}_2, b_1, \mathbf{t}_1) \mid \mathbf{s}_1 < \mathbf{t}_1 < \mathbf{s}_2, a_1(\mathbf{t}_1 - \mathbf{s}_2) + a_2(\mathbf{s}_1 - \mathbf{t}_1) < b_1(\mathbf{s}_1 - \mathbf{s}_2)\} \\
&\cup \{(a_1, \mathbf{s}_1, a_2, \mathbf{s}_2, b_1, \mathbf{t}_1) \mid \mathbf{s}_2 < \mathbf{t}_1 < \mathbf{s}_1, a_2(\mathbf{t}_1 - \mathbf{s}_1) + a_1(\mathbf{s}_2 - \mathbf{t}_1) < b_1(\mathbf{s}_2 - \mathbf{s}_1)\},
\end{aligned}$$

which is defined by unions and intersections of linear and quadratic polynomial inequalities. The parameters defining decision boundaries consisting of a single point are $\Theta(1, 2, 1) \setminus \mathcal{S}$.

# 5 Activation Fan and Activation Polytope

In Section 2 we have seen that the parameter space of linear classifiers allows for a subdivision induced by a hyperplane arrangement. This subdivision is the normal fan of a zonotope, and its cells correspond to maximal covectors of an oriented matroid. In this section we make a first step towards a generalization of this theory for continuous piecewise linear functions. Following the ideas from Section 4, we consider subdivisions of the input space into $n + m$ regions, which are induced by tropical hypersurfaces. This allows us to introduce the *activation fan*, a polyhedral fan which is the normal fan of the *activation polytope*. As an analog to covectors, we label the cones in this fan by bipartite graphs, called *activation patterns*. In the later Section 6 we will then assign signs to obtain the full analog for tropical rational functions.

In Section 5.1 we introduce the concepts and investigate the general combinatorial and geometric structure of these objects. In Section 5.2 we relate these to known concepts, namely oriented matroids and tropical oriented matroids.

## 5.1 General Structure

Let $\mathcal{D} \subseteq \mathbb{R}^d$ be a fixed finite data set. In Section 3 we have seen that to any vector of parameters $\theta \in \Theta(d, n, m)$ we can associate a tropical rational function $g_\theta \oslash h_\theta$, which induces a classification of the data set. Our goal is to understand the underlying combinatorics of this separation in parameter space. In Section 4 we have seen that the combinatorics of this separation is determined by the combinatorics of the tropical hypersurface $\mathcal{T}(g \oplus h)$. Here, $g \oplus h$ is a tropical signomial with $n + m$ terms and $\mathcal{T}(g \oplus h)$ separates the data into $n + m$ classes. We devote this section to the study of the combinatorics of tropical signomials with $N = n + m$ terms within their parameter space. Later, in Section 6 we will extend these considerations to tropical rational functions.

Consider a fixed finite data set $\mathcal{D} = \{\mathbf{p}_1, \ldots, \mathbf{p}_M\} \subset \mathbb{R}^d$ and denote a tropical signomial $f \colon \mathbb{R}^d \to \mathbb{R}$, $f(\mathbf{x}) = \max_{i \in [N]}(a_i + \langle \mathbf{s}_i, \mathbf{x} \rangle)$, which is uniquely defined by its parameter vector $\theta = (a_1, \mathbf{s}_1, \ldots, a_N, \mathbf{s}_N)$. We denote by $\Theta(d, N)$ the $N(d+1)$-dimensional parameter space of tropical signomials with $N$ terms (in numerator and denominator combined). Given a vector of parameters $\theta \in \Theta(d, N)$, we write $f_\theta$ for the corresponding function. For a graph $G = (V(G), E(G))$ and a node $v \in V(G)$ we denote the neighborhood of $v$ by $N(v; G) = \{w \in V(G) \mid vw \in E(G)\}$. For two graphs $G = (V, E), G' = (V, E')$ on the same set of nodes, we write $G \cup G'$ for the graph with nodes $V$ and edges $E \cup E'$.

**Definition 5.1** (Activation Pattern of Tropical Signomial). The data point $\mathbf{p}_j \in \mathcal{D}$ *activates* the $(i^*)^{\text{th}}$ term of the tropical signomial $f(\mathbf{x}) = \max_{i \in [N]}(a_i + \langle \mathbf{s}_i, \mathbf{x} \rangle)$ if $f(\mathbf{p}_j) = a_{i^*} + \langle \mathbf{s}_{i^*}, \mathbf{x} \rangle$. The *activation pattern* of $(f, \mathcal{D})$ is the bipartite graph $G = (V(G), E(G))$ on nodes $V(G) = \mathcal{D} \sqcup [N]$ with edges $E(G) = \{\mathbf{p}i \mid$

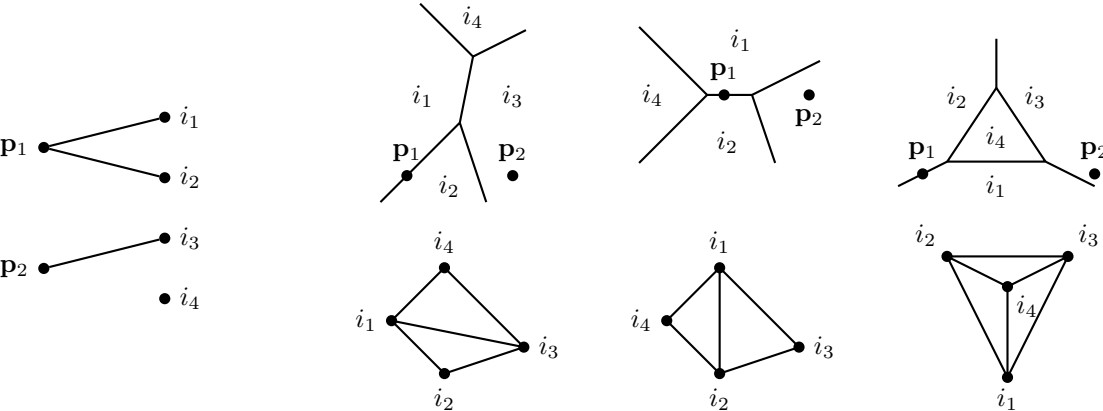

(a) Activation pattern.  (b) Subdivisions in input space (top) and dual Newton polytopes (bottom).

Figure 5: The activation pattern $G$ from Example 5.2, partitions $\mathcal{T}(f_\theta)$ of the input space for three different values of $\theta \in C_\mathcal{D}(G)$ and dual subdivisions of Newton polytopes $\mathrm{Newt}(f_\theta)$.

$\mathbf{p}$ activates the $i^{\text{th}}$ term of $f$}. The degree of a vertex $\mathbf{p}$, $\deg(\mathbf{p}) = |N(\mathbf{p}; G)|$, is the number of terms that it activates. The *activation cone* f a fixed bipartite graph $G = (\mathcal{D} \sqcup [N], E(G))$ is the polyhedral cone

$$C_\mathcal{D}(G) = \mathrm{cl}\left(\{\theta = (a_1, \mathbf{s}_1, \ldots, a_N, \mathbf{s}_N) \in \Theta(d, N) \mid G \text{ is the activation pattern of } (f_\theta, \mathcal{D})\}\right)$$

in the parameter space of tropical signomials, where cl denotes the Euclidean closure.

In the notation of Section 4, the point $\mathbf{p}$ activates the $(i^*)^{\text{th}}$ term if and only if $\mathbf{p} \in \mathcal{R}_{i^*}$, where $\mathcal{R}_{i^*}$ is a maximal region of the subdivision of $\mathbb{R}^d$ induced by the tropical hypersurface $\mathcal{T}(f)$. The activation cone $C_\mathcal{D}(G)$ is thus the space of tropical signomials where each data point $\mathbf{p}$ lies in a fixed set of regions, namely in regions indexed by $N(\mathbf{p}; G)$. The activation pattern may be thought of as a multivalued generalization of a covector of an oriented matroid, and the corresponding activation cone serves as an analog of a chamber in the hyperplane arrangement from Section 2.

**Example 5.2** (Activation Patterns). Let $\mathcal{D} = \{\mathbf{p}_1, \mathbf{p}_2\} \subset \mathbb{R}^2$ with $\mathbf{p}_1 = (0, 0), \mathbf{p}_2 = (1, 0)$, let $N = 4$ and $G$ be the activation pattern on nodes $V(G) = \mathcal{D} \sqcup \{i_1, i_2, i_3, i_4\}$ with edges $E(G) = \{(\mathbf{p}_1, i_1), (\mathbf{p}_1, i_2), (\mathbf{p}_2, i_3)\}$. This graph represents the set of tropical signomials in $d = 2$ variables with $N = 4$ terms such that the point $\mathbf{p}_1$ lies in the intersection of the regions $\mathcal{R}_{i_1}, \mathcal{R}_{i_2}$ corresponding to the first and second term, and the point $\mathbf{p}_2$ lies in the region $\mathcal{R}_{i_3}$. Figure 5 shows some tropical hypersurfaces $\mathcal{T}(f_\theta)$ and dual subdivisions of $\mathrm{Newt}(f_\theta)$ for $\theta \in C_\mathcal{D}(G)$. More explicitly, the parameter of the leftmost tropical hypersurface can be chosen as $\theta = (0, -1, 1, \ 0, 0, 0, \ -1, 1.5, 0.5, \ -2, 0, 2)$. Recall from Theorem 4.4 that the parameter space is subdivided into semialgebraic sets, one for each combinatorial type of subdivisions of $\mathrm{Newt}(f_\theta)$. This example also shows that the cone $C_\mathcal{D}(G)$ intersects all such semialgebraic sets nontrivially.

Note that, depending on the geometry of the data $\mathcal{D}$, the cone $C_\mathcal{D}(G)$ may be empty. On the other hand, every bipartite graph $H = (\mathcal{D} \sqcup [N], E(H))$ defines a polyhedral cone

$$\tilde{C}_\mathcal{D}(H) = \{\theta = (a_1, \mathbf{s}_1, \ldots, a_N, \mathbf{s}_N) \in \Theta(d, N) \mid a_i + \langle \mathbf{s}_i, \mathbf{p} \rangle \leq a_{i^*} + \langle \mathbf{s}_{i^*}, \mathbf{p} \rangle \text{ for all } \mathbf{p} \in \mathcal{D}$$
$$\text{and all } i, i^* \in [N] \text{ such that } \mathbf{p}i^* \in E(H)\}.$$

If $\tilde{C}_\mathcal{D}(H) \neq \emptyset$ then $\tilde{C}_\mathcal{D}(H) = C_\mathcal{D}(G)$, where $G$ is the smallest activation pattern such that $E(H) \subseteq E(G)$. We write $G = \overline{H}$ for this smallest graph.

**Remark 5.3** (Checking existence through linear programs). In theory it may be difficult to determine whether $\tilde{C}_\mathcal{D}(H)$ is empty or not. In practice, this can be done via a linear program, where $\tilde{C}_\mathcal{D}(H)$ is the set of feasible solutions. This program is defined by $|E(H)|(N-1)$ linear inequalities (where $|E(H)| \leq N|\mathcal{D}|$), and $\tilde{C}_\mathcal{D}(H) \neq \emptyset$ if and only if the corresponding linear program has at least one solution. The activation

pattern $\overline{H}$ can be computed by checking the validity of all $N(N-1)|\mathcal{D}|$ inequalities on $\tilde{C}_\mathcal{D}(H)$. More specifically, $\overline{H}$ contains the edge $i^*\mathbf{p}$ if $a_i + \langle \mathbf{s}_i, \mathbf{p} \rangle \leq a_{i^*} + \langle \mathbf{s}_{i^*}, \mathbf{p} \rangle$ is a valid inequality for all parameters in $\tilde{C}_\mathcal{D}(H)$ for all $i \in [N]$.

**Definition 5.4** (Activation Fan). The *activation fan* $\Sigma_\mathcal{D}(N)$ of a finite data set $\mathcal{D} \subset \mathbb{R}^d$ and tropical signomials with $N$ monomials is the set of all activation cones in the parameter space $\Theta(d, N)$.

**Proposition 5.5.** *The activation fan $\Sigma_\mathcal{D}(N)$ is a complete polyhedral fan, i.e. a collection of polyhedral cones such that the intersection of any two cones is a face of both, and that their union covers the entire ambient space.*

*Proof.* Let $G, G'$ be activation patterns. $C_\mathcal{D}(G)$ is indeed a polyhedral cone, as it is defined by linear inequalities, and for every $\theta \in C_\mathcal{D}(G)$, $\lambda \in \mathbb{R}$ one has $\lambda\theta \in C_\mathcal{D}(G)$. Its faces are of the form $C_\mathcal{D}(G')$ where $E(G) \subset E(G')$, and any such activation pattern defines a face of $C_\mathcal{D}(G)$. For any pair of activation patterns holds $C_\mathcal{D}(G) \cap C_\mathcal{D}(G') = C_\mathcal{D}(\overline{G \cup G'})$ is a face both of $C_\mathcal{D}(G)$ and $C_\mathcal{D}(G')$. Therefore, the collection of all activation cones forms a polyhedral fan. Every vector of parameters $\theta = (a_1, s_1, \ldots, a_N, s_N) \in \Theta(d, N)$ gives rise to a tropical signomial $f_\theta(\mathbf{x})$, and $\theta \in C_\mathcal{D}(H)$, where $H$ is the activation pattern of $(f_\theta, \mathcal{D})$. Thus, the activation fan is complete. $\square$

The activation fan serves as a generalization of the polyhedral fan $\Sigma_\mathcal{D}$ from Section 2 which is induced by the hyperplane arrangement $\mathcal{H}_\mathcal{D}$. The fan $\Sigma_\mathcal{D}$ is the normal fan of a zonotope, i.e. the Minkowski sum of 1-dimensional simplices, one for each data point in $\mathcal{D}$. We now show an analogous statement for the activation fan $\Sigma_\mathcal{D}(N)$.

**Definition 5.6** (Activation Polytope). A point $\mathbf{p} \in \mathbb{R}^d$ defines a simplex

$$\Delta(\mathbf{p}) = \operatorname{conv}((1, \mathbf{p}, 0, \mathbf{0}, \ldots, 0, \mathbf{0}), (0, \mathbf{0}, 1, \mathbf{p}, \ldots, 0, \mathbf{0}), (0, \mathbf{0}, 0, \mathbf{0}, \ldots, 1, \mathbf{p})) \subset \Theta(d, N)$$

of dimension $(N-1)$, where $\mathbf{0} = (0, \ldots, 0) \in \mathbb{R}^d$. For a finite data set $\mathcal{D} \subset \mathbb{R}^d$, the *activation polytope* $P_\mathcal{D}(N)$ is the Minkowski sum $P_\mathcal{D}(N) = \sum_{\mathbf{p} \in \mathcal{D}} \Delta(\mathbf{p})$.

**Theorem 5.7.** *The activation fan $\Sigma_\mathcal{D}(N)$ is the normal fan of the activation polytope $P_\mathcal{D}(N)$.*

*Proof.* Since $P_\mathcal{D}(N)$ is a Minkowski sum, the normal fan of $P_\mathcal{D}(N)$ is the common refinement of the normal fans of its Minkowski summands $\Delta(\mathbf{p}), \mathbf{p} \in \mathcal{D}$. For such a simplex, the vertices are of the form

$$\mathbf{v}_i = (\underbrace{0, \ldots, 0}_{(i-1)(d+1)}, 1, \mathbf{p}, \underbrace{0, \ldots, 0}_{(N-i)(d+1)}), \quad \text{for } i \in [N].$$

We describe the normal cone of a vertex $\mathbf{v}_{i^*}$ for some $i^* \in [N]$. Since $\Delta(\mathbf{p})$ is a simplex, the normal cone is a simplicial cone, whose facets are orthogonal to the directions of edges $\mathbf{v}_{i^*}\mathbf{v}_i$ for all $i \in [N] \setminus \{i^*\}$. The normal cone is thus

$$
\begin{aligned}
N_{\mathbf{v}_{i^*}}^{\mathbf{P}} &= \{\theta \in \Theta(d, N) \mid \langle \theta, \mathbf{v}_{i^*} - \mathbf{v}_i \rangle \geq 0 \ \forall i \in [N]\} \\
&= \{(a_1, \mathbf{s}_1, \ldots, a_N, \mathbf{s}_N) \mid a_{i^*} + \langle \mathbf{s}_i^*, \mathbf{p} \rangle \geq a_i + \langle \mathbf{s}_i, \mathbf{p} \rangle \ \forall i \in [N]\} \\
&= \{(a_1, \mathbf{s}_1, \ldots, a_N, \mathbf{s}_N) \mid \max_{i \in [N]}(a_i + \langle \mathbf{s}_i, p \rangle) = a_{i^*} + \langle \mathbf{s}_{i^*}, p \rangle\}.
\end{aligned}
$$

Any maximal cone of the normal fan of $P_\mathcal{D}(N)$ is of the form $C = \bigcap_{\mathbf{p} \in \mathcal{D}} N_{\mathbf{v}_{i^*(\mathbf{p})}}^{\mathbf{P}}$, where $i^*(\mathbf{p}) \in [N]$ with possible repetitions. In other words,

$$C = \{(a_1, \mathbf{s}_1, \ldots, a_N, \mathbf{s}_N) \mid \max_{i \in [N]}(a_i + \langle \mathbf{s}_i, \mathbf{p} \rangle) = a_{i^*(\mathbf{p})} + \langle \mathbf{s}_{i^*(\mathbf{p})}, \mathbf{p} \rangle \ \forall \mathbf{p} \in \mathcal{D}\},$$

so $C = C_\mathcal{D}(G)$, where the activation pattern is the bipartite graph $G = (\mathcal{D} \sqcup [N], E(G))$ with edge set $E(G) = \{\mathbf{p} i^*(\mathbf{p}) \mid \mathbf{p} \in \mathcal{D}, i^*(\mathbf{p}) \in [N], C \subseteq N_{\mathbf{v}_{i^*(\mathbf{p})}}^{\mathbf{P}}\}$. We have shown that the maximal cones of the normal fan of $P_\mathcal{D}(N)$ are contained in $\Sigma_\mathcal{D}(N)$. Since the normal fan is a complete fan, this finishes the proof. $\square$

**Remark 5.8** (Linear Classifiers are a Special Case). If $N = 2$ then a tropical hypersurface is defined through the linear equation $\langle \mathbf{s}_1 - \mathbf{s}_2, \mathbf{x} \rangle + (a_1 - a_2) = 0$. The activation polytope $P_{\mathcal{D}}(2)$ is then equivalent to the zonotope $P_{\mathcal{D}} = \sum_{p \in \mathcal{D}} \text{conv}((0, \mathbf{0}), (1, \mathbf{p})) \subseteq \mathbb{R}^{d+1}$ from Section 2, however, this equivalence is not obvious. To make this precise, consider the activation polytope $P_{\mathcal{D}}(2) = \sum_{\mathbf{p} \in D} \text{conv}((1, \mathbf{p}, 0, \mathbf{0}), (0, \mathbf{0}, 1, \mathbf{p})) \subseteq \mathbb{R}^{2(d+1)}$. The activation polytope is contained in the $(d+1)$-dimensional affine subspace with $e_i = e_{i+d+1}$ for $i \in [d+1]$. Projecting $P_{\mathcal{D}}(2)$ onto the first $d + 1$ coordinates induces an isomorphism between the zonotope $P_{\mathcal{D}}$ and the activation polytope $P_{\mathcal{D}}(2)$. An activation pattern is a bipartite graph with nodes $\mathcal{D} \sqcup \{1, 2\}$. The isomorphism identifies the labeling $\{1, 2\}$ with the labeling $\{+, -\}$, recovering the covectors of the oriented matroid.

**Proposition 5.9.** *The dimension of the activation polytope $P_{\mathcal{D}}(N) \subset \Theta(d, N)$ is $(N-1)(\dim(\text{aff}(\mathcal{D})) + 1)$, where $\text{aff}(\mathcal{D})$ denotes the smallest affine subspace containing $\mathcal{D}$.*

*Proof.* Recall that $\dim(P_{\mathcal{D}}(N)) = \dim(\text{aff}(P_{\mathcal{D}}(N)))$. Moreover, the linear space which is parallel to $\text{aff}(P_{\mathcal{D}}(N))$ is orthogonal to the lineality space $\mathcal{L}$ of the normal fan of $P_{\mathcal{D}}(N)$, i.e. the largest linear space which is contained in each cone of the fan. By construction, the normal fan of $P_{\mathcal{D}}(N)$ is a common refinement of the normal fans of the simplices $\Delta(\mathbf{p}), \mathbf{p} \in \mathcal{D}$, and so the lineality space $\mathcal{L}$ of the normal fan of $P_{\mathcal{D}}(N)$ is the intersection of the lineality spaces of the simplices. We thus characterize the lineality space $\mathcal{L}(\mathbf{p})$ of the normal fan of a simplex $\Delta(\mathbf{p})$ for a fixed $\mathbf{p} \in \mathcal{D}$. Again, $\mathcal{L}(\mathbf{p})$ is orthogonal to the linear space parallel to $\text{aff}(\Delta(\mathbf{p}))$. Since $\Delta(\mathbf{p})$ is a $(N - 1)$-dimensional simplex, we have $\dim(\mathcal{L}(\mathbf{p})) = (d + 1)N - \dim(\text{aff}(\Delta(\mathbf{p}))) = (d + 1)N - (N - 1) = dN + 1$. By construction, $\mathcal{L}(\mathbf{p})$ contains

$$(\mathbf{e}_i, \mathbf{e}_i, \ldots, \mathbf{e}_i), \ i \in [d + 1], \ \text{and} \ (V(\mathbf{p}), \mathbf{0}, \ldots, \mathbf{0}), \ (\mathbf{0}, V(\mathbf{p}), \ldots, \mathbf{0}), \ \ldots, \ (\mathbf{0}, \mathbf{0}, \ldots, V(\mathbf{p})),$$

where $V(\mathbf{p}) = (1, \mathbf{p})^{\perp}$. We denote $E = \text{span}(\{(\mathbf{e}_i, \ldots, \mathbf{e}_i) \mid i \in [d + 1]\})$ and $W(\mathbf{p}) = \text{span}((V(\mathbf{p}), \mathbf{0}, \ldots, \mathbf{0}), \ldots, (\mathbf{0}, \mathbf{0}, \ldots, V(\mathbf{p}))$ The dimension of the total linear span is $\dim(E) + \dim(W(\mathbf{p})) - \dim(E \cap W(\mathbf{p})) = (d + 1) + Nd - d = Nd + 1$, so the span equals $\mathcal{L}(\mathbf{p})$. The intersection of all these lineality spaces $\mathcal{L}(\mathbf{p}), \mathbf{p} \in \mathcal{D}$ is spanned by $E$ and $W = \bigcap_{\mathbf{p} \in \mathcal{D}} W(\mathbf{p}) = \text{span}((V, \mathbf{0}, \ldots, \mathbf{0}), \ldots, (\mathbf{0}, \mathbf{0}, \ldots, V))$, where $V = \bigcap_{\mathbf{p} \in \mathcal{D}} V(\mathbf{p}) = \text{span}(\{1\} \times \text{aff}(\mathcal{D}))^{\perp}$. We thus obtain for the dimension of the lineality space

$$\begin{aligned}
\dim(\mathcal{L}) &= \dim(E) + \dim(W) - \dim(E \cap W) \\
&= (d + 1) + N(d - \dim(\text{aff}(\mathcal{D}))) - (d - \dim(\text{aff}(\mathcal{D}))) \\
&= (d + 1) + (N - 1)(d - \dim(\text{aff}(\mathcal{D}))).
\end{aligned}$$

This yields $\dim(P_{\mathcal{D}}(N)) = N(d + 1) - \dim(\mathcal{L}) = (N - 1)(\dim(\text{aff}(\mathcal{D})) + 1)$. $\square$

From the above proof we immediately obtain the following dual result.

**Proposition 5.10.** *The lineality space of the activation fan $\Sigma_{\mathcal{D}}(N)$ has dimension $(d+1)+(N-1)(d-\text{aff}(\mathcal{D}))$ and is generated by*

$$(\mathbf{e}_i, \mathbf{e}_i, \ldots, \mathbf{e}_i), \ i \in [d + 1], \ \text{and} \ (V, \mathbf{0}, \ldots, \mathbf{0}), (\mathbf{0}, V, \ldots, \mathbf{0}), \ldots, (\mathbf{0}, \mathbf{0}, \ldots, V),$$

*where $\mathbf{e}_i \in \mathbb{R}^{d+1}$ denotes a standard basis vector, and where $V = \text{span}(\{1\} \times \text{aff}(\mathcal{D}))^{\perp} \subseteq \mathbb{R}^{d+1}$ is a $(d - \text{aff}(\mathcal{D}))$-dimensional vector space.*

In the linear case, the cells of the hyperplane arrangement in parameter space are labelled by covectors of the oriented matroid, and the activation patterns naturally generalize covectors. The maximal chambers are labelled by covectors without $0$ entries, corresponding to the fact that each data point in input space lies on precisely one side of the dual hyperplane. We now show that the analog holds for activation patterns: Given a parameter vector in the interior of a maximal cone in the activation fan, each data point lies in precisely one region of the dual tropical hypersurface in input space.

As in the proof of Theorem 5.7, let $\mathbf{v}_i(\mathbf{p}) = (0, \ldots, 0, 1, \mathbf{p}, 0, \ldots, 0)$ denote the $i^{\text{th}}$ vertex of $\Delta(\mathbf{p})$. Since $P_{\mathcal{D}}(N)$ is a Minkowski sum, every face of $P_{\mathcal{D}}(N)$ can be written uniquely as a sum of faces of $\Delta(\mathbf{p})$, for $\mathbf{p} \in \mathcal{D}$. In the following, we write $F = \sum_{\mathbf{p} \in \mathcal{D}} F(\mathbf{p})$ for a face of $P_{\mathcal{D}}(N)$, where $F(\mathbf{p})$ is a face of $\Delta(\mathbf{p})$.

**Proposition 5.11.** *Let $C_{\mathcal{D}}(G) \in \Sigma_{\mathcal{D}}(N)$, and let $F \in P_{\mathcal{D}}(N)$ be the dual face. Then $F = \sum_{\mathbf{p} \in \mathcal{D}} \text{conv}(\mathbf{v}_i(\mathbf{p}) \mid i \in N(\mathbf{p}; G))$ and $N(\mathbf{p}; G) = \{i \in [N] \mid \mathbf{v}_i(\mathbf{p}) \text{ is a vertex of } F(\mathbf{p})\}$.*

*Proof.* We have that $C_{\mathcal{D}}(G) = \bigcap_{\mathbf{p} \in \mathcal{D}} N_{F(\mathbf{p})}$, where $N_{F(\mathbf{p})}$ is the normal cone of the face $F(\mathbf{p})$. Given any $\theta \in N_{F(\mathbf{p})}$ with associated tropical signomial $f_\theta$, the activated terms of $f_\theta(\mathbf{p})$ are $\{i \in [N] \mid \mathbf{v}_i(\mathbf{p})$ is a vertex of $F(\mathbf{p})\}$. Thus, for any activation cone $C_{\mathcal{D}}(H) \subseteq N_{F(\mathbf{p})}$ with activation pattern $H$ we have that $N(\mathbf{p}; H) = \{i \in [N] \mid \mathbf{v}_i(\mathbf{p})$ is a vertex of $F(\mathbf{p})\}$. Given $\theta \in C_{\mathcal{D}}(G) = \bigcap_{\mathbf{p} \in \mathcal{D}} N_{F(\mathbf{p})}$ the edges of the activation pattern are thus $\bigcup_{\mathbf{p} \in \mathcal{D}} \{\mathbf{p}i \mid i \in [N], \mathbf{v}_i(\mathbf{p})$ is a vertex of $F(\mathbf{p})\}$. Dually, this yields $F = \sum_{p \in \mathcal{D}} \mathrm{conv}\,(\mathbf{v}_i(\mathbf{p}) \mid i \in N(\mathbf{p}; G))$. $\square$

Since every face has at least one vertex and every vertex is exactly the Minkowski sum of vertices we get the following.

**Corollary 5.12.** *If $G$ is an activation pattern, then $\deg(\mathbf{p}) \geq 1$ for all $\mathbf{p} \in \mathcal{D}$. The activation cone $C_{\mathcal{D}}(G)$ is of maximal dimension if and only if $\deg(\mathbf{p}) = 1$ for all $\mathbf{p} \in \mathcal{D}$.*

In the linear classification case, we have seen that any target dichotomy separates the data into two sets $\mathcal{D}_+^{C^*}, \mathcal{D}_-^{C^*}$, and this dichotomy exists as a maximal covector of the oriented matroid if and only if $\mathrm{conv}(\mathcal{D}_+^{C^*}) \cap \mathrm{conv}(\mathcal{D}_-^{C^*}) = \emptyset$. Note that under the identification of a covector with a bipartite graph $G$ on nodes $\mathcal{D} \sqcup \{+, -\}$, we have that $\mathcal{D}_+^{C^*}$ is the set of points which are neighbors of $+$ in $G$, i.e. $\mathcal{D}_+^{C^*} = N(+; G)$, and similarly $\mathcal{D}_-^{C^*} = N(-; G)$. We now discuss necessary conditions on the geometry of the data in the more general case, i.e. such that an activation pattern or cone may exist in the activation fan.

**Theorem 5.13.** *Let $C_{\mathcal{D}}(G) \in \Sigma_{\mathcal{D}}(N)$ be a nonempty cone in the activation fan. Then the data set $\mathcal{D} \subset \mathbb{R}^d$ satisfies the following necessary conditions:*

(i) *(Convexity for maximal cones) If $C_{\mathcal{D}}(G)$ is maximal, then for every distinct $i, j \in [N]$ one has $\mathrm{conv}(N(i; G)) \cap \mathrm{conv}(N(j; G)) = \emptyset$.*

(ii) *(Convexity for arbitrary cones) Let $i \in [N]$ be a region and $\mathbf{p}_1, \ldots, \mathbf{p}_k \in N(i; G)$. Then for every $\mathbf{p} \in \mathcal{D}$ one has: $\mathbf{p} \in \mathrm{conv}(\mathbf{p}_1, \ldots, \mathbf{p}_k) \implies \mathbf{p} \in N(i; G)$.*

(iii) *(Regularity) There exists a subdivision of the input space into maximal regions $\mathcal{R}_1, \ldots, \mathcal{R}_N$ such that $N(i; G) \subseteq \mathcal{R}_i$ for each $i \in [N]$, and a dual subdivision of $N$ points which is regular.*

*Proof.* For cones of arbitrary dimension, (ii) follows from convexity of the regions of $\mathcal{T}(f_\theta)$ for any $\theta \in C_{\mathcal{D}}(G)$. By Corollary 5.12 the activation patterns of maximal cones satisfy $N(i; G) \cap N(j; G) = \emptyset$ for all distinct $i, j \in [N]$. This, together with (ii) implies (i). Finally, since $C_{\mathcal{D}}(G)$ is nonempty, there exists a tropical signomial $f_\theta$ such that $G$ is the activation pattern, and the tropical hypersurface $\mathcal{T}(f_\theta)$ induces the subdivision. By Section 4, this subdivision is dual to a regular subdivion of the Newton polytope $\mathrm{Newt}(f_\theta)$. $\square$

We have given necessary conditions for an activation cone to be nonempty by considering the geometry of the data. Recall that the set of covectors of oriented matroids obey the axiom system (C I)–(C IV) on page 6. The following result mimics a system in this spirit to describe the set of activation patterns.

**Theorem 5.14.** *Let $\mathcal{G}$ be the set of activation patterns of $\Sigma_{\mathcal{D}}(N)$. Then $\mathcal{G}$ satisfies the following properties.*

(A I) *(Complete Graph) $K_{N,\mathcal{D}} \in \mathcal{G}$.*

(A II) *(Symmetry) $G \in \mathcal{G} \implies$ any graph isomorphic to $G$ which arises through relabeling of the nodes $i \in [N]$ is contained in $\mathcal{G}$.*

(A III) *(Composition) If $G, H \in \mathcal{G}$ then $(G \circ H) \in \mathcal{G}$, where*

$$N(\mathbf{p}; G \circ H) = \begin{cases} N(\mathbf{p}; G) & \text{if } N(\mathbf{p}; G) \cap N(\mathbf{p}; H) = \emptyset, \\ N(\mathbf{p}; G) \cap N(\mathbf{p}; H) & \text{otherwise.} \end{cases}$$

(A IV) *(Elimination) If $G, H \in \mathcal{G}$ and $\mathbf{p} \in \mathcal{D}$ then there exists a graph $F \in \mathcal{G}$ with $N(\mathbf{p}; F) = N(\mathbf{p}; G) \cup N(\mathbf{p}; H)$.*

(A V) (*Boundary*) *Let $S \subseteq [N]$ be fixed and $G$ be the bipartite graph with edges $E(G) = \{\mathbf{p}i \mid \mathbf{p} \in \mathcal{D}, i \in S\}$. Then $G \in \mathcal{G}$.*

(A VI) (*Comparability*) *For any point $\mathbf{p} \in [\mathcal{D}]$ the comparability graph $CG_{G,H}^{\mathbf{p}}$ of any two patterns $G, H \in \mathcal{G}$ is acyclic.*

For a pair $G, H$ of activation patterns, we consider the *comparability graph $CG_{GH}^{\mathbf{p}}$* of a data point $\mathbf{p} \in \mathcal{D}$. This graph has nodes $[N]$ and contains directed and undirected edges. We draw an edge between $j$ and $k$ if $j \in N(\mathbf{p}; G)$ and $k \in N(\mathbf{p}; H)$. This edge is undirected if $j, k \in N(\mathbf{p}, G) \cap N(\mathbf{p}, H)$ and $j \to k$ otherwise.

(A I),(A II),(A III), and (A IV) are analogs to axioms (C I)–(C IV) of covectors of oriented matroids. On the other hand, (A III), (A IV), (A V), and (A VI) are analogs to axioms (T I)–(T IV) of covectors of tropical oriented matroids, which we will define formally in Section 5.2. For tropical oriented matroids, the composition axiom is replaced by the surrounding axiom, which follows from stricter conditions on possible perturbations of tropical hyperplanes.

*Proof.* (A I) is realized by $\theta = (0, \dots, 0)$, or any point in the lineality space of $\Sigma_{\mathcal{D}}(N)$. (A II) holds since the construction of $\Sigma_{\mathcal{D}}(N)$ is symmetric in $i \in [N]$. (A V) will be proven separately in Theorem 5.17.

For (A III), let $\mathbf{x} \in C_{\mathcal{D}}(G)$ and $\mathbf{y} \in C_{\mathcal{D}}(H)$. These are vectors of parameters of the form $\mathbf{x} = (a_1^x, \mathbf{s}_1^x, \dots, a_N^x, \mathbf{s}_N^x)$ and $\mathbf{y} = (a_1^y, \mathbf{s}_1^y, \dots, a_N^y, \mathbf{s}_N^y)$. For $\varepsilon > 0$ small enough, consider $\mathbf{z} = \mathbf{x} + \varepsilon \mathbf{y}$. Then for any $\mathbf{p} \in \mathcal{D}$ such that $N(\mathbf{p}; G) \cap N(\mathbf{p}; H) = \emptyset$ holds

$$\underset{i \in [N]}{\arg\max}(a_i^z + \langle \mathbf{s}_i^z, \mathbf{p} \rangle) = \underset{i \in [N]}{\arg\max}(a_i^x + \langle \mathbf{s}_i^x, \mathbf{p} \rangle + \varepsilon(a_i^y + \langle \mathbf{s}_i^y, \mathbf{p} \rangle)) = \underset{i \in [N]}{\arg\max}(a_i^x + \langle \mathbf{s}_i^x, \mathbf{p} \rangle).$$

If $N(\mathbf{p}; G) \cap N(\mathbf{p}; H) \neq \emptyset$, then $\max_{i \in [N]}(a_i^z + \langle \mathbf{s}_i^z, \mathbf{p} \rangle) = (a_{i^*}^x + \langle \mathbf{s}_{i^*}^x, \mathbf{p} \rangle + \varepsilon(a_{i^*}^y + \langle \mathbf{s}_{i^*}^y, \mathbf{p} \rangle)$ if and only if both $a_{i^*}^x + \langle \mathbf{s}_{i^*}^x, \mathbf{p} \rangle = \max_{i \in [N]} a_i^x + \langle \mathbf{s}_i^x, \mathbf{p} \rangle$ and $a_{i^*}^y + \langle \mathbf{s}_{i^*}^y, \mathbf{p} \rangle = \max_{i \in [N]} a_i^y + \langle \mathbf{s}_i^y, \mathbf{p} \rangle$. Thus, $\mathbf{z} \in C_{\mathcal{D}}(G \circ H)$.

For (A IV), let again $\mathbf{x} \in C_{\mathcal{D}}(G)$ and $\mathbf{y} \in C_{\mathcal{D}}(H)$. There are fixed values $\mu_x(\mathbf{p}), \mu_y(\mathbf{p}) \in \mathbb{R}$ such that for every $i^* \in N(\mathbf{p}; G)$ and $j^* \in N(\mathbf{p}; G)$ holds

$$\max_{i \in [N]}(a_i^x + \langle \mathbf{s}_i^x, \mathbf{p} \rangle) = a_{i^*}^x + \langle \mathbf{s}_{i^*}^x, \mathbf{p} \rangle = \mu_x(\mathbf{p}), \quad \max_{i \in [N]}(a_i^y + \langle \mathbf{s}_i^y, \mathbf{p} \rangle) = a_{j^*}^y + \langle \mathbf{s}_{j^*}^y, \mathbf{p} \rangle = \mu_y(\mathbf{p}).$$

Choose $\mathbf{z} = (a_1^z + \mu^z(\mathbf{p}), \mathbf{s}_1^z, \dots, a_N^z + \mu^z(\mathbf{p}), \mathbf{s}_N^z)$ such that

$$a_i^z + \langle \mathbf{s}_i^z, \mathbf{p} \rangle + \mu^z(\mathbf{p}) = \max(a_i^x + \langle \mathbf{s}_i^x, \mathbf{p} \rangle + \mu^y(\mathbf{p}), a_i^y + \langle \mathbf{s}_i^y, \mathbf{p} \rangle + \mu^x(\mathbf{p})).$$

For $\mathbf{p}$ holds

$$\max_{i \in [N]}(a_i^z + \langle \mathbf{s}_i^z, \mathbf{p} \rangle + \mu^z(\mathbf{p})) = \max_{i \in [N]}\left(\max(a_i^x + \langle \mathbf{s}_i^x, \mathbf{p} \rangle + \mu^y(\mathbf{p}), a_i^y + \langle \mathbf{s}_i^y, \mathbf{p} \rangle + \mu^x(\mathbf{p})\right)$$

$$= \max\left(\max_{i \in [N]}(a_i^x + \langle \mathbf{s}_i^x, \mathbf{p} \rangle + \mu^y(\mathbf{p})), \max_{i \in [N]}(a_i^y + \langle \mathbf{s}_i^y, \mathbf{p} \rangle + \mu^x(\mathbf{p}))\right)$$

$$= \max(\mu^x(\mathbf{p}) + \mu^y(\mathbf{p}), \mu^y(\mathbf{p}) + \mu^x(\mathbf{p}))$$

and the maximizers are precisely the indices in $N(\mathbf{p}; G) \cup N(\mathbf{p}; H)$.

For (A VI), suppose that $i_0, i_1, \dots, i_n, i_{n+1} = i_0 \in [N]$ is a cycle in $CG_{G,H}^{\mathbf{p}}$. After contracting all undirected edges we can assume that all edges are directed. Let again $\mathbf{x} \in C_{\mathcal{D}}(G), \mathbf{y} \in C_{\mathcal{D}}(H)$. An edge from $i_k$ to $i_{k+1}$ indicates that we have $i_k \in N(\mathbf{p}; G)$ and $i_{k+1} \in N(\mathbf{p}; H)$. This gives

$$a_{i_{k+1}}^x + \langle \mathbf{s}_{i_{k+1}}^x, \mathbf{p} \rangle \leq a_{i_k}^x + \langle \mathbf{s}_{i_k}^x, \mathbf{p} \rangle, \text{ and } a_{i_k}^y + \langle \mathbf{s}_{i_k}^y, \mathbf{p} \rangle \leq a_{i_{k+1}}^y + \langle \mathbf{s}_{i_{k+1}}^y, \mathbf{p} \rangle.$$

and at least one of these two inequalities is strict, since the edge is directed. Together this implies

$$a_{i_{k+1}}^x - a_{i_{k+1}}^y + \langle \mathbf{s}_{i_{k+1}}^x - \mathbf{s}_{i_{k+1}}^y, \mathbf{p} \rangle < a_{i_k}^x - a_{i_k}^y + \langle \mathbf{s}_{i_k}^x - \mathbf{s}_{i_k}^y, \mathbf{p} \rangle$$

Adding these conditions up for all $i_0, \dots, i_n$ yields $0 < 0$. $\qquad\square$

Theorem 5.14 gives necessary conditions on the set of activation patterns which appear as labels of the activation cones. The following statement shows that not all bipartite graphs will appear as activation patterns, unless the data points are affinely independent and are thus at most $d + 1$ many.

**Theorem 5.15.** *The activation fan $\Sigma_{\mathcal{D}}(N)$ has at most $N^M$ maximal cones and $(2^N - 1)^M$ cones of arbitrary dimension, where $N$ is the number of monomials and $M = |\mathcal{D}|$ is the number of data points. This bound is attained if and only if the points are affinely independent.*

*Proof.* Recall from Corollary 5.12 that for any activation pattern $\deg(\mathbf{p}) \geq 1$ for all $\mathbf{p} \in \mathcal{D}$. The number of bipartite graphs on $M \sqcup N$ without isolated nodes is $M$ is $(2^N - 1)^M$, and is thus an upper bound for the number of cones of arbitrary dimension. For cones of maximal dimension, Corollary 5.12 implies that the activation patterns satisfy $\deg(\mathbf{p}) = 1$ for all $\mathbf{p} \in \mathcal{D}$, and the number of such bipartite graphs is $N^M$. It thus remains to show that this bound is attained if and only if the points are affinely independent. We first assume that the points are affinely dependent and show that there exists a graph which does not occur as an activation pattern. Let $\mathbf{p}_1, \ldots, \mathbf{p}_M$ be affinely dependent, i.e., there are distinct data points $\mathbf{p}_1, \ldots, \mathbf{p}_k$, $\mathbf{p}'_1, \ldots, \mathbf{p}'_l$, and $\mu_i, \mu_j > 0, i \in [k], j \in [l]$ such that there is an affine dependency

$$\sum_{i=1}^{k} \mu_i \mathbf{p}_i = \sum_{j=1}^{l} \mu_j \mathbf{p}'_j, \qquad \sum_{i=1}^{k} \mu_i = \sum_{j=1}^{l} \mu_j.$$

Let $\lambda_i = \frac{\mu_i}{\sum_{i=1}^{k} \mu_i}, \lambda_j = \frac{\mu_j}{\sum_{j=1}^{l} \mu_j}$. Then this gives the point

$$\mathbf{q} = \sum_{i=1}^{k} \lambda_i \mathbf{p}_i = \sum_{j=1}^{l} \lambda_j \mathbf{p}'_j \in \mathrm{conv}(\mathbf{p}_1, \ldots, \mathbf{p}_k) \cap \mathrm{conv}(\mathbf{p}'_1, \ldots, \mathbf{p}'_l),$$

and $\mathbf{q}$ lies in the relative interiors of both convex hulls. Therefore, Theorem 5.13 (ii) implies that such a bipartite graph $G$ which has $N(i; G) = \{\mathbf{p}_1, \ldots, \mathbf{p}_k\}$ and $N(j; G) = \{\mathbf{p}'_1, \ldots, \mathbf{p}'_l\}$ is not an activation pattern of $\Sigma_{\mathcal{D}}(N)$. Conversely, let $\mathbf{p}_1, \ldots, \mathbf{p}_M$ be affinely independent, and let $G$ be any bipartite graph such that $N(\mathbf{p}; G) \neq \emptyset$ for all $\mathbf{p} \in \mathcal{D}$. Note that since the points are affinely independent we have $M \leq d + 1$. This implies that for any $i \in [N]$ there exists an affine linear functional $f_i(\mathbf{x}) = c_i + \langle \mathbf{u}_i, \mathbf{x} \rangle$ such that $f_i(\mathbf{p}) = 0$ for all $\mathbf{p} \in N(i; G)$ and $f_i(\mathbf{p}) < 0$ for all $\mathcal{D} \setminus N(i; G)$. Choose $a_i = c_i$ and $\mathbf{s}_i = \mathbf{u}_i$ for all $i \in [N]$ such that $N(i; G) \neq \emptyset$, and otherwise choose $a_i, \mathbf{s}_i$ small enough. Then $\theta = (a_1, \mathbf{s}_1, \ldots, a_N, \mathbf{s}_N)$ is a vector of parameters whose activation pattern is $G$. $\square$

**Remark 5.16.** The number of maximal cones in the activation fan $\Sigma_{\mathcal{D}}(N)$ equals the number of vertices of the activation polytope $P_{\mathcal{D}}(N)$. Montúfar et al. (2022) give a sharp upper bound for the number of vertices of generic Minkowski sums. However, the Minkowski summands $\Delta(\mathbf{p})$ are not entirely generic in our case.

## 5.2 Linear and Tropical Classification within the Activation Fan

In Section 5.1 we have seen how activation patterns can be thought of as multivalued analogs of covectors of oriented matroids. Specifically, Remark 5.8 describes how $N = 2$ recovers the linear case. In this section we describe how also for $N > 2$ every activation fan carries a family of hyperplane arrangements (and thus oriented matroids) with it, by considering intersections with affine spaces. We extend our findings to tropical oriented matroids, which are analogs of oriented matroids arising through arrangements of tropical hyperplanes.

Recall from the case of linear classification, that the dichotomies are given by a hyperplane arrangement $\mathcal{H}_{\mathcal{D}}$ in parameter space, the normal fan of the zonotope $P_{\mathcal{D}}$.

**Theorem 5.17.** *For any $S \subseteq [N]$ the bipartite graph with edges $E(G) = \{\mathbf{p}i \mid \mathbf{p} \in \mathcal{D}, i \in S\}$ is an activation pattern of a cone in $\Sigma_{\mathcal{D}}(N)$. The cone $C_{\mathcal{D}}(G)$ is the normal cone of a face $F$ of the activation polytope $P_{\mathcal{D}}(N)$, where $F$ is itself an activation polytope $P_{\mathcal{D}}(|S|)$. In particular, ranging over all sets with $|S| = 2$, one gets $\binom{N}{2}$ many faces of the activation polytope $P_{\mathcal{D}}(N)$ which are equal to the zonotope $P_{\mathcal{D}}$.*

*Proof.* For notational convenience, we identify a $(d+1)N$-dimensional point $\theta \in \Theta(d, n, m)$ with a matrix $M$ of size $(d+1) \times N$, where the first column is identified with the first $(d+1)$ entries of the vector, and so on.

Fix $S \subseteq [N]$. Since $\mathcal{D}$ is a finite set of points, there exists a linear functional $l \in \mathbb{R}^{d+1}$ such that $\langle (1, \mathbf{p}), l \rangle > 0$ on all $\mathbf{p} \in \mathcal{D}$. We define the $i^{\text{th}}$ column of the matrix $M^S$ as

$$M_i^S = \begin{cases} l & \text{if } i \in S, \\ \mathbf{0} & \text{if } i \in [N] \setminus S. \end{cases}$$

Since $M^S$ corresponds to a vector of parameters of a tropical signomial $f$, it has an associated activation pattern $G$, which we now describe. Recall that $N(i; G) = \{\mathbf{p} \in \mathcal{D} \mid \mathbf{p}$ activate the $i^{\text{th}}$ term$\}$. For $\mathbf{p} \in \mathcal{D}$ we have $f(\mathbf{p}) = \langle (1\mathbf{p}), l \rangle > 0$, and precisely the terms $i \in S$ are active. Thus, $E(G) = \{\mathbf{p}i \mid \mathbf{p} \in \mathcal{D}, i \in S\}$, and this is the activation pattern of the cone $C_{\mathcal{D}}(G)$ which contains $M^S$ in its relative interior.

We now describe the face $F^S$ of the activation polytope whose normal cone is $C_{\mathcal{D}}(G)$. By Proposition 5.11, we have that $F^S = \sum_{\mathbf{p} \in \mathcal{D}} F(\mathbf{p})$, where the face $F(\mathbf{p})$ of $\Delta(\mathbf{p})$ is the $(|S-1|)$-dimensional simplex $\text{conv}(M^i \mid i \in S)$. The polytope $F^S$ lies inside an affine space of codimension $(|S|-1)(\dim(\text{aff}(\mathcal{D})+1)$. Inside this affine space, we have $F^S = P_{\mathcal{D}}(|S|)$, which is embedded into $\Theta(d, N)$ by inserting $\mathbf{0}$ for all columns indexed by $[N] \setminus S$, when we view the vertices of $P_{\mathcal{D}}(|S|)$ as matrices of size $(d+1) \times |S|$ and the vertices of $F^S$ as matrices of size $(d+1) \times N$. $\qquad\square$

We dualize the above statement to find the hyperplane arrangement $\mathcal{H}_{\mathcal{D}}$ in the activation fan $\Sigma_{\mathcal{D}}(N)$. The $k$-dimensional face $F_{ij}$ identified in the former proof is dual to a $(N(d+1)-k)$-dimensional cone $C_{ij} \in \Sigma_{\mathcal{D}}(N)$. More specifically, $C_{ij} = C_{\mathcal{D}}(G)$ where $G$ is the activation pattern such that $N(\mathbf{p}, G) = \{i, j\}$ for all $\mathbf{p} \in \mathcal{D}$. The $k'$-dimensional faces of $F_{ij}$ are dual to $(N(d+1)-k')$-dimensional cones of $\Sigma_{\mathcal{D}}(N)$ which contain $C_{ij}$. The collection of these cones is called the *star* of $C_{ij}$, and denoted by $\text{star}(C_{ij})$. Let $\text{lin}(C_{ij})$ denote the smallest linear space containing $C_{ij}$. Projecting the cones in the star of $C_{ij}$ onto the orthogonal space $\text{lin}(C_{ij})^\perp$ yields the normal fan of the zonotope $F_{ij}$. We thus obtain the following dual statement:

**Theorem 5.18.** *For each $i, j \in [N]$, let $C_{ij} = C_{\mathcal{D}}(G)$ be the activation cone for the pattern $G$ with $N(\mathbf{p}; G) = \{i, j\}$ for all $\mathbf{p} \in \mathcal{D}$. Then the projection of $\text{star}(C_{ij})$ onto $\text{lin}(C_{ij})^\perp$ is the polyhedral fan induced by the hyperplane arrangement $\mathcal{H}_{\mathcal{D}}$.*

We now move towards tropical hyperplane arrangements and tropical oriented matroids. Tropical oriented matroids can be viewed as multivalued analogs of oriented matroids, which arise through tropical hyperplane arrangements. We will see that activation patterns can also be viewed as generalization of realizable tropical oriented matroids.

A *tropical hyperplane* $H(-\mathbf{a})$ is the tropical hypersurface $\mathcal{T}(f)$ defined by the linear tropical polynomial $f_{\mathbf{a}}(\mathbf{x}) = \bigoplus_{i \in [d]} a_i \odot \mathbf{x}_i$. It is an affine translate of the normal fan of a standard simplex with *apex* $-\mathbf{a} = -(a_1, \ldots, a_d) \in \mathbb{R}^d$. Thus, any tropical hyperplane subdivides the ambient space into $d$ maximal regions, one for each term of $f_{\mathbf{a}}$. Given a finite number of tropical hyperplanes $H(\mathbf{a}^1), \ldots, H(\mathbf{a}^M)$ we consider the common refinement of all these subdivisions, and label each region $\mathcal{R}$ by a *tropical covector* $(A_1, \ldots, A_M)$ where $A_i \subseteq [d]$ is the set of indices of the terms of $f_{\mathbf{a}^i}$ which are active on $\mathcal{R}$. We obtain an activation pattern $G = ([d] \sqcup \mathcal{D}, E(G))$ from a tropical covector by setting $N(\mathbf{p}_i; G) = A_i$ for $\mathbf{p}_i \in \mathcal{D} = \{\mathbf{p}_1, \ldots, \mathbf{p}_M\}$. Any set of tropical covectors which arise through such an arrangement of tropical hyperplanes is called a *realizable* tropical oriented matroid.

Similarly to the linear case, tropical oriented matroids are defined through a set of axioms, and it was shown that sets satisfying these axioms are in bijection with subdivisions of products of simplices (Horn, 2016). A set $T \subseteq \{(A_1, \ldots, A_M) \mid A_i \subseteq [N], i \in [M]\}$ is the set of tropical covectors of a *tropical oriented matroid* if it satisfies the following axioms (Ardila & Develin, 2008, Def. 3.5).

(T I) Boundary: For each $j \in [N]$ holds $(\{j\}, \ldots, \{j\}) \in T$.

(T II) Elimination: If $A, B \in T$ and $j \in [M]$ then there exists a type $C \in T$ with $C_j = A_j \cup B_j$ and $C_k \in \{A_k, B_k, A_k \cup B_k\}$ for all $k \in [M]$.

(T III) Comparability: The comparability graph $CG_{A,B}$ of any two types $A$ and $B$ in $T$ is acyclic.

(T IV) Surrounding: If $A \in T$ the any refinement is also in $T$.

Let $A, B$ be tropical covectors and $G, H$ be the corresponding activation patterns. The comparability graph of the tropical covectors is $CG_{A,B} = \bigcup_{\mathbf{p} \in \mathcal{D}} C_{G,H}^{\mathbf{P}}$, where the comparability graph $C_{G,H}^{\mathbf{P}}$ is as defined in Theorem 5.14. The refinement of a type $A = (A_1, \ldots, A_M)$ with respect to an ordered partition $(P_1, \ldots, P_r)$ of $[M]$ is $A = (A_1 \cap P_{m(1)}, \ldots, A_M \cap P_{m(M)})$, where $m(i)$ is the largest index for which $A_i \cap P_{m(i)} \neq \emptyset$. Note that (A III),(A V),(A IV) and (A VI) of Theorem 5.14 are generalizations of axioms of tropical oriented matroids.

Similarly to Theorem 5.18 we can recover realizable tropical oriented matroids by intersection with an affine subspace.

**Theorem 5.19.** *Let $N = d$ and consider the $d$-dimensional affine space*

$$\mathcal{A} = \{(a_1, \mathbf{s}_1, \ldots, a_d, \mathbf{s}_d) \in \Theta(d, d) \mid \mathbf{s}_i = \mathbf{e}_i \text{ for } i \in [d]\}.$$

*Then $\Sigma_{\mathcal{D}}(d) \cap \mathcal{A}$ is the collection of cells in the tropical hyperplane arrangement $\bigcup_{\mathbf{p} \in \mathcal{D}} H(-\mathbf{p})$.*

*Proof.* Let $C_{\mathcal{D}}(G) \in \Sigma_{\mathcal{D}}(d)$ such that $C_{\mathcal{D}}(G) \cap \mathcal{A} \neq \emptyset$, and consider the coordinate projection $\pi(C_{\mathcal{D}}(G))$ onto coordinates $(a_1, \ldots, a_d)$. Any parameter $\theta \in \mathcal{A}$ defines a function $f_\theta(\mathbf{x}) = \bigoplus_{i \in [d]} a_i \odot \mathbf{x}_i$. By construction, $\theta \in C_{\mathcal{D}}(G) \cap \mathcal{A}$ if and only if $\mathbf{a} = \pi(\theta)$ satisfies $\max_{i \in [d]} a_i + \mathbf{p}_i = a_{i^*} + \mathbf{p}_{i^*}$ for all edges $i^*\mathbf{p} \in E(G)$. In other words, $\mathbf{a}$ is contained in the region $i^*$ of $H(-\mathbf{p})$ for all such edges, and $\pi(C_{\mathcal{D}}(G))$ is a region in the tropical hyperplane arrangement. $\square$

We can extend the above statement to more parameters $N > d$ by requiring that the additional monomials $\left(\begin{smallmatrix} a_i \\ \mathbf{s}_i \end{smallmatrix}\right), i > d$ lie below the lifted polytope $\text{conv}(\left(\begin{smallmatrix} a_1, \\ \mathbf{e}_1 \end{smallmatrix}\right), \ldots, \left(\begin{smallmatrix} a_d \\ \mathbf{e}_d \end{smallmatrix}\right))$. In this case they are not visible in any regular subdivision of the dual Newton polytope, and can thus be neglected in the above proof.

The dual statement to Theorem 5.19 identifies a subcomplex of the boundary of the activation polytope $P_{\mathcal{D}}(d)$. This subcomplex corresponds to a regular subdivision of products of simplices, and coincides with the boundary complex of the unbounded polyhedron in Develin & Sturmfels (2004, Lemma 10), whose bounded cells determine the tropical convex hull of the points in $\mathcal{D}$.

# 6 Classification Fan

In Section 4 we have seen how we can understand tropical rational functions $g \oslash h$ via tropical signomials $f$ by separating the $N$ terms of $f$ into $n$ positive and $m$ negative terms. This defines $g$ and $h$ respectively, such that $f = g \oplus h$ and $N = n + m$. This procedure subdivides the regions defined by $\mathcal{T}(f)$ into positive and negative regions. In terms of the activation patterns, this amounts to a coloring of the nodes $[N]$ as positive or negative nodes. In this section, we consider the classification fan, the fan which arises through the coloring of the nodes. In analogy to the linear case, we will consider the level sets and sublevel sets of the 0/1-loss function.

## 6.1 Dividing the Parameter Space

Let $\mathcal{D} \subseteq \mathbb{R}^d$ be a finite set of points, $N \in \mathbb{N}$ and fix $n, m \in \mathbb{N}_{\geq 1}$ such that $n + m = N$. We split the parameter space into *parameters of the numerator $g$*, which we denote $a_i, \mathbf{s}_i$ for $i \in [n]$ and *parameters of the denominator $h$*, which are given by $b_j = a_{n+j}, \mathbf{t}_j = \mathbf{s}_{n+j}$ for $j \in [m]$. Given a vector of parameters $\theta = (a_1, \mathbf{s}_1, \ldots, a_n, \mathbf{s}_n, b_1, \mathbf{t}_1, \ldots, b_m, \mathbf{t}_m) \in \Theta(d, n, m)$, this defines a numerator $g_\theta(\mathbf{x}) = \max_{i \in [n]}(a_i + \langle \mathbf{s}_i, \mathbf{x} \rangle)$ and denominator $h_\theta(\mathbf{x}) = \max_{j \in [m]}(b_j + \langle \mathbf{t}_j, \mathbf{x} \rangle)$, such that $f_\theta = g_\theta \oplus h_\theta$. Given an activation pattern $G$ with nodes $V(G) = \mathcal{D} \sqcup [N]$, the split $n + m = N$ induces a coloring of the nodes $[N]$ into $[n] \sqcup [m]$.

**Definition 6.1** (Activation Pattern of Tropical Rational Function)**.** The *activation pattern* of $(g \oslash h, \mathcal{D})$ is the bipartite graph $G = (V(G), E(G))$ on nodes $V(G) = \mathcal{D} \sqcup ([n] \sqcup [m])$ with edges $E(G) = \{\mathbf{p}i \mid \mathbf{p}$ activates the $i^{\text{th}}$ term of $g\} \sqcup \{\mathbf{p}j \mid \mathbf{p}$ activates the $j^{\text{th}}$ term of $h\}$.

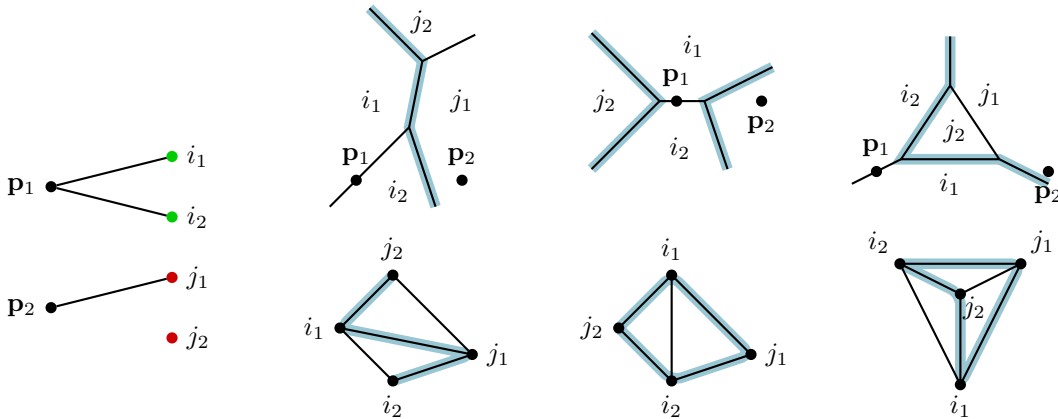

(a) Activation pattern.

(b) Decision boundary in input space (top) and dual sign-mixed Newton polytopes (bottom).

Figure 6: A coloring of the bipartite graph and the corresponding regions, as described in Example 6.2. The positive regions are labeled by $i_1, i_2$ and colored in green, the negative regions are labeled by $j_1, j_2$ and colored in red, and the decision boundary is highlighted in blue.

**Example 6.2** (From Signomials to Rational Functions). We continue with the activation pattern of the tropical signomials $f$ from Example 5.2. Partitioning $N = 4$ into $n = 2, m = 2$ yields

(i) a partition of the monomials of $f$ into two signomials $g(\mathbf{x}) = a_1 \odot \mathbf{x}^{\odot \mathbf{s}_1} \oplus a_s \odot \mathbf{x}^{\odot \mathbf{s}_2}$ and $h(\mathbf{x}) = b_1 \odot \mathbf{x}^{\odot \mathbf{t}_1} \oplus b_s \odot \mathbf{x}^{\odot \mathbf{t}_2}$,

(ii) a partition of the regions defined by $\mathcal{T}(f)$ into positive regions $i_1, i_2$ and negative regions $j_1, j_2$,

(iii) a partition of the vertices of the dual regular subdivision of $\mathrm{Newt}(f)$ into positive and negative vertices.

The positive and negative regions are separated by the decision boundary of $g \oslash h$. Figure 6 shows all of these partitions for the configuration from Example 5.2.

With this partition of parameters, the activation cone of $G$ is

$$C_{\mathcal{D}}(G) = \Big\{ \theta \mid \max_{i \in [n], j \in [m]} (a_i + \langle \mathbf{s}_i, \mathbf{p} \rangle, b_j + \langle \mathbf{t}_j, \mathbf{p} \rangle) = a_{i^*} + \langle \mathbf{s}_{i^*}, \mathbf{p} \rangle \text{ for all } \mathbf{p} i^* \in E(G) \text{ and }$$

$$\max_{i \in [n], j \in [m]} (a_i + \langle \mathbf{s}_i, \mathbf{p} \rangle, b_j + \langle \mathbf{t}_j, \mathbf{p} \rangle) = b_{j^*} + \langle \mathbf{t}_{j^*}, \mathbf{p} \rangle \text{ for all } \mathbf{p} j^* \in E(G) \Big\}$$

$$\subseteq \Theta(d, n, m).$$

**Lemma 6.3.** *Fix a partition $[N] = [n] \sqcup [m]$ and an activation pattern $G$ of tropical rational functions. Then for any $\theta \in C_{\mathcal{D}}(G)$ it holds that $(g_\theta \oslash h_\theta)(\mathbf{p}) \geq 0$ if and only if $\mathbf{p} i^* \in E(G)$ for some $i^* \in [n]$, and $(g_\theta \oslash h_\theta)(\mathbf{p}) \leq 0$ if and only if $\mathbf{p} j^* \in E(G)$ for some $j^* \in [m]$. The inequality is strict if and only if the set of neighbors of $\mathbf{p}$ in $G$ is monochromatic, i.e. $N(\mathbf{p}; G) \subseteq [n]$ or $N(\mathbf{p}; G) \subseteq [m]$.*

*Proof.* If $\max_{i \in [n], j \in [m]} (a_i + \langle \mathbf{s}_i, \mathbf{p} \rangle, b_j + \langle \mathbf{t}_j, p \rangle) = a_{i^*} + \langle \mathbf{s}_{i^*}, \mathbf{p} \rangle$ then $g_\theta(\mathbf{p}) \geq h_\theta(\mathbf{p})$ for all $i^* \in [n]$ such that $\mathbf{p} i^* \in E(G)$. This inequality is strict if and only if the maximum is not attained in any term of $h_\theta(\mathbf{p})$. Similarly, $g_\theta(\mathbf{p}) \leq h_\theta(\mathbf{p})$ for all $j^* \in [m]$ such that $\mathbf{p} j^* \in E(G)$. $\square$

**Definition 6.4** (Classification Fan). The *classification fan* $\Sigma_{\mathcal{D}}(n, m)$ of a finite data set $\mathcal{D} \subset \mathbb{R}^d$ and tropical rational functions with $n$ monomials in the numerator and $m$ monomials in the denominator is the set of all activation cones in parameter space $\Theta(d, n, m)$ of tropical rational functions.

We emphasize that the fan structures of the activation fan $\Sigma_\mathcal{D}(n+m)$ and classification fan $\Sigma_\mathcal{D}(n,m)$ coincide. The only difference is a partition of the coordinates in the underlying spaces. However, this distinction is crucial as we now want to distinguish between monomials of the numerator and monomials of the denominator.

## 6.2 The Space of Dichotomies

In this section, we subdivide the parameter space $\Theta(d,n,m)$ into sets, each corresponding to those parameters which classify the data according to a fixed target dichotomy. These (open) sets generalize the chambers of the hyperplane arrangement from Section 2. We will also describe indecision surfaces as analogs of the hyperplanes in the arrangement, which will turn out to be decision boundaries inside the parameter space.

**Definition 6.5** (Indecision Surface)**.** The *indecision surface* of a data point $\mathbf{p} \in \mathcal{D}$ is

$$\mathcal{S}(\mathbf{p}) = \{\theta \in \Theta(d,n,m) \mid (g_\theta \oslash h_\theta)(\mathbf{p}) = 0\}$$

in the parameter space of tropical rational functions.

We will see that the indecision surface plays the role of the hyperplane $(1,\mathbf{p})^\perp$ from the linear case (cf. Section 2) in this more general setup.

**Theorem 6.6.** *The indecision surface $\mathcal{S}(\mathbf{p})$ is a sign-mixed subfan of the normal fan of $\Delta(\mathbf{p})$ of dimension $(n+m)(d+1)-1$. It divides the parameter space $\Theta(d,n,m)$ into two open connected components $\mathcal{S}^+(\mathbf{p}) = \{\theta \in \Theta(d,n,m) \mid (g_\theta \oslash h_\theta)(\mathbf{p}) > 0\}$ and $\mathcal{S}^-(\mathbf{p}) = \{\theta \in \Theta(d,n,m) \mid (g_\theta \oslash h_\theta)(\mathbf{p}) < 0\}$.*

*Proof.* First note that the indecision surface is itself the decision boundary of the tropical rational function $\bigoplus_{i \in [n]} a_i \odot \mathbf{s}_i^{\odot \mathbf{p}} \oslash \bigoplus_{j \in [m]} b_j \odot \mathbf{t}_j^{\odot \mathbf{p}}$ in indeterminates $a_i, \mathbf{s}_i, b_j, \mathbf{t}_j$. The Newton polytope of this tropical rational function is the simplex $\Delta(\mathbf{p})$. We use the notation from the proof of Theorem 5.7, and fix a partition $[N] = [n] \sqcup [m]$ as explained in Section 6.1. Theorem 4.2 implies that $\mathcal{S}(\mathbf{p})$ is a polyhedral fan, whose maximal cones are dual to edges $\mathrm{conv}(\mathbf{v}_i, \mathbf{v}_j)$ where $i \in [n]$ is a monomial of $g$ and $j \in [m]$ is a monomial of $h$. The Euclidean closure of the region $\mathcal{S}^+(\mathbf{p})$ is the union of all normal cones of vertices $\mathbf{v}_i, i \in [n]$ and since $\Delta(\mathbf{p})$ is a simplex, this union is connected. Similarly, the closure of $\mathcal{S}^-(\mathbf{p})$ is the union of normal fans of $\mathbf{v}_j, j \in [m]$. $\square$

Fix a target dichotomy $C^* \in \{+,-\}^M$. We obtain the *space of solutions of a fixed dichotomy* as $\Theta_\mathcal{D}^{C^*}(d,n,m) = \bigcap_{k=1}^M \mathcal{S}^{C_k^*}(\mathbf{p}_k) = \{\theta \in \Theta(d,n,m) \mid \mathrm{sgn}\,(g_\theta(\mathbf{p}_k) \oslash h_\theta(\mathbf{p}_k)) = C_k^* \text{ for all } k \in [M]\}$.

**Corollary 6.7.** *The arrangement of indecision surfaces $\mathcal{S}(\mathbf{p})$ over all $\mathbf{p} \in \mathcal{D}$ divides the parameter space into open sets $\Theta_\mathcal{D}^{C^*}(d,n,m)$ over all dichotomies $C^* \subseteq \{+,-\}^M$.*

This arrangement can be viewed as the true analog of the hyperplane arrangement $\mathcal{H}_\mathcal{D}$ from the linear case, and $\Theta_\mathcal{D}^{C^*}(d,n,m)$ plays the role of an open chamber. In Section 6.3 we will fix a target dichotomy $C^*$ and study (the closure of) the space $\Theta_\mathcal{D}^{C^*}(d,n,m)$. We will see that this space is disconnected (cf. Theorem 6.11), but it allows a natural fan structure which is inherited from the classification fan $\Sigma_\mathcal{D}(n,m)$.

The subdivision into these different subfans relates to the growth function and Vapnik-Chervonenkis dimension (Vapnik & Chervonenkis, 1971). This is a fundamental concept in statistical learning theory which relates the complexity of a function class with the generalization ability of any learning algorithm based on that function class. Let $c(n,m,\mathcal{D}) = |\{C^* \mid \Theta_\mathcal{D}^{C^*}(d,n,m) \neq \emptyset\}|$ denote the number of dichotomies which can be attained by tropical rational functions with $n+m$ terms on the data $\mathcal{D}$. The *growth function* $\Pi_{n,m} \colon \mathbb{N} \to \mathbb{N}$ is given by $\Pi_{n,m}(M) = \max(c(\mathcal{D},n,m) \mid \mathcal{D} \subset \mathbb{R}^d, |\mathcal{D}| = M)$. The *VC-dimension* of tropical rational functions is $\max(M \mid \Pi_{n,m}(M) = 2^M)$. To compute the growth function, a common strategy is to count the number of non-empty regions $c(\mathcal{D},n,m)$ for a given data set $\mathcal{D}$ and then maximize this number over the choice of the data set having a fixed cardinality $M$. We refer the reader to Anthony & Bartlett (1999) for an overview of these techniques, and Bartlett et al. (2019) for recent bounds on the VC dimension of neural networks with piecewise linear activation functions.

### 6.3 Classifying Points Correctly

We now fix a target dichotomy $C^* \in \{+, -\}^M$ of the data. The dichotomy divides the data points into two sets $\mathcal{D}_+^{C^*} = \{\mathbf{p}_i \in \mathcal{D} \mid C_i^* = +\}$ and $\mathcal{D}_-^{C^*} = \mathcal{D} \setminus \mathcal{D}_+^{C^*}$. An activation pattern $G$ of $(g \oslash h, \mathcal{D})$ is *compatible* with $C^*$ if $N(\mathbf{p}; G) \subseteq [n]$ for all $\mathbf{p} \in \mathcal{D}_+^{C^*}$ and $N(\mathbf{p}; G) \subseteq [m]$ for all $\mathbf{p} \in \mathcal{D}_-^{C^*}$. An activation pattern is *weakly compatible* with $C^*$ if $N(\mathbf{p}; G) \cap [n] \neq \emptyset$ for all $\mathbf{p} \in \mathcal{D}_+^{C^*}$ and $N(\mathbf{p}; G) \cap [m] \neq \emptyset$ for all $\mathbf{p} \in \mathcal{D}_-^{C^*}$.

**Definition 6.8** (Perfect Classification Fan). The *perfect classification fan* $\Sigma_{\mathcal{D}}^0(n, m)$ is the subfan of the classification fan consisting of those closed cones whose activation patterns are weakly compatible with $C^*$. In other words, $\Sigma_{\mathcal{D}}^0(n, m)$ is the restriction of the classification fan onto the set $\overline{\Theta_{\mathcal{D}}^{C^*}(d, n, m)} = \{\theta \in \Theta(d, n, m) \mid g_\theta(\mathbf{p}) \geq h_\theta(\mathbf{p}) \ \forall \mathbf{p} \in \mathcal{D}_+^{C^*}, \ g_\theta(\mathbf{p}) \leq h_\theta(\mathbf{p}) \ \forall \mathbf{p} \in \mathcal{D}_-^{C^*}\}$.

The wisdom behind this notation will become clear in Section 6.4, where we consider level-sets of the 0/1-loss. We have chosen to consider activation cones $C_{\mathcal{D}}(G)$ as closed cones. As a consequence, we obtain lower-dimensional cones on the boundary of $\overline{\Theta_{\mathcal{D}}^{C^*}(d, n, m)}$, whose activation patterns are only weakly compatible with $C^*$. On the other hand, this certifies nice geometric properties of the perfect classification fan.

**Proposition 6.9.** *The perfect classification fan $\Sigma_{\mathcal{D}}^0(n, m)$ is pure, i.e. every inclusion-maximal cone is full-dimensional.*

*Proof.* Any lower-dimensional cone in $\Sigma_{\mathcal{D}}^0(n, m)$ is the intersection of two full-dimensional cones from the activation fan. Let $C_{\mathcal{D}}(G), C_{\mathcal{D}}(H)$ be two full-dimensional cones of the activation fan. We show that if $C_{\mathcal{D}}(G), C_{\mathcal{D}}(H) \notin \Sigma_{\mathcal{D}}^0(n, m)$ then $C_{\mathcal{D}}(G) \cap C_{\mathcal{D}}(H) \notin \Sigma_{\mathcal{D}}^0(n, m)$. Since $C_{\mathcal{D}}(G), C_{\mathcal{D}}(H)$ are maximal, Corollary 5.12 implies that $C_{\mathcal{D}}(G), C_{\mathcal{D}}(H) \in \Sigma_{\mathcal{D}}^0(n, m)$ if and only if $G, H$ are compatible with the target dichotomy $C^*$. Therefore, if $C_{\mathcal{D}}(G) \notin \Sigma_{\mathcal{D}}^0(n, m)$ then there exists some $\mathbf{p}_G^+ \in \mathcal{D}_+^{C^*}$ such that $N(\mathbf{p}_G^+; G) \subseteq [m]$ or $\mathbf{p}_G^- \in \mathcal{D}_-^{C^*}$ such that $N(\mathbf{p}_G^-; G) \subseteq [n]$. Similarly, $C_{\mathcal{D}}(H) \notin \Sigma_{\mathcal{D}}^0(n, m)$ implies that there exists some $\mathbf{p}_H^+ \in \mathcal{D}_+^{C^*}$ such that $N(\mathbf{p}_H^+; H) \subseteq [m]$ or $\mathbf{p}_H^- \in \mathcal{D}_-^{C^*}$ such that $N(\mathbf{p}_H^-; G) \subseteq [n]$. Let $L = \overline{G \cup H}$. Then $C_{\mathcal{D}}(G) \cap C_{\mathcal{D}}(H) = C_{\mathcal{D}}(L)$, and since $E(L) \supseteq E(G) \cup E(H)$ we have that $N(\mathbf{p}_G^+; L) \cap [m] \neq \emptyset$ or $N(\mathbf{p}_H^+; L) \cap [m] \neq \emptyset$ or $N(\mathbf{p}_G^-; L) \cap [n] \neq \emptyset$ or $N(\mathbf{p}_H^-; L) \cap [n] \neq \emptyset$, i.e. $L$ is not weakly compatible with $C^*$. $\square$

The perfect classification fan can be viewed as a subfan of the activation fan, and its full-dimensional cones are thus dual to vertices of the activation polytope. To describe these vertices, recall that $P_{\mathcal{D}}(n + m) = \sum_{k=1}^M \Delta(\mathbf{p}_k)$. Under the partition of the parameters, each simplex $\Delta(\mathbf{p}_i)$ has vertices $\mathbf{v}_i^k, i \in [n]$ and $\mathbf{v}_j^k, j \in [m]$, and, similarly to the construction in Section 4, we label each of these vertices with $\mathrm{sgn}(\mathbf{v}_i^k) = +$ for $i \in [n]$ and $\mathrm{sgn}(\mathbf{v}_j^k) = -$ for $j \in [m]$. Each vertex $\mathbf{v}$ of $P_{\mathcal{D}}(n + m)$ can be written uniquely as $\mathbf{v} = \sum_{k=1}^M \mathbf{v}_{l(k)}^k$ for indices $l(k) \in [n] \sqcup [m]$ and we equip $\mathbf{v}$ with the dichotomy $d(\mathbf{v}) = (\mathrm{sgn}(\mathbf{v}_{l(1)}^1), \ldots, \mathrm{sgn}(\mathbf{v}_{l(M)}^M))$.

**Theorem 6.10.** *A vertex $\mathbf{v}$ of $P_{\mathcal{D}}(n + m)$ is dual to a full-dimensional cone of $\Sigma_{\mathcal{D}}^0(n, m)$ if and only if $d(\mathbf{v}) = C^*$.*

*Proof.* A vertex $\mathbf{v}$ has $d(\mathbf{v}) = C^*$ if and only if $\mathbf{v}$ lies in the intersection of the normal cones of $\mathbf{v}_{l(k)}^k$ for all $k \in [M]$ and $\mathrm{sgn}(\mathbf{v}_{l(k)}^k) = C_k^*$. By the proof of Theorem 5.7, this is equivalent to $C_{\mathcal{D}}(G)$ being the normal cone of $\mathbf{v}$, where $G$ is compatible with $C^*$. $\square$

In the case of linear classifiers, the set of perfect solutions is the single polyhedral cone in the hyperplane arrangement $\mathcal{H}_{\mathcal{D}}$ which corresponds to the dichotomy $C^*$. In contrast, the perfect classification fan is an entire polyhedral fan. Clearly, as we consider closed polyhedral cones, they always intersect in the origin. We thus consider a stronger notion for polyhedral fans. A pure $d$-dimensional polyhedral fan $\Sigma$ is *strongly connected* if for all maximal cones $C, D$ there exists a sequence of maximal cones $C = C_0, C_1, \ldots, C_k, C_{k+1} = D$ such that $\dim(C_i \cap C_{i+1}) = d - 1$. Any fan decomposes into *strongly connected components*, i.e. inclusion-maximal subfans of $\Sigma$ which are strongly connected. The strongly connected components of the perfect classification fan are in bijection with the connected components of $\Theta_{\mathcal{D}}^{C^*}(d, n, m)$, which is the interior of the support of $\Sigma_{\mathcal{D}}^0(n, m)$.

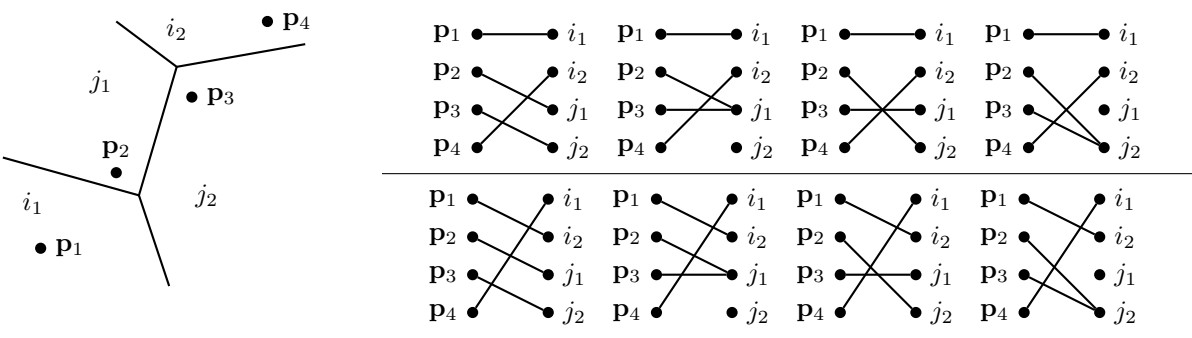

(a) The point configuration.  (b) Activation patterns of cones in the two connected components.

Figure 7: The 12 cones of correct classification from the example in the proof of Theorem 6.11, grouped in strongly connected components.

**Theorem 6.11.** *The perfect classification fan is not always strongly connected.*

*Proof.* Consider the data set $\mathcal{D} = \{\mathbf{p}_1, \mathbf{p}_2, \mathbf{p}_3, \mathbf{p}_4\}$ with $\mathbf{p}_1 = (0,0), \mathbf{p}_2 = (1,1), \mathbf{p}_3 = (2,2), \mathbf{p}_4 = (3,3)$, and the target dichotomy is $C^* = (+,-,-,+)$. The perfect classification fan $\Sigma_{\mathcal{D}}^0(2,2) \subset \Theta(2,2,2)$ consists of 8 cones of dimension 12 (the maximal dimension), which are divided into two strongly connected components: The intersection of any cone from the first connected component and any cone from the second component is solely contained in the lineality space of the fan. The activation patterns of the 8 cones are shown in Figure 7. In this example we can identify two fundamentally different types of cones, assigning $\mathbf{p}_2, \mathbf{p}_3$ either to the same negative index or to two separate negative indices. Then we have discrete symmetries of the parametrization via permutation of the positive indices $i_1, i_2$ and permutation of the negative indices $j_1, j_2$. The permutation of $j_1, j_2$ does not strongly disconnect the set, but the permutation of $i_1, i_2$ does. $\qquad\square$

In Theorem 5.15 we have seen that the trivial upper bound for the number of maximal cones in the activation fan is attained if the data points are affinely independent. We now obtain an analogous result for the classification fan, however, the following statement does not require the entire data set to be affinely independent.

**Theorem 6.12.** *Let $n, m \geq 2$ and fix a target dichotomy $C^*$. Then the perfect classification fan $\Sigma_{\mathcal{D}}^0(n,m)$ has at most $n^{|\mathcal{D}_+^{C^*}|} m^{|\mathcal{D}_-^{C^*}|}$ cones of maximal dimension. This bound is attained if and only if $\mathcal{D}_+^{C^*}, \mathcal{D}_-^{C^*}$ are linearly separable and both $\mathcal{D}_+^{C^*}$ and $\mathcal{D}_-^{C^*}$ are affinely independent sets.*

*Proof.* Since $\Sigma_{\mathcal{D}}^0(n,m)$ is a subfan of the activation fan $\Sigma_{\mathcal{D}}(n+m)$, Corollary 5.12 implies that the cones of maximal dimension are labeled by activation patterns $G$ such that $N(G; \mathbf{p}) = \{i(\mathbf{p})\} \subset [n]$ for $\mathbf{p} \in \mathcal{D}_+^{C^*}$ and $N(G, \mathbf{p}) = \{j(\mathbf{p})\} \subset [m]$ for $\mathbf{p} \in \mathcal{D}_-^{C^*}$. Thus, the number of maximal cones is at most $n^{|\mathcal{D}_+^{C^*}|} m^{|\mathcal{D}_-^{C^*}|}$. Any affine dependency within $\mathcal{D}_+^{C^*}$ or $\mathcal{D}_-^{C^*}$ forbids at least one such graph. This can be proven by following the argument in the proof of Theorem 5.15, namely that if $\mathcal{D}_+^{C^*}, \mathcal{D}_-^{C^*}$ are not linearly separable, then their convex hulls intersect, and Theorem 5.13 (i) implies the existence of a forbidden graph. Thus, the bound is not attained. We now show that the bound can be attained for affinely independent sets $\mathcal{D}_+^{C^*}$ and $\mathcal{D}_-^{C^*}$, which are linearly separable. Let $G$ be a bipartite graph in which every point in $\mathcal{D}$ is incident to exactly one edge, connecting every point in $\mathcal{D}_+^{C^*}$ with a node in $[n]$ and every point in $\mathcal{D}_-^{C^*}$ with a node in $[m]$. Let $l(\mathbf{x}) = c + \langle \mathbf{u}, \mathbf{x} \rangle$ be the equation defining a separating hyperplane, i.e. $l(\mathbf{p}^+) > 0$ and $l(\mathbf{p}^-) < 0$ for all $\mathbf{p}^+ \in \mathcal{D}_+^{C^*}, \mathbf{p}^- \in \mathcal{D}_-^{C^*}$. Since the points in $\mathcal{D}_+^{C^*}$ are affinely independent, there exists an affine function $f_i(\mathbf{x}) = d_i + \langle \mathbf{v}_i, \mathbf{x} \rangle$ for every $i \in [n]$, such that $f_i(\mathbf{p}^+) = 0$ for all $\mathbf{p}^+ \in N(i; G)$ and $f_i(\mathbf{p}^+) < 0$ for all $\mathbf{p}^+ \in \mathcal{D}_+^{C^*} \setminus N(i; G)$. Similarly, there exists an affine function $f_j(\mathbf{x}) = e_j + \langle \mathbf{w}_j, \mathbf{x} \rangle$ for every $j \in [m]$ such that $f_j(\mathbf{p}^-) = 0$ for all $\mathbf{p}^- \in N(j; G)$ and $f_j(\mathbf{p}^-) < 0$ for all $\mathbf{p}^- \in \mathcal{D}_-^{C^*} \setminus N(j; G)$. Since $l(\mathbf{p}^-) < 0$ for all $\mathbf{p}^- \in \mathcal{D}_-^{C^*}$ we can choose a scalar $\lambda > 0$ such that for all $\mathbf{p}^- \in \mathcal{D}_-^{C^*}$ holds $-\lambda l(\mathbf{p}^-) > \max_{i \in [n]} f_i(\mathbf{p}^-)$. Similarly, since $l(\mathbf{p}^+) > 0$ for all $\mathbf{p}^+ \in \mathcal{D}_+^{C^*}$ we can choose $\mu > 0$ such that for all $\mathbf{p}^+ \in \mathcal{D}_+^{C^*}$ holds $\mu l(\mathbf{p}^+) > \max_{j \in [m]} f_j(\mathbf{p}^+)$.

For $i^* \in [n], \mathbf{p}^+ \in N(i^*; G)$ and $j^* \in [m], \mathbf{p}^- \in N(j^*; G)$ we obtain

$$\max_{i \in [n]}(\lambda l(\mathbf{p}^+) + f_i(\mathbf{p}^+)) = \quad \lambda l(\mathbf{p}^+) + f_{i^*}(\mathbf{p}^+) = \lambda l(\mathbf{p}^+) > 0$$

$$\max_{j \in [m]}(-\mu l(\mathbf{p}^+) + f_j(\mathbf{p}^+)) = -\mu l(\mathbf{p}^+) + \max_{j \in [m]}(f_j(\mathbf{p}^+)) < 0$$

$$\max_{i \in [n]}(\lambda l(\mathbf{p}^-) + f_i(\mathbf{p}^-)) = \quad \lambda l(\mathbf{p}^-) + \max_{i \in [n]}(f_i(\mathbf{p}^+)) < 0$$

$$\max_{j \in [m]}(-\mu l(\mathbf{p}^-) + f_j(\mathbf{p}^-)) = -\mu l(\mathbf{p}^-) + f_{j^*}(\mathbf{p}^+) = -\mu l(\mathbf{p}^-) > 0$$

We can thus choose $a_i = \lambda c + d_i$, $\mathbf{s}_i = \lambda u + \mathbf{v}_i$, $b_j = -\mu c + e_j$, $\mathbf{t}_j = -\mu u + \mathbf{w}_j$. This defines a vector of parameters with activation pattern $G$. $\qquad\square$

We close this section on perfect classification with an observation concerning tropical semialgebraic sets.

**Remark 6.13** ($\Sigma_{\mathcal{D}}^0(n,m)$ is a tropical semialgebraic set)**.** The set $\Sigma_{\mathcal{D}}^0(n,m)$ is the set of solutions to a set of tropical polynomial inequalities. Concretely, it is given by the tropical polynomial inequalities $\bigoplus_{i \in [n]} a_i \odot \mathbf{s}_i^{\odot \mathbf{P}} \geq \bigoplus_{j \in [m]} b_j \odot \mathbf{t}_j^{\odot \mathbf{P}}$ for all $\mathbf{p} \in \mathcal{D}_+^{C^*}$ and $\bigoplus_{i \in [n]} a_i \odot \mathbf{s}_i^{\odot \mathbf{P}} \leq \bigoplus_{j \in [m]} b_j \odot \mathbf{t}_j^{\odot \mathbf{P}}$ for all $\mathbf{p} \in \mathcal{D}_-^{C^*}$. Tropicalizations of semialgebraic sets have close relations to positive tropicalizations (cf. Section 4) and their systematic study has been established in the work of Jell et al. (2020).

## 6.4 Sublevel Sets of the 0/1-Loss Function

Recall that the 0/1-loss function counts the number of mistakes of any function $f$ with respect to the target dichotomy $C^*$, i.e. $\mathrm{err}_{C^*}(\theta) = |\{i \in [M] \mid \mathrm{sgn}(f_\theta(\mathbf{p}_i)) = -C_i^*\}|$. If $f_\theta = g_\theta \oslash h_\theta$ is a tropical rational function contained in a cone $C_{\mathcal{D}}(G)$, then $\mathrm{err}_{C^*}$ is constant along $C_{\mathcal{D}}(G)$. We thus extend the function $\mathrm{err}_{C^*}$ to cones and activation patterns. If $C_{\mathcal{D}}(G)$ is maximal, then $\mathrm{err}_{C^*}(G)$ counts the number of edges in $G$ between $\mathcal{D}_+^{C^*}$ and $[m]$, and between $\mathcal{D}_-^{C^*}$ and $[n]$.

**Definition 6.14.** The $k^{th}$ *level set* is the polyhedral fan $\Sigma_{\mathcal{D}}^k(n,m) = \bigcup C_{\mathcal{D}}(G)$, where the union runs over all activation patterns $G$ such that $\mathrm{err}_{C^*}(G) = k$. We define the $k^{th}$ *sublevel set* as $\Sigma_{\mathcal{D}}^{\leq k}(n,m) = \bigcup_{k'=0}^{k} \Sigma_{\mathcal{D}}^{k'}(n,m)$.

**Proposition 6.15.** *If $n = m$, then for any level set it holds that $\Sigma_{\mathcal{D}}^k(n,n) \cong \Sigma_{\mathcal{D}}^{|D|-k}(n,n)$.*

*Proof.* Let $\theta \in \Sigma_{\mathcal{D}}^k(n,n)$ and define $\theta'$ through parameters

$$a_i^{\theta'} = b_i^\theta, \ \mathbf{s}_i^{\theta'} = \mathbf{t}_i^\theta, \ b_i^{\theta'} = a_i^\theta, \ \mathbf{t}_i^{\theta'} = \mathbf{s}_i^\theta \text{ for } i \in [n].$$

Note that the corresponding tropical rational functions satisfy $f_\theta(\mathbf{p}) = -f_{\theta'}(\mathbf{p})$. Thus, $f_\theta$ makes a mistake at $\mathbf{p}$ if and only if $f_{\theta'}$ classifies $\mathbf{p}$ correctly, and vice versa. $\qquad\square$

Theorem 6.10 implies that the level sets $\Sigma_{\mathcal{D}}^0(n,m), \Sigma_{\mathcal{D}}^1(n,m), \dots, \Sigma_{\mathcal{D}}^{n+m}(n,m)$ induce a partition of the vertices of the activation polytope $P_{\mathcal{D}}(n+m)$, and counting the number of vertices in each part we obtain a sequence of natural numbers. In Algebraic Combinatorics one likes to describe sequences which are symmetric, unimodal or log-concave. Proposition 6.15 allows us to obtain a statement in this spirit:

**Corollary 6.16.** *Let $l_k$ denote the number of maximal cones in the $k^{th}$ level set $\Sigma_{\mathcal{D}}^k(n,n)$. Then $l_k = l_{|D|-k}$, i.e. the sizes of level sets are symmetric.*

Recall that Proposition 2.4 shows that in the case of linear classifiers, a cone of $k$ mistakes is always (strongly) connected to a cone of fewer mistakes. We now observe that the analog does not necessarily hold in the more general setup of tropical rational functions.

**Theorem 6.17.** *The sublevel sets $\Sigma_{\mathcal{D}}^{\leq k}(n,m)$ are not always strongly connected, even if the data points are in general position. More specifically, let $C \subset \Sigma_{\mathcal{D}}^k(n,m)$ be a strongly connected component. It may happen that all full-dimensional neighbors $D \in \Sigma_{\mathcal{D}}(n,m)$ of $C$ (which are adjacent through codimension 1) satisfy $\mathrm{err}_{C^*}(D) > k$.*

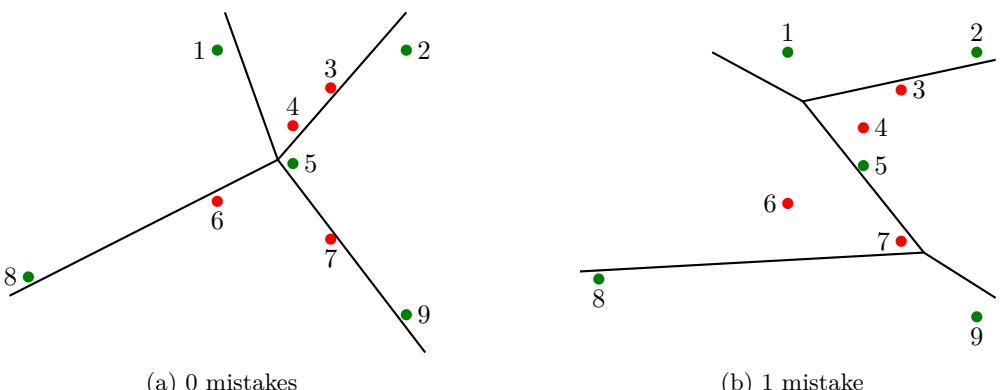

(a) 0 mistakes             (b) 1 mistake

Figure 8: The point configuration from Theorem 6.17 and two tropical hypersurfaces classifying the points perfectly (left) and according to $G$ (right).

*Proof.* To be precise, here we consider the standard notion of general position: $M$ points in $\mathbb{R}^d$ are in *general position* if no $k$ of them, for $k \leq d+1$, are contained in an affine subspace of dimension strictly less than $k-1$. We give an example of a configuration of points in general position which can be classified perfectly, for which however there exists a cone $C_{\mathcal{D}}(G)$ with $\text{err}_{C^*}(C_{\mathcal{D}}(G)) = 1$ from which there exist no (weakly) decreasing path to any cone with 0 mistakes. Let $\mathcal{D}_+^{C^*} = \{\mathbf{p}_1 = (-2, 3), \mathbf{p}_2 = (3, 3), \mathbf{p}_5 = (0, 0), \mathbf{p}_8 = (-7, -3), \mathbf{p}_9 = (3, -4)\}$ and $\mathcal{D}_-^{C^*} = \{\mathbf{p}_3 = (1, 2), \mathbf{p}_4 = (0, 1), \mathbf{p}_6 = (-2, -1), \mathbf{p}_7 = (1, -2)\}$, as shown in Figure 8. We computed the 12-dimensional fan $\Sigma_{\mathcal{D}}^{\leq 2}(2, 2)$ using the software `SageMath` (Sage Developers, 2021). By definition, we have $\Sigma_{\mathcal{D}}^{\leq 2}(2, 2) = \Sigma_{\mathcal{D}}^0(2, 2) \cup \Sigma_{\mathcal{D}}^1(2, 2) \cup \Sigma_{\mathcal{D}}^2(2, 2)$. The perfect classification fan $\Sigma_{\mathcal{D}}^0(2, 2)$ consists of 16 maximal cones, which divide into 8 strongly connected components, and the first level set consists of 304 cones, which divide into 28 components. The connected component containing $C_{\mathcal{D}}(G)$ consists of 20 maximal cones, but none of them is adjacent to any of the 16 cones of $\Sigma_{\mathcal{D}}^0(2, 2)$ through codimension 1. More precisely, the connected component which contains $C_{\mathcal{D}}(G)$ intersects $\Sigma_{\mathcal{D}}^1(2, 2)$ in dimension 3, which is the dimension of the lineality space of $\Sigma_{\mathcal{D}}(2, 2)$. In this sense, the connected component containing $C_{\mathcal{D}}(G)$ intersects the perfect classification fan $\Sigma^0(2, 2)$ trivially. $\qquad\square$

**Corollary 6.18.** *The 0/1-loss function has local non-global minima.*

## 7 Conclusion

We discussed binary classification by signs of parametric piecewise linear functions from the perspective of real tropical geometry. In this context, we described sets of ReLU networks as semialgebraic sets within sets of tropical rational functions with a fixed number of monomials in the numerator and denominator. We highlighted on the one hand the subdivision of the parameter space by the combinatorial type of the represented functions, specifically the combinatorial type of the decision boundary, and on the other hand the subdivision of the parameter space by the different activation patterns that are recorded on a given dataset. These two subdivisions are complementary to each other since a priori the decision boundary is independent of the particular data under consideration. In future work it will be interesting to study how the two interact; concretely, for a particular input dataset one might study the types of decision boundaries that permit certain types of classifications of that particular input dataset. The discussion of activation patterns gives rise to a generalization of oriented matroids. We showed how this can be used to obtain results on the structure of the loss landscape.

We hope that this work contributes to the further enhancement of the synergies between research on neural networks and polyhedral geometry, particularly as approached in real tropical geometry. As we observed, in some cases the same concepts appear in different communities under different names and leveraging results and perspectives from each side can facilitate advances and motivate new interesting future research. We hope that the presented discussion might serve as a springboard for further explorations including: the containment of neural networks with piecewise linear activations within simple parametrizations of tropical

rational functions; the structure of the subdivisions of the parameter space by the different properties of the represented functions, including the combinatorial type of the decision boundary, as well as the interface with other subdivisions arising from a particular dataset, including complexity measures and the combinatorial structure of the loss landscape.

**Acknowledgments**

We thank Hanna Tseran for various valuable discussions. We are grateful to Günter Rote for clarifying the proof of the composition property for Theorem 5.14, Arnau Padrol for pointing us to relevant literature on $k$-sets and $k$-facets, and Stefano Mereta for valuable input on the formulation of Theorem 3.2. We thank anonymous TMLR reviewers for constructive feedback. This project has been supported by DFG grant 464109215 "Combinatorial and implicit approaches to deep learning" within the priority programme SPP 2298 "Theoretical Foundations of Deep Learning". GM has also been supported by NSF DMS-2145630, NSF CCF-2212520, and BMBF in DAAD project 57616814.

## Table of notations

| | |
|---|---|
| $\Theta(d)$ | parameter space for linear classification 4 |
| $\Theta(d, n, m)$ | parameter space for classification by tropical rational function with fixed number of terms 9 |
| $\Theta(d, N)$ | parameter space of tropical signomials 16 |
| $\mathcal{H}_{\mathcal{D}}$ | hyperplane arrangement $\cup_{\mathbf{p} \in \mathcal{D}}(1, \mathbf{p})^{\perp}$ 4 |
| $P_{\mathcal{D}}$ | polytope whose normal fan is $\Sigma_{\mathcal{D}}$ 5 |
| $C_{\sigma}$ | sign vector associated to the cone $\sigma$ of a hyperplane arrangement 5 |
| $\Sigma_{\mathcal{D}}$ | polyhedral fan arising in linear classification induced by the data 5 |
| $\Sigma_{\mathcal{D}}(N)$ | activation fan: polyhedral fan arising in classification with tropical rational function with $N$ terms 18 |
| $P_{\mathcal{D}}(N)$ | activation polytope: polytopes whose normal fan is $\Sigma_{\mathcal{D}}(N)$ 18 |
| $\mathrm{err}_{C^*}(\theta)$ | loss in classification with parameters $\theta$ 6 |
| $C_{\mathcal{D}}(G)$ | activation cone: parameter cone with labeling $G$ 17 |
| $\Sigma_{\mathcal{D}}(n, m)$ | classification fan: a set of cones in parameter space 25 |
| $\Sigma_{\mathcal{D}}^0(n, m)$ | perfect classification fan: a subfan of the classification fan 27 |

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
