# OpenReview forum: "The Real Tropical Geometry of Neural Networks for Binary Classification"
_TMLR — Accepted by TMLR_

### Review · Reviewer_jJJV · 2024-07-18

**Summary Of Contributions:**

The paper considers deep neural networks from the perspective of tropical geometry. Its contributions are mainly theoretical, and the paper derives several theoretical results. For example:
- Theorem 2.3 characterizes a relationship between linear separability and connectedness of loss sublevel sets through walls and furthermore provides a bound on the error along a given path.
- Theorem 3.2 concerns whether deep ReLU networks can be represented by tropical rational functions.
- Theorem 4.2 provides some characterization of the decision boundary via Newton polytopes.
- Section 5 contains multiple results characterizing some geometric properties of activation fas and activation polytopes.
- Section 6 contains multiple results for classification fans, and an interesting result is Theorem 6.7, showing that the sublevel sets are not always strongly connected.

**Audience:**

Yes

**Broader Impact Concerns:**

The paper is theoretical and it does not have ethical implications to my knowledge.

**Claims And Evidence:**

Yes

**Requested Changes:**

See the above.

**Strengths And Weaknesses:**

This is a technically solid paper that proves many theorems. However, while the paper is well-written, the details are difficult to follow due to its technical nature. Below are more detailed comments, which I hope will be useful.

- Perhaps the paper could have a more suggestive title. The current title does not say more than the paper of Zhang et al. (2018), which is titled "Tropical Geometry of Neural Networks". Clearly, the present manuscript builds upon Zhang et al. (2018) and has deeper explorations in many aspects. Personally, I think the paper's content is more about classification and less about ReLU neural networks, so maybe including"classification" in the title is a good idea.

- The notion of *polyhedral fan* is not defined. I only know what it means through Wikipedia (https://en.wikipedia.org/wiki/Polyhedral_complex). However, the Wikipedia page does not define *maximal cones* of a polyhedral fan. It would be nice if the authors could add a definition of it, therefore making the paper more self-contained.
- Does the wall of a polyhedral fan always exist? It seems to be the case, hence it would be great if the author could comment on that when defining it.
- I found the one-dimensional example given after Theorem 2.3 quite interesting. To make the paper easier to comprehend, it would be great to have similar examples for walls and paths even before introducing Theorem 2.3. Even better: Give such examples any time when introducing a notation and if appropriate.
- In Theorem 3.2, the bound on $m$ seems to be very large. Could the authors comment on its tightness? On the other hand, how fast the parameter space $\Theta(d,2m, m)$ grow with respect to $m$? Answering this would give us some idea of whether the bound on $m$ is good or not.
- The phrase "stratification" or "stratified into" is not very clear and it is unclear what it means mathematically. Could the authors make it precise? Otherwise it is difficult to understand Theorem 4.4.

---

> ### Author Response · Authors · 2024-08-30
> **Response to Reviewer jJJV**
>
> We thank the reviewer for their thoughtful comments. We have revised the article to address their comments and offer responses below.
>
> > *"Perhaps the paper could have a more suggestive title"*
>
> We added "for Binary Classification"
>
> > *"The notion of polyhedral fan"*
>
> We added definitions and examples with illustrations, which we hope make the concepts more intuitive.
>
> > *"Does the wall of a polyhedral fan always exist?"*
>
> Walls are the boundaries between adjacent maximal cones. If there are at least two adjacent maximal cones, then the polyhedral fan will have walls. A polyhedral fan could consist of only one maximal cone, in which case it will have no walls. In principle one could also have more exotic types of fans where two or more maximal cones are not adjacent through a face of codimension one, and which in turn would also have no walls. In our present discussion however the polyhedral fans cover the entire parameter space, which is a vector space, and in turn if they have two or more maximal cones then they will also have walls.  We hope the added illustrations make this clearer.
>
> > *"one-dimensional example"*
>
> We expanded the example and added illustrations to make the concepts more intuitive.
>
> > *"tightness"*
>
> This is an interesting question. We explained before the theorem that we are presenting a generous bound. Anticipating interest in the tightness question, in the remark following the theorem we discuss the tightness question. We provide references of works investigating minimal decompositions of piecewise linear functions as differences of convex piecewise linear functions as well as results bounding the number of linear regions of functions represented by ReLU networks. There we also observe that for a projection of a variety, the degrees of the defining equations will generally be higher but there is no reasonable generally applicable bound. With this we tried to give as good a short overview as we could about what we think is the state of knowledge in regard to the tightness.
>
> > *"stratification is not very clear"*
>
> For clarity we have substituted the word 'stratification' with 'partition'.

---

> > ### Comment · Reviewer_jJJV · 2024-09-05
> > **Reply**
> >
> > Dear Authors,
> >
> > Thanks for your rebuttal, clarification, and the revised manuscript. The revision includes several modifications that enhance the clarity and accessibility of the paper (e.g., Examples 2.2 and 2.3, and some review on polyhedral geometry).

---

> > > ### Author Response · Authors · 2024-09-05
> > > **Re: Reply**
> > >
> > > Thank you for your time and the positive feedback on our revision.

---

### Review · Reviewer_CXWW · 2024-08-16

**Summary Of Contributions:**

The contributions are a collection of conceptualisations and theorems connecting concepts from tropical geometry to neural networks. See abstract for details. The key idea of this paper is find connections: that between activation patterns and sets of covectors of oriented and tropical oriented matroids is just one example.

**Audience:**

Yes

**Broader Impact Concerns:**

None.

**Claims And Evidence:**

Yes

**Requested Changes:**

I suspect many data-driven people would be unfamiliar
even with the terms such as fan, chambers, zonotopes, etc. (Note that
e.g the term "chambers" is used quite early on, but not defined until
page 4.) I suggest the authors explicitly point such people to where
they can learn the basics.

I encourage the authors to make an argument for theoretical results to
those, the majority, working on neural networks who are most concerned
with empirical performance. For example, the difference of two convex
piecewise linear functions is called a tropical rational function. But
what do we gain by so describing a difference of two convex piecewise
linear functions? It allows us to exploit known results from tropical
geometry: but how, if at all, can use them to get better results when
applying neural networks. But what one hopes from finding
a connection between neural networks and some area of mathematics is
to use theorems from that area to find previously unknown properties
of neural networks which can, at least in the long term, inform the
practice of neural networks.


SMALL POINTS

Prop 5.10: "span" should be in Roman font.
Proof of Prop 5.11: Odd to start with "Then"
Proof of Theorem 6.12:
"Thus in the bound is not attained" -> "Thus the bound is not attained"

The English is neither consistently American nor consistently
British. We have: "coloring", "neighbors" (US English) but also
"labelling" "analogues" (UK English). I personally am not too bothered
about this, but some might find it jarring.

**Strengths And Weaknesses:**

The two main issues for this submission are: (1) are the results
presented correct? and (2) are of they of sufficient
interest/significance for publication in TMLR?

Concerning (1) I found no errors in the proofs. Since I, unlike quite
a few other people, do not work on the combinatorics of neural
networks, I don't know whether some of these results have appeared
elsewhere. However, given the authors' evident extensive knowledge of
this field, I think it unlikely that any of the paper's purported
contributions are re-discoveries. (The authors note some rediscoveries
found elsewhere, so they are presumably aware of this 'risk'.)

Theorem 3.2 (the converse of the already-known Theorem 3.1) is a
useful, certainly non-obvious (to me) result.  The proof is long and
complex, but perhaps there is no simpler proof. I found Theorem 6.17
the most interesting, even though its truth is not too surprising.

The Introduction section does a nice job of introducing the research
area with an excellent, extensive set of references. Generally, the
writing is excellent and there are virtually no grammatical or
typographical errors. The examples that are dotted around the paper
are useful and welcome. However, the paper ends somewhat abruptly; I
think some pointers to future work would be a good idea.

One of the target audiences for the paper is: "A data-driven audience
with a background in machine learning interested in understanding the
underlying combinatorics". Such a person might have to do a lot of
'homework' before they could use this paper to gain such an
understanding.

---

> ### Author Response · Authors · 2024-08-30
> **Response to Reviewer CXWW**
>
> We thank the reviewer for the careful examination of our manuscript and their thoughtful comments.
> We hope to have addressed all items raised in the revision and the responses below.
>
> > *Strengths and weaknesses 1: "the paper ends somewhat abruptly; I think sime pointers to future work would be a good idea."*
>
> We added a conclusion section including pointers to future work as suggested by the reviewer.
>
> > *Strengths and weaknesses 2: "target audiences" and Requested changes 1 "I suggest the authors explicitly point such people to where they can learn the basics."*
>
> We added definitions in the early sections to make the text more accessible, along with pointers to references on the basics as suggested by the reviewer. We also added examples with illustrations to make the notions more intuitive.
>
> > *Requested changes 2 "make an argument for theoretical results"*
>
> First, we added a sentence early on in the introduction to emphasize this point.
> Then, in the new Conclusion section, we added commentary highlighting the role of the interdisciplinary discussion and some of the research we consider interesting and hope the presented discussion could help advance.
>
> > *Small points "span in roman font"*
>
> We are now using \operatorname\{span\}
>
> > *Small points "Proof of Prop 5.11: Odd to start with "Then""*
>
> We have rephrased this as "We have that"
>
> > *Small points "Proof of Theorem 6.12: "Thus in the bound is not attained" $\to$ "Thus the bound is not attained""*
>
> We corrected this.
>
> > *British vs American English.*
>
> We have substituted 'analogue' with 'analog' and 'labelling' with 'labeling'.

---

> > ### Comment · Reviewer_CXWW · 2024-09-05
> > **Revised version**
> >
> > Dear authors,
> >
> > Thanks for the revised version. You have clearly made a concerted effort to improve the accessibility of and motivation for the paper, as suggested by all 3 reviewers. This has resulted in an improved paper. I didn't explicitly put it in my required changes that "chambers" should be defined before it is used - could you make that change?

---

> > > ### Author Response · Authors · 2024-09-05
> > > **Re: Revised version**
> > >
> > > Thank you for the positive feedback. We have added the requested definition of "chambers" in the first occurrence on page 3 as "connected components of the complement of $\mathcal{H}_\mathcal{D}$".

---

### Review · Reviewer_62qL · 2024-08-22

**Summary Of Contributions:**

This paper studies a binary classification problem from the lens of tropical geometry. In particular, the work draws connections between characterization results on linear classifiers and piecewise linear neural networks using tropical geometry. The authors focus mainly on two directions: in the first, a decision boundary is fixed, which induces a subdivision of the parameter space into semialgebraic sets. In the second one, the emphasis is placed on the polyhedral fan induced by the activation patterns. At a higher level, the authors’ analyses allows them to make formal statements about which parameter sets can lead to perfect classification, but also describe the level-sets, which yields statements about optimality.

**Audience:**

Yes

**Broader Impact Concerns:**

No broader impact concerns.

**Claims And Evidence:**

Yes

**Requested Changes:**

**Requested Changes**

First and foremost, I think the work would greatly benefit from the visualization sequence I mentioned in the Weaknesses section. You could also visualize the Newton polytopes as I again think that will greatly help with readability, but I leave that to your discretion. For Section 5.2, I think again the paper would greatly benefit by the visualization of a tropical hyperplane, as I’m sure most readers are not familiar with those.

Another suggestion would be to rephrase the abstract. Currently, it is very dense and hard to convey the main contributions. The second and third sentences are hard to parse without context. Along a similar vein, it is a bit unclear exactly how all the results are connected, and essentially what is the impact. I believe that the description alone is interesting, however what are the takeaways and intuitions out of it? It would be very helpful for the work, I believe, to emphasize the contributions and the intuition/impact behind them, especially in the abstract and introduction.

**Comments**

I believe that Charisopoulos and Maragos (2018) also discussed maxout units and if memory serves also had results on the number of linear regions. I’m mentioning this as in the second paragraph of the introduction it is stated that this relation was first proposed by Montufar et al. (2022). Still on the topic of references, the earliest reference I’m aware of for function representations as differences of convex functions is by Hartman, “On functions representable as a difference of convex functions”, 1959. I think that one should be included alongside the other relevant references.

At the bottom of page 7 it is stated that “Given such a fixed architecture…”. The definition of $\operatorname{ReLU}(d_i)$ I’m assuming contains the piecewise linear functions when $m, n \to \infty$, and that hardly sounds like a fixed architecture. I think the authors agree with that statement, hence the parameter space is defined as $\Theta(d, n, m)$ in the case of neural networks, unlike the case of the data. I think there are two issues with this: one, I don’t think it’s exactly accurate to treat this infinite set as corresponding to a fixed architecture, and two I think it hurts readability; I think referring to that as a “fixed architecture” would greatly confuse machine learning audiences.

Before Definition 6.1 it is stated that given an activation pattern $G$, a coloring is induced. However, it seems that the coloring is independent of $G$ (since nodes are colored by $g$ or $h$). Concretely, the complaint is the coloring is independent of the activation patten (which is listed as $G$), and only depends on $V(G)$. The complaint is consistent with Lemma 6.3, which fixed a partition and then an activation pattern.

Finally, there is no conclusion, which seems peculiar. I think adding one would help once more to emphasize the contributions and help readers conceptualize the connections between the results.

**Minor questions**

I had a question about the oriented matroid: it seems that there needs to exist a cone that places all points in the data set on the boundary (C I). It’s not clear to me how this can be true in general configurations. Wouldn’t that require all the points to lie on the hull?

I’m also a bit skeptical of a statement in the proof of 6.17. An example is given of points in a general position. However, I would hardly call these points in general position, as $\\{p_2, p_3, p_4, p_5, p_6, p_8\\}$ are roughly colinear, and so are $\\{p_1, p_4, p_5, p_7, p_9\\}$.

**Mistakes**

- Second paragraph of second page “if the data is linearly separable” (data are plural, datum is singular, but understandably archaic)
- Same paragraph, “For networks with piecewise linear activation” (missing s)
- In Example 4.1 the definition of $f$ is different from the preceding section and is not a tropical signomial.
- Typo final line before 6.3 (should say functions).
- Below figure 5 “Thus, in the bound is not attained”. (remove in)

**Strengths And Weaknesses:**

**Strengths**

The paper is very well-written and most concepts are introduced intuitively. To my knowledge, the results are novel: certainly the connection between neural networks and tropical geometry has been explored, but I am not aware of any works formally making geometric statements about binary classification. I believe the formalism can serve as a playground for interesting follow up works.

I also found the paper very well referenced. Most of the works I’m familiar with that seemed relevant to me were cited, and many general references and reviews are provided for the interested reader.

Finally, I particularly enjoyed some of the proofs, for example Theorem 3.2 or Theorem 5.14. Both of them were very clearly presented.

**Weaknesses**

Clearly the work is very, very dense, and on a topic most people are not familiar with. It is noted that part of the intended audience is machine learning people with interest in combinatorics. However, most people coming from a machine learning background will be completely lost starting at Theorem 2.1. Some visualizations of the terms or further explanations would go a long way. Page 4 would greatly benefit from further visualizations. For example, for a point p in $\mathbb{R}^2$ the authors could show the hyperplane, in the next subfigure show the hyperplane arrangement (considering a few more points), next show the fan/cones, and then walls. That would greatly help readers create a mental picture of the concepts. Similarly, visualizations of the Newton polytope and its construction can greatly aid understanding.

There is a relative lack of coherence in the presentation of the paper. At times, sections feel like independent results with no direct connection. For example, it is not exactly clear to me how the two views that the authors study (the subdivision of the parameter set induced by the boundary and the subdivision by the activation patterns) are connected. Another example is practically Section 5.2. It feels disconnected from the rest of the text: here, tropical hyperplanes are considered, in contrast with the rest of the text, but if I’m understanding things correctly, Section 6 switches back to regular hyperplanes. Finally, Section 3 is a bit disconnected. Section 2 presents results in linear classifiers, and then analogues to these results are drawn in Sections 4 and Sections 5, 6. Section 3 does show the correspondence between piecewise linear neural networks and tropical geometry, which is needed. However, considering these correspondences have been shown before (at the level of showcasing them), the section feels unnecessarily long. Then Theorem 3.2 is presented, which feels very disconnected from the rest of the results. This is, in my opinion, an issue, as it is very easy to get lost with the big picture and what the end goal is, with Section 3 being a large contributor to this effect.

---

> ### Author Response · Authors · 2024-08-30
> **Response to Reviewer 62qL**
>
> We thank the reviewer for the thorough examination of our manuscript and their comments. We hope to have addressed these thoroughly in the revision and the responses shown below.
>
> >*Weaknesses 1 "visualizations of the terms” and Requested Changes 1 ”visualization sequence: point, hyperplane; several points, hyperplane arrangement, fan / cones, walls; Newton polytope; tropical hyperplane”.*
>
> We added a sequence of examples with visualizations of the hyperplane arrangements, polyhedral fan, and polytope. In addition, we included definitions of some of these concepts which we hope make the discussion more accessible.
>
> >*Weaknesses 2 ”coherence in the presentation” subdivision of parameters induced by boundary and by activation patterns; Section 5.2 feels disconnected from the rest; Section 3 is a bit disconnected and Requested Changes 2 ”rephrase the abstract ... very dense; how the results are connected; what are the takeaways and intuitions”*
>
> We hope that the added examples provide a more intuitive picture of the different concepts that are being discussed in the article and thereby that their coherence is more apparent. To further convey the coherence of the topics and the takeaways we also added a conclusion section including comments on possible future directions.
>
> In response to your specific comments: The subdivision of parameter space by the combinatorial type of the decision boundary is complementary to the subdivision by activation patterns, because the decision boundary is independent of the input data under consideration, as we pointed out following Theorem 4.4. We believe that further investigations of the interplay between these two subdivisions are an interesting prospect that we hope can be facilitated by the discussion we have presented. We added a corresponding comment in the conclusion section.
>
> We agree that the manuscript includes discussions in different directions, as we are drawing from different branches of discrete / combinatorial / tropical geometry to showcase how existing results can be transferred to the study of classification by ReLU neural networks. Section 5.2 explains how the discussion of hyperplane arrangements, activation polytopes, and oriented matroids in the case of linear classification is reflected in the activation polytope and extends to tropical oriented matroids. We also added a comment in the conclusion section.
>
> We took a careful consideration of the abstract and adapted the formulation to give a lighter start for the reader. Overall, it gives an accurate description of how we see this article. Particularly, as stated there, our findings extend and refine the connection between neural networks and tropical geometry by observing structures established in real tropical geometry, such as positive tropicalizations of hypersurfaces and tropical semialgebraic sets.
>
> >*Comments 1 "Charisopoulos and Maragos (2018)" and "Hartman (1959)".*
>
> We have added the reference 'Charisopoulos and Maragos (2018)' in the second paragraph of the introduction.
>
> Thank you for pointing us at the work of Harman (1959) titled 'On functions representable as a difference of convex functions'. We have added the reference. This work investigates the following relevant question:
>
> If $f_1(x)$ and $f_2(x)$ are d.c.\ functions (a function is called d.c. if it can be written as the difference of two convex functions), then so are the product $f_1(x)f_2(x)$, the quotient $f_1(x)/f_2(x)$ when $f_2(x) \neq 0$, and the composite $f_1(f_2(x))$ under suitable conditions on $f_2$.
>
> To also reference an explicit discussion of continuous piecewise linear functions as differences of convex piecewise linear functions, we have added a pointer to the work of Wang (2004) titled "General Constructive Representations for Continuous Piecewise-Linear Functions".
>
> We have removed the previous reference "Bitter (1970)" [misspelled as Butner (1970) in Springer Link]. Although this work also contains constructive theorems on the representation of functions as the difference of convex functions, their discussion of continuous piecewise linear functions focuses on the univariate case.
>
> >*Comments 2 ”bottom of page 7”.*
>
> Here we are referring only to those piecewise linear functions that can be represented by that particular ReLU network. We have rephrased for clarity as follows:
>
> Let $\mathbf{ReLU}(d_0,d_1,\ldots,d_{L-1},d_L)$ be the set of piecewise-linear functions that can be represented by a fully-connected ReLU network with $d_{0}=d$ inputs and $L$ layers of sizes $d_1,\ldots,d_L\in \mathbb{N}$, with $d_L = 1$.
> Given a fixed architecture of this kind, ...

---

> ### Author Response · Authors · 2024-08-30
> **Response to Reviewer 62qL part II**
>
> > *Comments 3 "Before Definition 6.1 it is stated that given an activation pattern $G$, a coloring is induced. However, it seems that the coloring is independent of $G$".*
>
> What we mean here is that the split $n+m=N$ induces a coloring of the nodes $[N]$ into $[n]\sqcup[m]$.
> To add clarity we have added "the split $n+m=N$" to the sentence:
>
> Given an activation pattern $G$ with nodes $V(G) = D \sqcup [N]$, the split $n+m=N$ induces a coloring of the nodes $[N]$ into $[n] \sqcup [m]$.
>
> > *Comments 4 "no conclusion".*
>
> We added a conclusion.
>
> > *Minor questions 1 "question about the oriented matroid: it seems that there needs to exist a cone that places all points in the data set on the boundary (C I)".*
>
> An interpretation of the oriented matroid axiom (CI) is that we can always assign the 'undecided' class $0$ (as opposed to $+$ or $-$) to every point in the dataset. This classification of the data is realized by the linear function $x\mapsto \langle 0, x\rangle$ (i.e., the zero function).
>
> Please observe that the cones that are being discussed on page 4 live in the space of parameters.
> An interesting aspect of the zero vector $\theta=0$ is that it lies on the boundary of every cone.
> Indeed, consider any possible classification of the data and suppose that it is realized by the linear function $x\mapsto \langle \theta, x\rangle$ with parameter $\theta$. Then, for any $\delta>0$ we have that $x\mapsto \langle \delta \theta, x\rangle$ produces the same classification. Moreover, if we send $\delta$ to zero, then $\delta \theta$ converges to the zero vector, and thus the zero vector is indeed at the boundary of the cone.
>
> > *Minor questions 2 "statement in the proof of 6.17 ... hardly call these points in general position as $\{p_2, p_3, p_4, p_5, p_6, p_8\}$ are roughly colinear".*
>
> We follow a standard notion but should have been more precise. We added the definition.
> A set of $M$ points in $\mathbb{R}^d$ is said to be in general position if no $k$ of them, $k\leq d+1$, are contained in an affine subspace of dimension strictly less than $k-1$.
>
> We also improved a typo (replacing the previous $D \in \Sigma_\mathcal{D}^k(n,m)$ with $D \in \Sigma_\mathcal{D}(n,m)$).
>
> > *Mistakes 1 "data are plural".*
>
> Thank you for pointing this out. As hinted in your comment, data is commonly written both with plural and singular verbs.
> We have tried to write data with singular verb consistently in the document.
>
> > *Mistakes 2 "activation".*
>
> We added the missing 's'.
>
> > *Mistakes 3 "Example 4.1 the definition of $f$ is different".*
>
> Thanks for pointing this out; we had inadvertently written $+$ instead of $\oplus$. We have corrected this.
>
> > *Mistakes 4 "Typo final line before 6.3 (should say functions)".*
>
> We corrected this.
>
> > *Mistakes 5 "Below figure 5 'Thus, in the bound is not attained'. (remove in)".*
>
> We corrected this.

---

> > ### Comment · Reviewer_62qL · 2024-09-05
> > **Revision**
> >
> > I thank the authors for addressing all of my concerns. I think Examples 2.2 and 2.3 are very helpful to visualize the concepts. My only comment would be that the subfigure captions in Figure 2 are identical (and the same as the caption of Figure 1(a)), which seems to be a mistake.

---

> > > ### Author Response · Authors · 2024-09-05
> > > **Re: Revision**
> > >
> > > Thank you for the positive feedback and pointing at the issue with the caption of Figure 2. Indeed, the caption in Figure 2 was incorrect. We have updates the sub captions to "Target dichotomy $C_1^∗$." and "Target dichotomy $C_2^∗$.".

---

### Decision · Action_Editor_3f5U · 2024-09-09

**Recommendation:** Accept as is

**Comment:**

All reviewers agree that the results are novel and insightful. The authors have addressed the reviewers' comments, including adding figures and further clarifications where appropriate. I am thus recommending acceptance.

**Audience:**

Part of the TMLR audience focusing on problems at the intersection of geometry and neural networks would be interested in these results.

**Claims And Evidence:**

This paper provides connections between ReLU networks and tropical geometric. In particular, it carries out a theoretical study of the partition of the parameter space of a ReLU networks as a semialgebraic set of the parameter space of tropical rational functions. The manuscript provides geometric and combinatorial insights into the classification fan associated with this problem, extending the connection between Relu networks tropical geometry.

---

> ### Author Response · Authors · 2024-09-18
> **Camera ready version**
>
> Dear AE, Thank you for your work handling our TMLR article! We have now uploaded the camera ready version.